# Beyond Seeing: Evaluating Multimodal LLMs on Tool-Enabled Image Perception, Transformation, and Reasoning

## Abstract

Multimodal Large Language Models (MLLMs) are increasingly applied in real-world scenarios where user-provided images are often imperfect, requiring active image manipulations such as cropping, editing, or enhancement to uncover salient visual cues. Beyond static visual perception, MLLMs must also *think with images*: dynamically transforming visual content and integrating it with other tools to solve complex tasks. However, this shift from treating vision as passive context to a manipulable cognitive workspace remains underexplored. Most existing benchmarks still follow a *think about images* paradigm, where images are regarded as static inputs. To address this gap, we introduce VISTOOLBENCH, a vision tool-use reasoning benchmark that rigorously evaluates MLLMs' ability to perceive, transform, and reason across complex visual–textual tasks under the *think with images* paradigm. VISTOOLBENCH comprises 1,204 challenging, open-ended vision tasks (603 single-turn, 601 multi-turn) spanning across five diverse domains, each paired with detailed rubrics to enable systematic evaluation. Our evaluation shows that current MLLMs struggle with tasks requiring effective integration of vision and general-purpose tools. Even the strongest model (GPT-5-think) reaches only 18.44% pass rate. We further observe divergent tool-use behaviors, with OpenAI models benefiting from diverse image manipulations while Gemini-2.5-pro shows no improvement. By introducing the first benchmark centered on *think with images*, VISTOOLBENCH offers critical insights for advancing visual intelligence in MLLMs.

## 1 Introduction

Multimodal Large Language Models (MLLMs), which integrate visual and textual understanding, have advanced rapidly in recent years and achieve impressive performance on a wide range of vision–language tasks, including image grounding (Rasheed et al., 2024; Zhang et al., 2024), image-based science problems (Zou et al., 2024; Lu et al., 2023; Yan et al., 2025b), visual question answering (Kuang et al., 2025; Liu et al., 2023a), optical character recognition (OCR) (Chen et al., 2025; Huang et al., 2025a), and spatial reasoning (Yang et al., 2025; Wu et al., 2025a; Tang et al., 2025). Current frontier MLLMs can interpret, describe, and reason about complex visual scenes in natural language, narrowing the gap between human and machine perception (Yin et al., 2024).

However, real-world use cases often need sophisticated processing of visual input and MLLMs may need to dynamically interact with them. For example, users often submit photos that are rotated, underexposed, cluttered, or poorly framed. In such cases, MLLMs need to work through multiple reasoning steps, including image manipulation, information extraction via tool-usage to arrive at the final answer. Standard models without tool support typically struggle under such degradations, revealing the need for active visual manipulation to solve such harder tasks. This distinction has recently been framed as *thinking about images* versus *thinking with images* recently (Su et al., 2025c; OpenAI, 2025). Current multimodal benchmarks mainly adopt the former paradigm of *thinking about images* and focus on perception and reasoning over fixed, static images. The latter, by contrast, emphasizes interactive, tool-augmented reasoning, where models autonomously manipulate visual inputs (e.g., cropping, editing, or enhancing) to extract fine-grained information for problem solving. Equipping MLLMs with such vision-specific tools during evaluation is therefore essential for robust

Table 1: Comparison of VISTOOLBENCH with representative multi-modal benchmarks.

| Benchmark | Vision Tool | General Tool | Rubrics | Expert-Curated | Reasoning | Multi-turn |
|---|---|---|---|---|---|---|
| ScienceQA (Lu et al., 2022) | ✗ | ✗ | ✗ | ✓ | ✓ | ✗ |
| MathVista (Lu et al., 2023) | ✗ | ✗ | ✗ | ✓ | ✓ | ✗ |
| MMMU (Yue et al., 2024) | ✗ | ✗ | ✗ | ✓ | ✓ | ✗ |
| V* (Wu & Xie, 2024) | ✗ | ✗ | ✗ | ✗ | ✗ | ✗ |
| GTA (Wang et al., 2024a) | ✓ | ✓ | ✗ | ✗ | ✗ | ✗ |
| ChartQA (Wang et al., 2024b) | ✗ | ✗ | ✗ | ✓ | ✓ | ✗ |
| MMDU (Liu et al., 2024a) | ✗ | ✗ | ✗ | ✗ | ✓ | ✓ |
| m & m's (Ma et al., 2024a) | ✓ | ✓ | ✗ | ✗ | ✗ | ✗ |
| VISTA (Scale AI, 2025) | ✗ | ✗ | ✓ | ✓ | ✓ | ✗ |
| VISTOOLBENCH (Ours) | ✓ | ✓ | ✓ | ✓ | ✓ | ✓ |

and generalizable reasoning. These capabilities transform visual inputs from passive perception into a dynamic cognitive workspace, enabling MLLMs to tackle tasks that would otherwise be infeasible. Existing benchmarks remain inadequate for capturing this dimension (Su et al., 2025c).

To bridge this gap, we introduce VISTOOLBENCH, a challenging benchmark for vision and general-purpose tool-use that systematically evaluates how well MLLMs can perceive, transform, and reason about images under the *think with image* paradigm[1]. Our key design principles are as follows:

- **Non-trivial visual perception.** Critical visual content is not easily accessible, models must apply appropriate image transformations (e.g., cropping, editing, or enhancement) to extract key visual details for better reasoning.

- **Realistic task settings.** Both prompts and images are designed to reflect practical, real-world scenarios rather than synthetic or overly simplified cases, ensuring that the benchmark closely mirrors real-world user needs.

- **Implicit tool-use requirements.** Tasks do not explicitly instruct the model which tool to use; instead, models must infer when and how to invoke tools based on contextual cues, making evaluation more faithful to realistic usage.

- **Multi-step, compositional reasoning.** Tasks are designed such that require combining visual transformations with multi-step reasoning (e.g., applying a sequence of tools, integrating extracted information, and synthesizing results), testing model's ability to plan and execute complex workflows.

To reflect real-world applications, we design five complementary task categories that capture diverse aspects of model performance. Two cases are tailored to single-turn tasks and three targeting multi-turn tasks (Sec. 2.1). Each task is authored by qualified contributors with proper training and undergoes multiple review stages to ensure high-quality data samples (Sec. 2.2). To further capture complex and realistic scenarios, tasks are presented in an open-ended format. Further, each task is accompanied by a set of rubrics that span multiple dimensions. These criteria are used for systematic model evaluations (Sec. 2.3). Finally, VISTOOLBENCH supports both vision tools, which expose a flexible Python API for generating image manipulation code and re-ingesting processed images into the reasoning process, and general-purpose tools: web search, Python interpreter, calculator, and historical weather lookup, aiding in retrieval and computation for more advanced tasks (Sec. 2.4).

By releasing VISTOOLBENCH and its accompanying evaluation toolkit, we aim to catalyze the development of MLLMs that seamlessly integrate image perception, tool use, and reasoning into a unified competency stack. Our contributions are four-fold:

1. **A first *think with image*-oriented multimodal benchmark.** VISTOOLBENCH is the first benchmark to systematically evaluate MLLMs on tasks that require active visual manipulation to solve complex reasoning problems.

2. **Rubric-based, multi-dimensional evaluation.** Moving beyond binary correctness or exact string matching, we design detailed rubrics that capture partial credit across multiple categories. This richer scoring framework provides nuanced diagnostic insights into both the strengths and limitations of MLLMs.

---

[1]See Table 1 for an overall comparison between VISTOOLBENCH and existing multimodal benchmarks.

| Statistic | Number |
|---|---|
| Total questions | 1,204 |
| - STEM | 238 (19.7%) |
| - Medical | 238 (19.7%) |
| - Finance | 243 (20.2%) |
| - Sports | 240 (20.0%) |
| - Generalist | 245 (20.4%) |
| Single-turn | 603 (50.1%) |
| - Region Switch Q&A | 281 (46.6%) |
| - Hybrid Tool-use | 322 (53.4%) |
| Multi-turn | 601 (49.9%) |
| - Follow-up Test | 198 (32.9%) |
| - Temporal Reasoning | 205 (34.2%) |
| - Progressive Reasoning | 198 (32.9%) |
| Total number of rubrics | 7,777 |
| Total number of images | 2,893 |
| Average prompt length | 48.41 |
| Average answer length | 128.93 |

Table 2: Key statistics of VISTOOLBENCH.

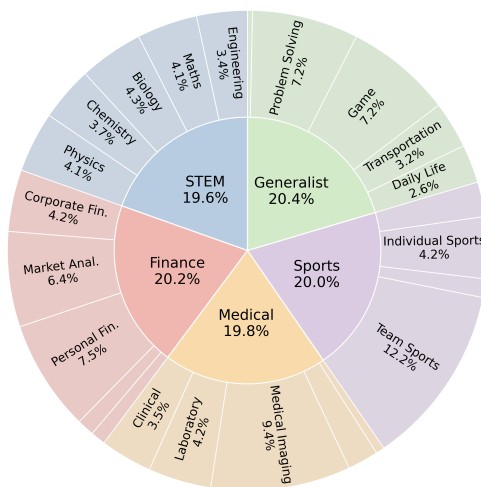

Figure 1: Topic distribution.

3. **Large-scale and systematic evaluations.** We evaluate 16 representative MLLMs with function-calling capabilities, covering both reasoning and non-reasoning, as well as open- and closed-source models, under consistent settings. To support this evaluation, we developed a dedicated toolkit for vision tool use that allows models to access transformed images during reasoning and preserves complete tool-use trajectories. Our results reveal substantial performance gaps, with all models achieving pass rates below 20%.

4. **Comprehensive error and tool-use analysis.** We provide a detailed failure analysis and an in-depth examination of tool-use behaviors. Most failures stem from visual perception errors, highlighting the inefficiency of current models in using vision tools to extract key content. Furthermore, our study reveals divergent tool-use behaviors: the top-performing model, GPT-5, leverages diverse image manipulations to achieve clear gains over its no-tool baseline, whereas Gemini-2.5-pro does not gain improvement from tool access. These findings underscore the critical role of effective tool use in advancing MLLMs' performance.

## 2 VISTOOLBENCH

In this section, we present VISTOOLBENCH, a multi-domain visual tool-use benchmark designed to evaluate MLLMs on challenging, real-world reasoning tasks. The benchmark includes both single-turn and multi-turn interactions and incorporates five complementary task categories to probe different aspects of MLLM capabilities. Tasks are open-ended to reflect realistic scenarios, and each task is accompanied by detailed rubrics to support systematic evaluation. Table 2 summarizes the key statistics of VISTOOLBENCH, while Figure 1 illustrates the topic distribution of tasks.

### 2.1 TASK CATEGORY

We design five complementary task categories, each targeting a critical aspect of real-world use case. Together, they assess not only visual perception but also the efficiency of tool use and the depth of multimodal reasoning. These categories are designed to mirror practical user scenarios, requiring models to think with images rather than relying solely on static perception.

**Region-Switch Q&A (Single-Turn).** The model answers a reasoning task that draws on information from multiple, spatially distinct regions of interest (RoIs) within a single image. Critical details may be small or dispersed, requiring the model to correctly identify, crop, and focus on relevant RoIs while disregarding irrelevant content. Success in this category reflects spatial selectivity, accurate region localization, and effective tool use for RoI extraction. Figure 2 illustrates a single-turn benchmark task for this evaluation focus in the generalist domain.

**Hybrid Tool Reasoning (Single-Turn).** The model must combine both vision-specific tools (e.g., python image processing) with other general-purpose tools (e.g., calculator, Python interpreter, or web search) to solve complex, multi-step tasks. This category evaluates the model's ability to accu-

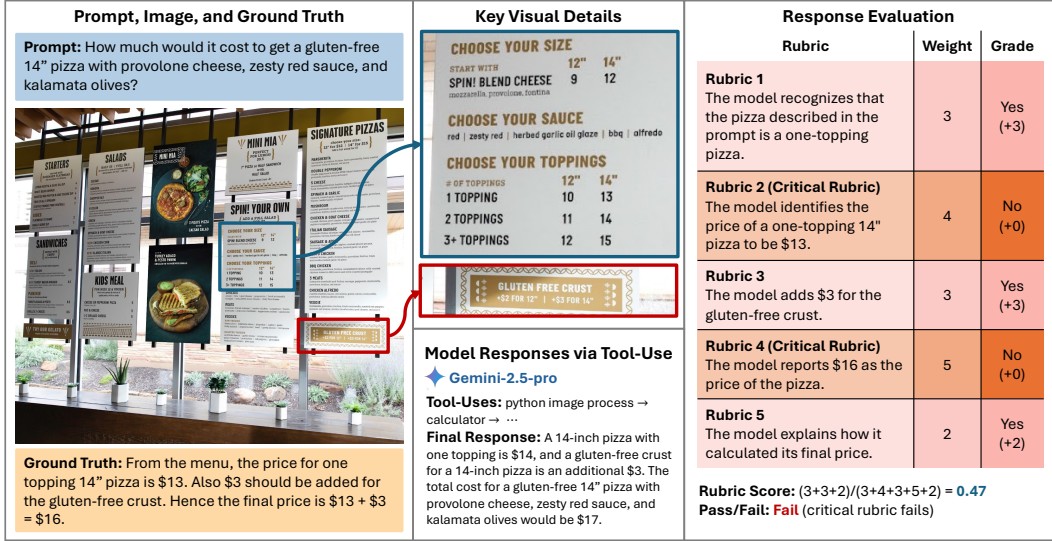

Figure 2: Demonstration example from VISTOOLBENCH (single-turn, generalist domain, region switch Q&A). The key visual content needed to solve the task is distributed across different regions of the image, requiring the model to crop multiple regions of interest (RoIs) for accurate perception and reasoning. Each task is paired with a detailed set of rubrics to evaluate model's responses. From these rubrics, we derive both a weighted rubric score between 0 and 1 and a binary pass/fail outcome, depending on whether critical rubrics are satisfied.

rately call multiple tools and orchestrate heterogeneous tool outputs into a coherent reasoning chain towards solving complex visual reasoning tasks.

**Follow-up Test (Multi-Turn).** In this scenario, the first-round user query is intentionally under-specified or ambiguous. The model must engage in clarifying dialogue and ask follow-up questions before producing an answer. This tests conversational proactivity, uncertainty management, and the ability to self-correct, all essential skills for real-world deployments where users may provide incomplete or noisy instructions.

**Temporal Visual Reasoning (Multi-Turn).** Here the model reasons over a sequence of images across multiple turns, requiring it to detect temporal changes, track motion, or infer causal relationships among multiple images. Tasks may involve following the progression of an event, monitoring evolving states, or interpreting multi-step visual instructions.

**Progressive Visual Reasoning (Multi-Turn).** The model solves a series of interdependent questions about the same image, where later queries could build upon earlier answers. This requires the model to maintain internal consistency, remember prior outputs, and construct a layered understanding of the scene. Success in this category demonstrates long-horizon reasoning, contextual memory, and the ability to sustain a coherent reasoning trajectory. Additional examples of VISTOOLBENCH are provided in Appendix D.2 and in our anonymous HuggingFace repository [link].

## 2.2 DATA COLLECTION

All VISTOOLBENCH tasks are authored by human contributors with diverse domain expertise. To ensure benchmark quality and realism, we adopt a rigorous multi-stage data collection pipeline:

1. **Contributor Training.** Contributors are first introduced to the project scope, task categories, and submission requirements. They are instructed to provide a task prompt, input image, reference answer, reference tool-use chain, and a set of evaluation rubrics.

2. **Initial Task Design.** Drawing on their domain expertise, contributors design tasks by selecting the appropriate domain and aligning with the specified task category. Each submission includes a text prompt and associated image(s), a golden answer, a reference tool-use chain that demonstrates a valid solution path, and well-defined evaluation rubrics.

3. **Initial Model Response Grading.** Contributors are presented with responses from three representative models (o3, Gemini-2.5-pro, and o4-mini) to their designed tasks. They then grade these responses against the rubrics[2]. A task is selected only if at least two of the three models fail, thereby ensuring that the benchmark captures genuinely challenging cases.

4. **First-Round Review.** A reviewer evaluates each task for realism, necessity of dynamic image-based reasoning, correctness of the reference answer, and appropriateness of the rubrics. Tasks with minor issues may be revised, while those that are fundamentally unsound (e.g., not requiring genuine visual-tool use) are discarded.

5. **Second-Round Review.** A second independent reviewer validates the first-round decision, ensuring consistency and reliability across the benchmark.

6. **Final Integration.** Tasks that pass both review stages are incorporated into the benchmark, ensuring high quality, broad domain coverage, and diverse reasoning requirements.

This layered pipeline ensures that every task is original, realistic, and rigorously validated, resulting in a benchmark that robustly evaluates genuine visual intelligence.

## 2.3 RUBRIC-BASED EVALUATIONS

We adopt rubric-based evaluation to capture nuanced aspects of model performance beyond correctness alone (Arora et al., 2025; Starace et al., 2025; Scale AI, 2025; Lin et al., 2024; Sirdeshmukh et al., 2025; Fast et al., 2024; Gunjal et al., 2025; Guo et al., 2025). For each task, contributors provide a comprehensive set of rubric criteria to assess model responses. A rubric item may range from specific factual requirements (e.g., providing the correct final short answer) to broader aspects of desirable behavior (e.g., presenting key intermediate steps). Specifically, rubric items are organized into five main categories for VISTOOLBENCH:

1. **Visual Understanding**: Correct identification, extraction, and explanation of relevant visual elements such as text, objects, or spatial relationships.

2. **Truthfulness**: Accuracy of all factual statements and correctness of the final answer.

3. **Instruction Following**: Precise adherence to the task requirements specified in the prompt.

4. **Reasoning**: Use of clear, step-by-step logic with justified inferences and calculations.

5. **Presentation**: Clarity, coherence, structure, and appropriate formatting of the response.

Each rubric criterion is assigned a weight $w \in \{1, 2, 3, 4, 5\}$ by the task contributor, where higher weights indicate greater importance. To evaluate a model's response, an auto-grader examines each rubric criterion independently and determines whether the response satisfies it. If the criterion is met, full points are awarded; otherwise, no points are given. The weighted rubric score for a task is then calculated as the sum of satisfied items, normalized by the total rubric weights. Rubric items with $w \geq 4$ are designated as **critical rubrics**. These rubrics typically correspond to essential aspects such as truthfulness and key visual understanding, and satisfying them indicates that the model has solved the task in a substantive way. Failure to meet any critical rubric results in an overall failure for that task, and this binary outcome is used to compute a task-level accuracy in VISTOOLBENCH. All rubrics together contribute to the weighted rubrics score, enabling more fine-grained analysis. The right panel of Figure 2 illustrates rubric-based grading.

## 2.4 TOOL SET

To enable broad, tool-augmented reasoning under the *think with images* paradigm, VISTOOLBENCH exposes a standardized API with six carefully selected tools: `python_image_processing`, `python_interpreter`, `web_search`, `calculator`, `browser-get-page-text`, and `historical_weather`. This compact yet diverse toolset spans core capabilities for image manipulation, computation, retrieval, and domain-specific lookups. Among them, the vision-specific tool `python_image_processing` is particularly central: it supports arbitrary manipulations such as cropping, editing, and brightness/contrast adjustments, enabling models to iteratively refine

---

[2]These annotations serve as golden human labels for rubric-based evaluation and enable subsequent analysis of LLM-as-judge versus human-judge alignment (Sec. B.9), facilitating scalable evaluation.

Table 3: APR (%) results of the evaluated models across domains. Domain abbreviations: **Med** (Medicine), **Fin** (Finance), **Spt** (Sports), and **Gen** (Generalist). The best results in each column are highlighted with a red background, and the second-best results are highlighted in blue.

| Model | Overall | Single-Turn | | | | | Multi-Turn | | | | |
|---|---|---|---|---|---|---|---|---|---|---|---|
| | | STEM | Med | Fin | Sprt | Gen | STEM | Med | Fin | Sprt | Gen |
| **Open-Source Models** | | | | | | | | | | | |
| Llama4-Maverick | 1.16 | 4.31 | 1.65 | 1.63 | 0.84 | 0.81 | 0.00 | 0.85 | 0.00 | 0.83 | 0.83 |
| Llama4-Scout | 1.65 | 1.72 | 2.47 | 0.81 | 3.36 | 2.42 | 0.00 | 2.38 | 0.85 | 0.00 | 1.90 |
| Pixtral-large | 1.43 | 2.89 | 4.13 | 1.63 | 3.36 | 1.61 | 0.90 | 0.00 | 0.00 | 0.00 | 0.00 |
| **Closed-Source Models** | | | | | | | | | | | |
| GPT-4o | 3.24 | 1.74 | 3.31 | 3.25 | 6.72 | 2.42 | 5.74 | 3.42 | 0.83 | 3.31 | 1.65 |
| GPT-4.1 | 5.73 | 4.35 | 12.40 | 2.44 | 8.40 | 4.03 | 7.38 | 5.98 | 1.67 | 6.61 | 4.13 |
| o3 | 15.53 | 29.57 | 24.79 | 13.01 | 18.49 | 24.19 | 11.48 | 6.84 | 13.33 | 6.61 | 7.44 |
| o4-mini | 11.21 | 16.52 | 22.31 | 8.94 | 12.61 | 19.35 | 4.10 | 10.26 | 5.83 | 5.79 | 5.79 |
| GPT-5 | 17.05 | 29.31 | 26.45 | 25.20 | 15.13 | 28.23 | 14.66 | 5.31 | 11.21 | 6.78 | 6.72 |
| GPT-5-think | 18.44 | 26.96 | 24.79 | 24.39 | 24.37 | 29.84 | 13.11 | 7.69 | 14.17 | 10.74 | 8.26 |
| Gemini-3-pro | 27.39 | 41.37 | 34.71 | 48.78 | 38.66 | 34.68 | 18.03 | 11.97 | 19.17 | 14.05 | 12.40 |
| Gemini-2.5-pro | 10.72 | 17.24 | 15.70 | 17.07 | 14.29 | 18.55 | 5.88 | 7.14 | 2.59 | 5.26 | 2.48 |
| Gemini-2.5-flash | 6.20 | 7.76 | 5.79 | 3.25 | 10.08 | 9.68 | 2.38 | 3.57 | 0.00 | 3.57 | 2.78 |
| Claude-sonnet-4 | 4.07 | 3.48 | 8.26 | 3.25 | 5.88 | 5.65 | 4.10 | 4.27 | 2.50 | 2.48 | 0.83 |
| Claude-opus-4.1 | 4.32 | 6.03 | 9.09 | 3.25 | 6.72 | 6.45 | 3.28 | 5.13 | 0.00 | 2.48 | 0.83 |
| Claude-sonnet-4-think | 4.49 | 1.72 | 10.74 | 4.07 | 7.56 | 10.48 | 0.82 | 3.41 | 2.50 | 1.65 | 1.65 |
| Claude-opus-4.1-think | 4.64 | 4.31 | 9.09 | 6.50 | 10.08 | 6.45 | 1.64 | 1.71 | 0.86 | 4.13 | 0.94 |
| Nova-Premier | 1.86 | 4.31 | 5.79 | 0.00 | 1.68 | 1.61 | 2.54 | 0.98 | 0.00 | 1.65 | 0.00 |

visual inputs and use images as an interactive scratchpad. This versatility makes it the cornerstone of our benchmark's *think-with-images* setup. Detailed tool descriptions are provided in Appendix E.

## 3 EXPERIMENT RESULTS

**Evaluation Setup.** We conduct large-scale evaluations using LiteLLM's function-calling interface (LiteLLM). Models are given the supported tools and invoke them by emitting the corresponding call arguments. For conventional (non-vision) tools, outputs are textual and are appended to the dialogue as a `tool` message. Vision tools, in contrast, return transformed images. We observe that placing encoded images directly in a `tool` message does not make them perceptible to the model. To ensure effective re-ingestion of visual results, after a vision tool executes and saves its outputs, we insert an additional `user` message containing the encoded image(s). This preserves models' ability to *think with images*, wherein newly produced images inform later reasoning. We set a cap of 20 tool calls per task, while human reference trajectories usually need less than 5 tool calls.

**Baseline Models.** We benchmark 16 representative MLLMs with function-calling capabilities, covering both reasoning and non-reasoning as well as open- and closed-source models. A complete list of models, along with their endpoints and configuration details, is provided in Appendix C.1.

**Evaluation Metrics.** We report two main metrics derived from rubric-based judgments: *Average Pass Rate (APR)* and *Average Rubric Score (ARS)*. (i) APR. Each task specifies a set of *critical* rubrics. A model's response *passes* only if all it passes all the critical rubrics; otherwise it *fails*. APR is then the percentage of tasks that pass across the dataset. (ii) ARS. Each rubric is assigned an integer weight from 1 to 5 by the contributor to indicate its importance. For a model's response on a task, we compute a weighted proportion of satisfied rubrics: the total weight of satisfied rubrics divided by the total weight of all rubrics for that task, as a weighted rubric score. The dataset-level ARS is then the average of these per-task rubric scores. Formal definitions and implementation details are provided in Appendix B.

### 3.1 MAIN RESULTS

We present the overall APR results in Table 3. Figure 3 represents the APR across task categories for top five performing models on VISTOOLBENCH. We make the following observations.

**VISTOOLBENCH is highly challenging.** From Table 3, it is clear that VISTOOLBENCH poses a challenging vision tool-use reasoning benchmark. Specifically, even the best-performing model,

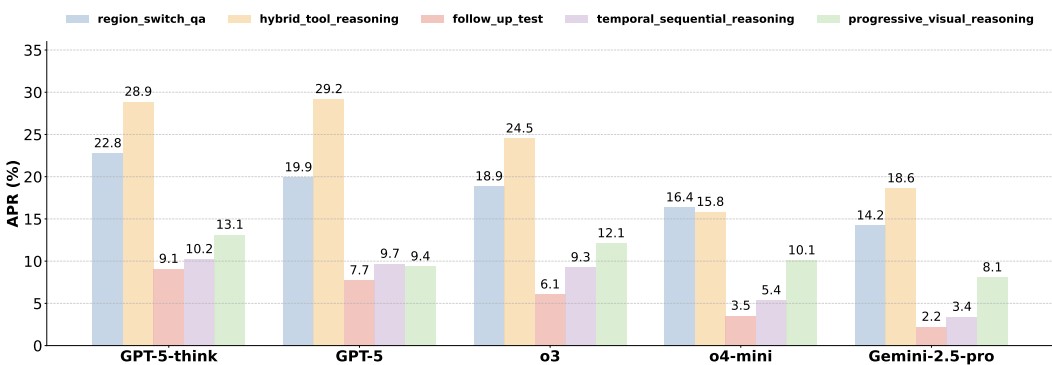

Figure 3: APR across task categories for the top five models on VISTOOLBENCH. Bars report model performance on each category, with exact APR values labeled above.

GPT-5-think, achieves only an 18.44% overall pass rate, and 11 out of 16 MLLMs obtain APRs below 10%. This highlights the limitations of current MLLMs and underscores the substantial room for improvement on visual-reasoning tasks where critical content is not directly accessible and must be extracted through vision tools.

**OpenAI models outperform others.** Models from OpenAI, including GPT-5, GPT-5-think, o3, and o4-mini, lead performance with APRs above 11%, showing a clear margin over competing models. This may be attributed to their specific training for solving *think with images* tasks (OpenAI, 2025). On the other hand, Gemini-2.5-pro also demonstrates relatively strong performance (10.72%), due to its advanced visual perception capabilities (Comanici et al., 2025).

**Multi-turn tasks are more difficult than single-turn tasks.** From Table 3 and Figure 3, we see that single-turn tasks (region-switch Q&A, hybrid-tool reasoning) achieve higher pass rates compared to multi-turn tasks (follow-up test, temporal sequential reasoning, and progressive visual reasoning). This is expected, as multi-turn tasks involve 2 to 5 conversational turns, introducing more opportunities for errors and compounding reasoning challenges.

### 3.2 TOOL-USE ANALYSIS

In this section, we perform tool-call analysis on the evaluated models. First, we quantify tool-use behavior with three descriptive metrics computed from execution traces: **proactivity**, **tool-call success rate**, and **tool-call volume**. Proactivity is the fraction of tasks in which at least one tool is invoked, capturing a model's tendency to integrate tools into its reasoning. Success rate measures the proportion of invocations that return a schema-valid, non-empty result, reflecting model's adherence to tool specifications. Tool-call volume is the average number of tool calls per task, indicating how heavily a model relies on tools to solve tasks. Precise definitions are provided in Appendix B.5.

**Models use tools ineffectively.** Figure 4 plots model APR against tool-use proactivity (left) and average tool-call volume per task (right). We observe a clear positive correlation: models that use tools more proactively and with higher call volume tend to achieve better performance. OpenAI models cluster in the upper right, showing both higher APR and stronger tool-use behaviors, while other families (Claude, Gemini, Llama, Nova, Pixtal) remain concentrated in the lower ranges with limited tool use and weaker performance. These findings indicate that most current MLLMs underexploit tool capabilities, and that proactive, consistent tool use is a key driver of success. More detailed tool-call analyses are provided in Appendix B.6.

**Vision tool is the most called tool**. Figure 5 presents the tool-use distributions of the top three APR models (o3, GPT-5-think, and Gemini-2.5-Pro). Across all three models, more than 50% of tool calls are to the `python_image_processing` (92% for GPT-5), underscoring that image manipulation is the primary operation required to solve tasks in our benchmark. Other tools such as `python_interpreter`, `web_search`, and `calculator` are invoked less frequent, reinforcing the inherently *think with images* nature of VISTOOLBENCH.

**Models exhibit diverse patterns of image manipulation.** We do not constrain models to a predefined set of image operations; instead, they can invoke arbitrary manipulations via Python through

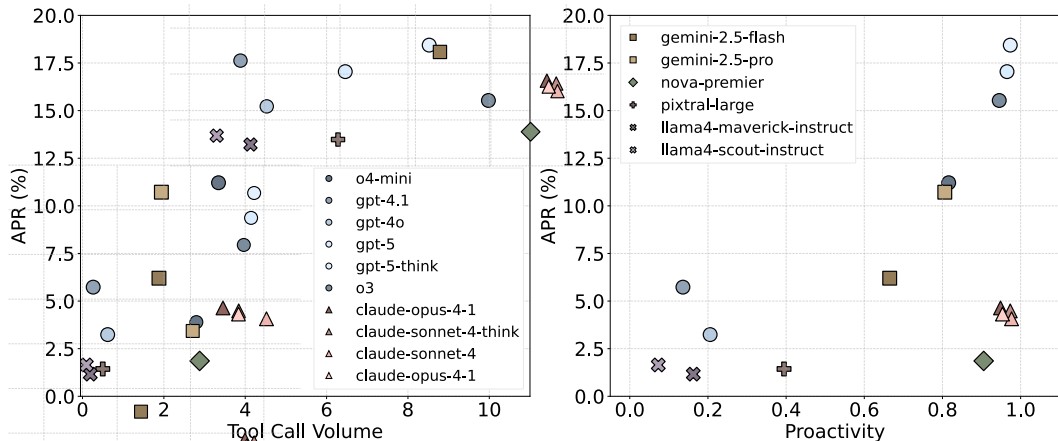

Figure 4: Relationship between APR and tool-use behaviors across models. Left: APR versus tool-use proactivity. Right: APR versus average tool call volume per task.

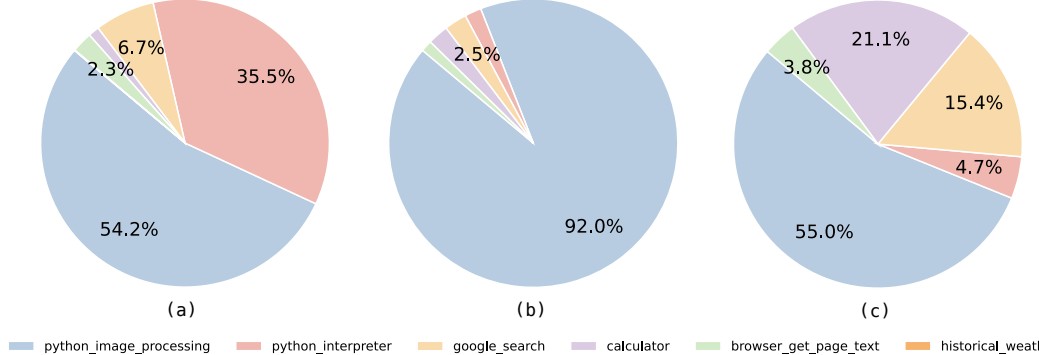

Figure 5: Tool-use distribution of top three APR models. Each subplot shows the percentage share of tool calls. Total number of tool calls: (a) o3: 6004; (b) GPT-5: 3805; (c) Gemini-2.5-pro: 1082.

the vision tool. To better understand tool usage in practice, we group the issued calls into eight representative categories: cropping, resizing, rotation, flipping, brightness adjustment, contrast adjustment, editing (e.g., annotation, inpainting, drawing), and others. Figure 6 reports the distribution of these operations across the top-performing models. GPT-5-think and GPT-5 stand out with both higher volumes and broader diversity of manipulations, reflecting more active exploration of tool capabilities[3]. o3 and o4-mini also perform frequent manipulations but with narrower operation profiles, while Claude-opus-4-1 and Gemini-2.5-pro show comparatively limited usage. Illustrative cases and qualitative edge examples are provided in Appendix E.3.

## 3.3 ERROR ANALYSIS

In this section, we analyze errors for three representative models (GPT-5, Gemini-2.5-pro, and Claude-opus-4.1) using the following four general categories: visual-perception error, reasoning, calculation, and others. Table 4 summarizes the distribution of error types. Across all models, visual-perception errors are the most common failure mode. By contrast, calculation mistakes are rare, and reasoning errors occur only occasionally. More discussions are provided in Appendix B.7.

## 3.4 ABLATION STUDY

Figure 7 compares APR across four different settings: tool-use with strong and weak system prompts, no vision tool, and no tools. For GPT-5, removing tools or weakening the prompt causes significant performance drops ($\approx 11 - 14\%$), showing clear gains from tool-augmented reasoning.

---

[3]Although GPT-5 and GPT-5-think issue fewer tool calls than o3 (see the first column of Table 8), they perform more image manipulations overall as shown in Figure 6. A deeper inspection shows that GPT-5 and GPT-5-think often execute multiple operations within a single vision-tool call, indicating greater tool-use efficiency compared to o3 and contributing to their superior performance. See a demonstration in Appendix E.4.

| Model | Error Type | Percentage |
|---|---|---|
| GPT-5 | visual perception | 71.73% |
| | reasoning | 11.56% |
| | calculation | 2.79% |
| | others | 13.92% |
| Gemini-2.5-pro | visual perception | 78.01% |
| | reasoning | 12.24% |
| | calculation | 5.74% |
| | others | 4.01% |
| Claude-opus-4.1 | visual perception | 82.11% |
| | reasoning | 9.36% |
| | calculation | 1.84% |
| | others | 6.69% |

Table 4: Error type statistics.

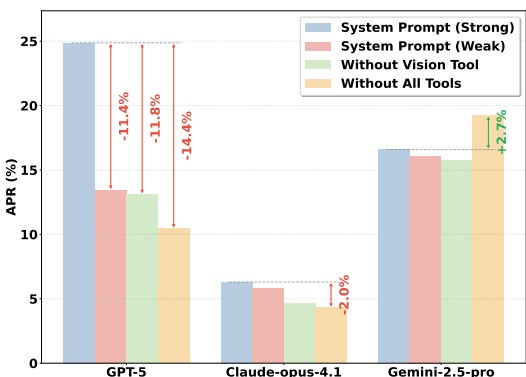

Figure 6: Image manipulation operations counts.

Surprisingly, Gemini-2.5-pro performs better **without tools** (+2.7%), implying the tools have counter-effect on its performance. In contrast, GPT-5 benefits from tools, boosting its performance further when enabled. These divergent trends likely reflect training differences: GPT-5 appears reinforced on tool-centric workflows, relying on iterative edits to offset weaker perception, whereas Gemini-2.5-pro, with stronger native vision, was likely exposed to fewer tool-use demonstrations and thus degrades when applying unnecessary tool calls.

Figure 7: Ablation study results.

## 4 RELATED WORK

Most existing multimodal benchmarks remain limited to *think about images*: they focus on passive visual Q&A without active interactions (Rahmanzadehgervi et al., 2024; Yue et al., 2024; Wang et al., 2024b; Lu et al., 2023; Zou et al., 2024; Wu & Xie, 2024; Scale AI, 2025), restrict tool use to basic operations like cropping (Wang et al., 2024a; Ma et al., 2024a), or evaluate tool-agnostic multi-turn dialogues without essential integration (Liu et al., 2024a; Yan et al., 2025a); see also (Li et al., 2024) for a broader overview.

Recent efforts toward teaching MLLMs to *think with images* include prompting-based methods that leverage language mediation, visual input manipulation, or expert integration (Zeng et al., 2022; Yang et al., 2023b; Wu et al., 2024a; Liu et al., 2025a), supervised fine-tuning to enable tool invocation and intrinsic manipulations such as cropping, grounding, and dynamic attention (Liu et al., 2023b; Shao et al., 2024), and reinforcement learning for adaptive exploration and tool orchestration (Wang et al., 2025c; Fan et al., 2025; Huang et al., 2025b; Zheng et al., 2025). Despite these advances, evaluations remain constrained to visually passive benchmarks. VISTOOLBENCH addresses this gap by providing a rigorous testbed for genuine visual intelligence under the *think with images* paradigm. See broader discussion in (Su et al., 2025c) and more related work in Appendix A.

## 5 CONCLUSION

In this work, we introduced VISTOOLBENCH, a benchmark designed to evaluate MLLMs under the *think with image* paradigm. Unlike prior efforts that treat images as static inputs, our benchmark emphasizes active visual manipulation through tool use across diverse domains and task types. Our experiments show that current MLLMs continue to struggle on tasks requiring dynamic visual manipulations. We hope VISTOOLBENCH will drive progress toward models that can more effectively think with images and tackle challenging real-world scenarios.

**Ethics and Reproducibility Statement.** We worked with experienced human annotators to design the prompts and evaluation rubrics. All annotators were engaged as independent contractors and were compensated at rates consistent with fair labor standards and applicable local regulations. Participation was entirely voluntary, and annotators could decline or discontinue tasks at any time. Importantly, no personally identifiable or sensitive information was collected during the study.

We have taken several steps to ensure the reproducibility of our results. Specifically, we provide detailed dataset examples along with the hyperparameters used in our experiments. In addition, both the dataset and the evaluation codebase have been submitted and will be publicly released to facilitate reproducibility.

**LLM Usage Acknowledgment.** We acknowledge the use of large language models (LLMs) to assist in improving clarity and polishing the language of certain sentences. All substantive ideas, analyses, and conclusions are solely the responsibility of the authors.

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

CONTENTS

# A MORE ON RELATED WORK

## A.1 EXISTING MLLM BENCHMARKS

**Existing Multi-modal Benchmarks.** Existing visual reasoning benchmarks are typically limited: (1) Passive Visual Q&A: focusing on "look-and-answer" type of tasks without any active interactions Bi et al. (2025); Rahmanzadehgervi et al. (2024); Yue et al. (2024); Wang et al. (2024b); Lu et al. (2023); Zou et al. (2024); Wu & Xie (2024); Scale AI (2025). (2) Superficial Tool Use: including only basic tools like cropping, failing to test deeper image manipulation Wang et al. (2024a); Ma et al. (2024a). (3) Tool-Agnostic Conversations: evaluation multi-turn dialogue but without integrating essential tools Liu et al. (2024a); Yan et al. (2025a). See (Li et al., 2024) for a more comprehensive survey of multimodal benchmarks.

## A.2 LEARNING TO THINK WITH IMAGES

**Prompting Methods** Prompt-based methods enable LMMs to coordinate predefined visual tools without parameter updates, turning static inputs into actively explorable workspaces via in-context learning. Early work like Socratic Models (Zeng et al., 2022), PromptCap (Hu et al., 2022), and MM-REACT (Yang et al., 2023b) showed that language can mediate collaboration, allowing text-only LLMs to function as visual reasoners by orchestrating vision experts through dialogue or targeted captions. Other approaches manipulate inputs directly: visual prompt engineering (e.g., red circles) (Shtedritski et al., 2023; Zhang et al., 2025b), Visualization-of-Thought (Wu et al., 2024a), and Visual Abstract Thinking (Liu et al., 2025a) enhance perception by highlighting, abstracting, or structuring images, while ZoomEye (Shen et al., 2024), ViCrop (Zhang et al., 2025c), Chain-of-Spot (Liu et al., 2024b), and VisuoThink (Wang et al., 2025b) extend this into systematic zooming and multimodal tree search. Finally, specialized experts can be integrated via prompting: Set-of-Mark (Yang et al., 2023a) leverages segmentation, DetToolChain (Wu et al., 2024b) structures detection reasoning, DyFO (Li et al., 2025) uses MCTS for adaptive focus, and Visual Thoughts (Cheng et al., 2025) frames expert outputs as cached "visual thoughts." Collectively, these works demonstrate that careful prompting—through language mediation, input manipulation, or expert integration—can significantly enhance multimodal reasoning without retraining.

**Think with Image via SFT** Supervised fine-tuning (SFT) is a primary method for teaching LMMs to use external tools or internal visual skills by training on datasets that demonstrate tool invocation and integration. For external orchestration, models like LLaVA-Plus (Liu et al., 2023b), TACO (Ma et al., 2024b), and VTS-V (Bai et al., 2025b) learn to compose tools (e.g., OCR, calculators) and follow procedural chains of reasoning. For internal manipulation, frameworks such as CogCoM (Qi et al., 2024), VGR (Wang et al., 2025a), and UniVG-R1 (Bai et al., 2025a) show how SFT can endow models with intrinsic capabilities like cropping, grounding, or fine-grained perception. Finally, SFT also cultivates dynamic visual attention: Visual CoT (Shao et al., 2024), IVM (Zheng et al., 2024), CMM-CoT (Zhang et al., 2025a), selective revisitation (Chung et al., 2025), and V* (Wu & Xie, 2024) demonstrate how training with attentional annotations transforms attention into an active, controllable skill. Across these directions, SFT provides the supervision that converts high-level reasoning into executable visual actions.

**Think with Image via Reinforcement Learning.** Reinforcement learning (RL) advances beyond supervised imitation by enabling models to optimize policies for visual reasoning through interaction and feedback. Foundational studies such as Jigsaw-R1 (Wang et al., 2025c), V-Triune (Ma et al., 2025), and VisionReasoner (Liu et al., 2025c) established that RL improves generalization over SFT and supports unified frameworks for diverse perception tasks. Building on this, GRIT (Fan et al., 2025), Point-RFT (Ni et al., 2025), and Seg-Zero (Liu et al., 2025b) demonstrated policies that embed spatial cues (e.g., bounding boxes, positional prompts) into reasoning, forming multimodal chains of thought. RL has also enabled active tool orchestration: VisTA (Huang et al., 2025b) learns tool-selection policies, Chain-of-Focus (Zhang et al., 2025d) and ACTIVE-o3 (Zhu et al., 2025) develop adaptive zooming and region proposals, while DeepEyes (Zheng et al., 2025) achieves interleaved multimodal reasoning without SFT. Exploration is further incentivized by Pixel-Reasoner (Su et al., 2025a), while VILASR (Wu et al., 2025b) leverages drawing-based reasoning, and OpenThinkIMG (Su et al., 2025b) introduces the first open-source end-to-end RL framework for invoking diverse external tools. Collectively, these approaches move LMMs from passive viewers to active visual agents.

# B  MORE ON EVALUATION RESULTS

## B.1  EXPANDED BASELINE MODEL COMPARISON

In this section, we now report single-turn results for a broader set of MLLMs, including the latest frontier models Gemini-3-pro, Claude-sonnet-4.5, and Claude-sonnet-4.5-thinking; mid-sized models Qwen3-VL-30B and Qwen3-VL-235B; and smaller models such as Thyme-7B (Zhang et al., 2025e) and Deepeyes-7B (Zheng et al., 2025), covering both commercial and open-source vision-language systems. The results for single-turn tasks are shown below (we also include the results for GPT-5-think for comparison).

| Model | Overall | STEM | Medical | Finance | Sports | Generalist |
|---|---|---|---|---|---|---|
| GPT-5-think | 26.04 | 26.96 | 24.79 | 24.39 | 24.37 | 29.84 |
| Gemini-3-pro | **39.64** | **41.37** | **34.71** | **48.78** | **38.66** | **34.68** |
| Claude-sonnet-4.5 | 7.40 | 5.26 | 10.99 | 3.70 | 7.59 | 8.70 |
| Claude-sonnet-4.5-thinking | 8.68 | 9.47 | 12.12 | 5.68 | 4.55 | 10.99 |
| Qwen3-VL-30B | 1.00 | 0.00 | 1.65 | 0.81 | 1.68 | 0.81 |
| Qwen3-VL-235B | 1.66 | 0.00 | 5.79 | 0.00 | 2.52 | 0.00 |
| Thyme-7B | 1.82 | 2.59 | 2.48 | 0.81 | 0.00 | 3.23 |
| Deepeyes-7B | 3.32 | 2.59 | 3.31 | 4.07 | 3.36 | 3.23 |

Table 5: Single-turn APR (%) results for additional baseline models across domains.

It is worth noting that the most recent model Gemini-3-pro achieves the current state of the art on VisToolBench, outperforming GPT-5-think by a substantial margin on single-turn tasks. We have updated Table 3 and Table 6 with the multi-turn results of Gemini-3-pro as well. Despite these improvements, overall performance remains modest: Claude-sonnet-4.5 models obtain an APR below 10%, and all mid-sized to small models score below 5%. Interestingly, several small open-source models—particularly Thyme-7B and Deepeyes-7B, which are explicitly trained for the think-with-image paradigm still perform poorly, likely due to their limited parameter scale. However, Deepeyes-7B achieves a 3.32% overall APR, outperforming several larger models such as Llama-Maverick, Pixtral-large, Qwen3-VL-30B, and Qwen3-VL-235B, suggesting that targeted training for vision-tool-use can yield meaningful gains. Nevertheless, substantial room for improvement remains before current models can saturate VISTOOLBENCH.

## B.2  MORE ON EVALUATION METRICS

In this section, we provide a detailed description on how to evaluate a model's response based on the rubrics. For each task $i \in \{1, 2, \cdots, N\}$ in VISTOOLBENCH, we perform the following steps:

1. **Obtain Model's Final Response.** We generate model's final response with {prompt, image} pair while allowing model to use a set of predefined tool sets.

2. **Rubric Grading.** For each rubric criterion $j \in \{1, 2, \cdots, N_i\}$, we use an LLM to grade whether the rubric criterion is met based on the model's response and the rubric criterion.

3. **Weighted Rubric Score for Task** $i$. Then we compute a final score for task $i$ using the following weighted average sum:

$$s_i = \frac{\sum_{j=1}^{N_i} \mathbb{1}_{r_{ij}} w_{ij}}{\sum_{j=1}^{N_i} w_{ij}}, \tag{1}$$

where $w_{ij} \in \{1, 2, 3, 4, 5\}$ is the assigned weight for each rubric criterion item, and $\mathbb{1}_{r_{ij}}$ is an indicator representing whether criterion $j$ is met.

The final score $S$ for the whole benchmark is then computed as the mean value of each task's score:

$$S = \frac{1}{N} \sum_{i=1}^{N} s_i. \tag{2}$$

## B.3 RUBRICS WEIGHTS

To ensure consistent evaluation, each rubric item in VISTOOLBENCH is assigned a weight $w \in \{1, 2, 3, 4, 5\}$ that reflects its relative importance. Table 6 outlines the five weight levels, ranging from *incidental* stylistic preferences to *critical* elements that determine overall task validity. Task contributors are instructed to assign rubric weights in accordance with these guidelines.

| Rubric Weight | Description |
|---|---|
| Critical (5) | Non-negotiable. This element must be present for the answer to count as valid. Failing it implies the task has failed, regardless of other strengths. |
| Significant (4) | Central to success. Leaving this out degrades output quality or creates confusion about how the task was solved. |
| Moderate (3) | Meaningfully important. Affects clarity or correctness; its absence weakens the answer but does not make it invalid. |
| Minor (2) | Adds polish or completeness but is not essential. Omitting it slightly lowers quality without breaking the core solution. |
| Incidental (1) | Marginally relevant. A nice-to-have detail or stylistic preference that does not impact whether the model solves the task. |

Table 6: Rubric criteria weights used in VISTOOLBENCH. Higher weights indicate greater importance, with critical rubrics determining task-level success.

## B.4 AVERAGE RUBRIC SCORE RESULTS

Table 7 presents the detailed rubric score for both single-turn and multi-turn tasks. It can be seen that APR and ARS are positively correlated, models have higher APR also have higher ARS.

Table 7: ARS results of the evaluated models across domains. Domain abbreviations: **Med** (Medicine), **Fin** (Finance), **Spt** (Sports), and **Gen** (Generalist). The best results in each column are highlighted with a red background, and the second-best results are highlighted in blue.

| Model | Overall | Single-Turn | | | | | Multi-Turn | | | | |
|---|---|---|---|---|---|---|---|---|---|---|---|
| | | STEM | Med | Fin | Sprt | Gen | STEM | Med | Fin | Sprt | Gen |
| *Open-Source Models* | | | | | | | | | | | |
| Llama4-Maverick | 0.1954 | 0.1942 | 0.1627 | 0.1174 | 0.1293 | 0.1435 | 0.2821 | 0.3188 | 0.2199 | 0.2094 | 0.1821 |
| Llama4-Scout | 0.1841 | 0.1988 | 0.1430 | 0.1223 | 0.1759 | 0.1621 | 0.2715 | 0.2733 | 0.2110 | 0.2123 | 0.1577 |
| Pixtral-large | 0.1123 | 0.1486 | 0.1165 | 0.1092 | 0.1381 | 0.1147 | 0.1564 | 0.1106 | 0.0870 | 0.0717 | 0.0757 |
| *Closed-Source Models* | | | | | | | | | | | |
| GPT-4o | 0.2194 | 0.2083 | 0.1628 | 0.1300 | 0.2337 | 0.1713 | 0.3092 | 0.2608 | 0.1943 | 0.2962 | 0.2290 |
| GPT-4.1 | 0.3302 | 0.2744 | 0.3027 | 0.2223 | 0.3311 | 0.2708 | 0.4330 | 0.4179 | 0.3014 | 0.4309 | 0.3210 |
| o3 | 0.4195 | 0.4337 | 0.4108 | 0.3418 | 0.3701 | 0.3929 | 0.4656 | 0.5255 | 0.4156 | 0.4556 | 0.3867 |
| o4-mini | 0.4054 | 0.4039 | 0.4097 | 0.3482 | 0.3596 | 0.4057 | 0.4464 | 0.4839 | 0.3749 | 0.4286 | 0.3925 |
| GPT-5 | 0.4700 | 0.4808 | 0.4404 | 0.4370 | 0.3771 | 0.4703 | 0.5590 | 0.5229 | 0.4497 | 0.4956 | 0.4739 |
| GPT-5-think | 0.4623 | 0.4473 | 0.4113 | 0.3720 | 0.4410 | 0.4646 | 0.5677 | 0.5267 | 0.4471 | 0.5031 | 0.4433 |
| Gemini-3-pro | 0.5880 | 0.5960 | 0.5881 | 0.7060 | 0.5934 | 0.5818 | 0.6041 | 0.5281 | 0.6029 | 0.5610 | 0.5160 |
| Gemini-2.5-pro | 0.4002 | 0.3747 | 0.3270 | 0.3777 | 0.3661 | 0.3840 | 0.4352 | 0.4999 | 0.3979 | 0.4701 | 0.3802 |
| Gemini-2.5-flash | 0.2331 | 0.2268 | 0.1781 | 0.1725 | 0.2045 | 0.1945 | 0.4032 | 0.4762 | 0.2650 | 0.4226 | 0.3050 |
| Claude-sonnet-4 | 0.2783 | 0.2560 | 0.2603 | 0.2163 | 0.2431 | 0.2421 | 0.3363 | 0.3857 | 0.2824 | 0.2988 | 0.2660 |
| Claude-opus-4.1 | 0.2951 | 0.2521 | 0.2507 | 0.2233 | 0.2558 | 0.2656 | 0.3739 | 0.4351 | 0.2605 | 0.3395 | 0.2977 |
| Claude-sonnet-4-think | 0.2828 | 0.2069 | 0.2667 | 0.2444 | 0.2270 | 0.2653 | 0.3495 | 0.4034 | 0.3101 | 0.2865 | 0.2689 |
| Claude-opus-4.1-think | 0.2550 | 0.2423 | 0.2574 | 0.2671 | 0.2862 | 0.2736 | 0.2555 | 0.2956 | 0.2024 | 0.2439 | 0.2203 |
| Nova-Premier | 0.2259 | 0.1984 | 0.2082 | 0.1530 | 0.2076 | 0.1929 | 0.3377 | 0.3325 | 0.1907 | 0.2364 | 0.2214 |

## B.5 TOOL-USE EVALUATION METRICS

We evaluate tool-use behaviors based on execution traces recorded by the evaluation harness. Let $\mathcal{T}$ denote the set of tasks ($N = |\mathcal{T}|$). For a given task $i \in \mathcal{T}$, let $\mathcal{T}_i = \{c_{i1}, c_{i2}, \dots\}$ represent the (ordered) multiset of tool invocations initiated by the model.

**Tool-Call Proactivity.** The fraction of tasks in which the model invoked at least one tool:

$$\text{Proactivity} \;=\; \frac{\left| \{\, i \in \mathcal{T} \mid |\mathcal{T}_i| > 0 \,\} \right|}{|\mathcal{T}|}. \tag{3}$$

Higher values indicate more frequent tool integration, though proactivity may reflect either beneficial or redundant calls.

**Tool-Call Success Rate.** The fraction of valid tool calls across all invocations:

$$\text{Success Rate} \;=\; \frac{\displaystyle\sum_{i \in \mathcal{T}} \sum_{c \in \mathcal{T}_i} \mathbb{1}[\text{valid}(c)]}{\displaystyle\sum_{i \in \mathcal{T}} |\mathcal{T}_i|}, \tag{4}$$

where $\mathbb{1}[\text{valid}(c)]$ is an indicator function that equals $1$ if tool call $c$ succeeds and $0$ otherwise. We determine validity by inspecting tool outputs: error messages imply $\text{valid}(c) = 0$, while all other outputs imply $\text{valid}(c) = 1$. This metric measures how reliably a model adheres to tool specifications.

**Tool-Call Volume.** The average number of tool calls per task:

$$\text{Volume} \;=\; \frac{1}{N} \sum_{i \in \mathcal{T}} |\mathcal{T}_i|, \tag{5}$$

where $N$ is the total number of tasks and $|\mathcal{T}_i|$ denotes the number of tool calls made in task $i$.

### B.6 Tool-Use Analysis

Table 8 reports the detailed tool-use metrics for all evaluated models. By combining these results with model performance (APR in Table 3 and ARS in Table 7), we make the following key observations:

1. **More tool calls do not necessarily translate to better performance.** For instance, o3 makes the largest number of calls (16,116), yet performs worse than GPT-5 (10,213 calls) and GPT-5-think (13,429 calls).

2. **High proactivity does not guarantee strong results.** Claude models exhibit very high proactivity, yet their performance remains poor, with overall APR below 5% and ARS values under 0.3.

3. **Low proactivity and low call volume generally correlate with poor performance.** For example, the Llama models, Pixtral-large, GPT-4o, and GPT-4.1 all demonstrate relatively low proactivity and correspondingly weak performance.

### B.7 More on Error Analysis

To better understand model weaknesses, we categorize failure cases in VISTOOLBENCH into four major error types:

**Visual Perception Error.** Errors arising from a model's inability to correctly perceive, interpret, or extract relevant information from images. Typical cases include misidentifying objects, overlooking salient regions, or unable to extract key visual content.

**Reasoning Error.** Errors caused by flawed logical inference or problem-solving steps in the model's reasoning process. Typical cases include the model's responses contain invalid intermediate steps, contradictions, or logically inconsistent conclusions.

**Calculation Error.** Errors stemming from incorrect arithmetic or symbolic computations. These include mistakes in basic arithmetic, misapplication of formulas, or numerical inaccuracies in intermediate or final answers.

Table 8: Tool call analysis of the evaluated models. The best results in each column are highlighted with a red background, and the second-best results are highlighted in blue.

| Model | Total # | Single-Turn | | | Multi-Turn | | |
|---|---|---|---|---|---|---|---|
| | | Proactivity | Success Rate | Volume | Proactivity | Success Rate | Volume |
| **Open-Source Models** | | | | | | | |
| Llama4-Maverick | 315 | 0.1625 | 0.6609 | 0.19 | 0.2562 | 0.1550 | 0.33 |
| Llama4-Scout | 110 | 0.0729 | 0.6140 | 0.09 | 0.0749 | 0.0943 | 0.09 |
| Pixtral-large | 1315 | 0.3947 | 0.6733 | 0.50 | 0.8170 | 0.4236 | 1.69 |
| **Closed-Source Models** | | | | | | | |
| GPT-4o | 1024 | 0.2056 | 0.6273 | 0.61 | 0.2463 | 0.2688 | 1.08 |
| GPT-4.1 | 535 | 0.1359 | 0.8038 | 0.26 | 0.2728 | 0.3263 | 0.63 |
| o3 | 16116 | 0.9453 | 0.8587 | 9.98 | 0.9750 | 0.3696 | 16.80 |
| o4-mini | 5337 | 0.8159 | 0.8199 | 3.34 | 0.8835 | 0.3138 | 5.53 |
| GPT-5 | 10212 | 0.9652 | 0.8555 | 6.46 | 0.9883 | 0.3548 | 10.53 |
| GPT-5-think | 13429 | 0.990 | 0.8660 | 7.45 | 0.9950 | 0.3105 | 13.79 |
| Gemini-2.5-pro | 3366 | 0.8060 | 0.7331 | 1.94 | 0.9251 | 0.3191 | 3.66 |
| Gemini-2.5-flash | 2313 | 0.6650 | 0.8605 | 1.87 | 0.4809 | 0.3144 | 1.96 |
| Claude-sonnet-4 | 6941 | 0.9768 | 0.9252 | 4.52 | 0.9917 | 0.4048 | 7.01 |
| Claude-opus-4-1 | 6538 | 0.9536 | 0.9524 | 3.83 | 0.9900 | 0.4151 | 7.03 |
| Claude-sonnet-4-think | 8704 | 0.9735 | 0.9352 | 3.84 | 0.7671 | 0.3946 | 3.91 |
| Claude-opus-4-1-think | 2082 | 0.9486 | 0.9524 | 3.45 | 0.9950 | 0.3941 | 6.72 |
| Nova-premier | 5109 | 0.9055 | 0.5444 | 2.88 | 0.9734 | 0.1823 | 5.61 |

**Other Errors.** Residual errors that do not fit the above categories. Examples include incomplete responses, refusals, or hit the maximum tool-calls.

### B.8 FINE-GRAINED TOOL-FAILURE ANALYSIS

In this section, we provide the fine-grained tool-use failure analysis. Specifically, our analysis covering eleven categories of common tool-related errors, including visual-perception mistakes, incorrect cropping, incorrect rotation/flip operations, OCR errors, distraction-induced mistakes, and incorrect tool selection. We perform such analysis on 100 common failure tasks on three evaluated models: GPT-5, Claude-opus-4.1, and Gemini-2.5-pro.

| Failure Type | Description |
|---|---|
| Visual Misinterpretation | The model misperceives or misunderstands original or processed visual content. |
| OCR Error | The model incorrectly reads, extracts, or transcribes text from an image. |
| Distraction | The model's response is distracted by the tool-usage. |
| Tool Execution Error | The model applies it incorrectly, producing invalid tool observations. |
| Incorrect Tool Selection | The model chooses the wrong tool for the task. |
| Incorrect Cropping | The model crops the wrong region, uses incorrect crop boundaries. |
| Incorrect Rotation/Flipping | The model rotates or flips the image in the wrong direction or when rotation was not required. |
| Incorrect Enhancing | The model incorrectly applies enhancement operations (brightness, contrast, sharpness). |
| Incorrect Editing | The model performs an inappropriate editing action. |
| Incorrect Resizing | The model resizes the image incorrectly, including wrong scaling factors or unintended aspect ratio changes. |
| Other | Any failure modes not captured by the predefined categories. |

Table 9: Fine-grained categories of vision tool-use failure.

To support large-scale evaluation, we prompt GPT-5 with the question, the model's tool-use chain, the golden answer, and the model's actual answer to generate an initial failure-category prediction. Human supervision is then applied to filter out any inappropriate or incorrect categories. We examined three frontier models and select 100 common failure tasks for this fine-grained tool-use failure analysis. As shown in Table **??**, all three frontier models exhibit substantial and systematic tool-use failures. For instance, visual perception remains the dominant failure mode across all models

(GPT-5: 75, Gemini-2.5-Pro: 56, Claude-Opus-4.1: 88), and errors such as incorrect cropping and execution failures also occur frequently (e.g., GPT-5: 32 cropping errors and 33 execution errors).

| Failure Type | GPT-5 (%) | Gemini-2.5-pro (%) | Claude-opus-4.1 (%) |
|---|---|---|---|
| Visual Misinterpretation | 34.40 | 32.56 | 39.46 |
| OCR Error | 11.01 | 12.21 | 7.62 |
| Distraction | 8.26 | 4.07 | 10.76 |
| Tool Execution Error | 15.14 | 12.79 | 6.28 |
| Incorrect Tool Selection | 2.75 | 3.49 | 5.83 |
| Incorrect Cropping | 14.68 | 12.21 | 17.94 |
| Incorrect Rotation/Flipping | 5.05 | 5.23 | 5.38 |
| Incorrect Enhancing | 1.83 | 0.58 | 1.35 |
| Incorrect Editing | 0.00 | 0.00 | 0.45 |
| Incorrect Resizing | 0.46 | 0.00 | 0.00 |
| Other | 6.42 | 16.86 | 4.93 |
| **Total Count** | **218** | **172** | **223** |

Table 10: Fine-grained vision tool-use failure percentages and total counts (percentage values shown without symbol).

We emphasize that reasonably good visual perception is essential for enabling effective tool use in our benchmark. For example, some tasks require the model to first localize a small object or region. However, as our examples illustrate later, models often fail to propose accurate cropping coordinates in one shot, leading to zooming into irrelevant regions, and this makes models to perform multiple unnecessary tools calls (see examples in Appendix F.3), which may propagate downstream errors. Overall, we believe that this fine-grained error analysis provides process-level insight into where current MLLMs struggle, complementing our existing metrics and strengthening the evaluation framework.

### B.9 LLM-AS-JUDGE VS. HUMAN-AS-JUDGE

To enable large-scale evaluation of VISTOOLBENCH, we employ LLMs as automatic judges and compare their assessments against human annotations. Table 12 reports the overall alignment rate on a subset of VISTOOLBENCH, as well as breakdowns for objective and subjective rubrics across three judge models.

Table 11: Alignment rates of different LLMs when serving as judges.

| Judge Model | Overall | Objective Rubrics | Subjective Rubrics |
|---|---|---|---|
| o4-mini | 0.8807 | 0.9170 | 0.7396 |
| GPT-4.1 | 0.8818 | 0.8983 | 0.8177 |
| GPT-4o | 0.8701 | 0.8916 | 0.7865 |

All models achieve high alignment on objective rubrics, while alignment on subjective rubrics is comparatively lower, reflecting the inherent ambiguity of subjective evaluation. Nevertheless, overall alignment rates remain close to 90%, underscoring the reliability of LLM-as-judge for our benchmark. We choose to use o4-mini as our judge model for the main experimental study.

### B.10 SUBJECTIVE RUBRIC ALIGNMENT VIA MAJORITY VOTING

We further examine how to enhance the grading accuracy of subjective rubrics. In particular, we compare the performance of our primary judge model, o4-mini, against a majority-vote ensemble

composed of three judge models: o4-mini, GPT-4.1, and GPT-4o. Our results show that incorporating majority voting yields a notable improvement, increasing human–model alignment by an additional 5.32% on subjective rubric evaluations.

Table 12: Alignment rates for different judging strategies.

| Judging Strategy | Overall | Objective Rubrics | Subjective Rubrics |
|---|---|---|---|
| o4-mini | 0.8807 | 0.9170 | 0.7396 |
| Majority Vote | 0.8860 | 0.9149 | 0.7928 (↑ 5.32%) |

## C  MORE ON EXPERIMENTAL SETUP

In this section, we provide more details on our experimental setup.

### C.1  BASELINE MODELS

In Table 13, we list the evaluated models' endpoints and detailed parameter settings. Whenever possible, we set the temperature to 0. For o3, o4-mini, GPT-5, and GPT-5-think, the temperature is set to 1. Since GPT-5 is inherently a reasoning model, we use its default reasoning effort (`medium`) for GPT-5, and GPT-5-thinking corresponds to the `high` reasoning effort. For o3 and o4-mini, we adopt their default reasoning effort (`medium`). For Claude's thinking mode, we set the reasoning budget to 5000 tokens. All other model API hyperparameters are kept at their default settings without further customization.

Table 13: Model Endpoints and Hyperparameter Setup

| Model Provider | Model Endpoint | Hyperparameter |
|---|---|---|
| OpenAI | o3 | `reason_effort = "medium"` |
| | o4-mini | `reason_effort = "medium"` |
| | gpt-4.1 | `temperature = 0.0` |
| | gpt-4o | `temperature = 0.0` |
| | GPT-5 | `reason_effort = "medium"` |
| | GPT-5-thinking | `reason_effort = "high"` |
| Anthropic | claude-sonnet-4-20250514 | `temperature = 0.0` |
| | claude-sonnet-4-20250514 (thinking) | `thinking_budget = 5000` |
| | claude-opus-4-1 | `temperature = 0.0` |
| | claude-opus-4-1 (thinking) | `thinking_budget = 5000` |
| Google | gemini-3-pro-preview | `temperature = 0.0` |
| | gemini-2.5-pro | `temperature = 0.0` |
| | gemini-2.5-flash | `temperature = 0.0` |
| Meta | llama4-maverick-instruct | `temperature = 0.0` |
| | llama4-scout-instruct | `temperature = 0.0` |
| Amazon | nova-premier-v1:0 | `temperature = 0.0` |
| Mistral AI | pixtral-large-latest | `temperature = 0.0` |

### C.2  PROMPTS

In our main experimental study, we use a strong system prompt to encourage models to use tools towards solving VISTOOLBENCH tasks. In addition, we design a weaker system prompt for ablation study. The strong and weak system prompts are provided below.

---

**System Prompt**

**System Prompt (Strong)**

You are a *proactive, tool-empowered* visual-reasoning assistant.
When user supplies an image and requests to solve a problem that requires visual content that are small, ambiguous, or not centered, you must:
1. Examine the image carefully and mentally list the visual clues most likely to locate the target object.
2. Proactively use the image-processing tools - such as crop, zoom, or enhance - to isolate and clarify the relevant region.
3. Save each transformed image. The updated image will be appended to the conversation for your reference.
4. Iterate as needed. Call the tools repeatedly until the visual evidence is clear enough to answer the user's request.
5. Double-check your observations. Confirm that the final transformed image supports an accurate, confident response before replying to the user.
6. Use other general-purpose tools if needed to answer the user's question.
7. Please use the tools wisely as you have limited tool calls.

**System Prompt (Weak)**

You are a helpful visual reasoning assistant with access to tools to help you answer the user's question.

---

## C.3 LLM-AS-JUDGE PROTOCOL

We employ an LLM-as-judge framework for large-scale evaluation. To improve reliability, each judgment considers one rubric at a time. All rubrics are atomic: each rubric targets a single, verifiable fact or behavior, thereby reducing ambiguity, limiting error propagation, and yielding clearer agreement signals.

The exact judge prompts are provided below. For each case, the judge receives: (i) the original question, (ii) the gold answer, (iii) a single rubric criterion, and (iv) the model's answer; it returns a verdict (met or not met) with a brief, evidence-grounded justification.

---

**LLM Judge Prompt**

You are an expert evaluator tasked with judging whether a model's answer meets a specific rubric criterion. You will be provided with:
- a question
- a golden (reference) answer
- a rubric criterion
- the model's answer

Your task is to decide if the model's answer **meets** or **does not meet** the given rubric criterion, referencing the golden answer only as needed.

### Inputs:
**Question:** question
**Golden Answer:** golden_answer
**Rubric Criterion:** rubric_criteria
**Model Answer:** model_answer

### Important Notes:
- The model's answer does not need to be correct to meet the criterion if correctness is not required.

---

*Example:* If the rubric is "The model should show its reasoning process to answer the question," the answer can be incorrect but still meet the rubric if model's reasoning process is present.
- For writing style or presentation rubrics, apply leniency.
*Example:* If the rubric asks for conciseness, answers that are slightly longer than the golden answer but still reasonably length should be considered as meeting the rubric.
- The model's answer may satisfy the rubric implicitly without explicitly mentioning the exact term. This should still be considered as meeting the criterion if model's answer is reasonable and makes sense.
*Example:* If the rubric is "The model should demonstrate understanding of photosynthesis," and the model states "Plants make their own food using sunlight," without explicitly mentioning the term "photosynthesis," it still meets the criterion.

### Output Format:
Return your judgment in the following JSON format:

```
{
    "explanation": "Brief explanation of your judgment",
    "judge_result": "Met" or "Not Met"
}
```

## D    MORE ON VISTOOLBENCH

### D.1    TASK DOMAIN AND SUB-DOMAIN TAXONOMY

In this section, we provide the full statistics of tasks across all subdomains in Table 14.

### D.2    MORE BENCHMARK EXAMPLES

In this section, we provide more demonstration examples of VISTOOLBENCH. Figure 8 and Figure 9 provide two more examples with model responses and grading process.

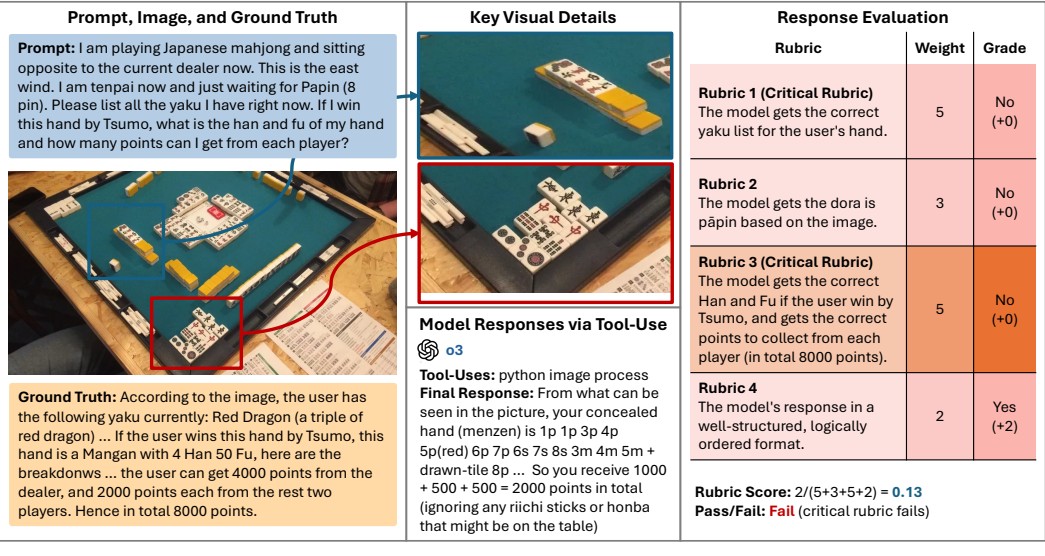

Figure 8: Demonstration example from VISTOOLBENCH (single-turn, generalist domain, region switch Q&A).

Below, we present additional benchmark examples covering both single-turn and multi-turn settings across all five task categories. Each example includes the task prompt, image input, golden answer, reference tool trajectory, and evaluation rubrics.

| Domain | Subdomain | Number of Tasks |
|--------|-----------|:---------------:|
| **STEM** | Engineering | 44 |
| | Physics | 49 |
| | Maths | 49 |
| | Biology | 52 |
| | Chemistry | 44 |
| **Finance** | Market Analysis | 77 |
| | Personal Finance | 90 |
| | Corporate Finance | 50 |
| | Currency Trends | 14 |
| | Taxation Analysis | 12 |
| **Medical** | Clinical Specialties | 42 |
| | Laboratory Medicine | 51 |
| | Medical Imaging | 113 |
| | Pharmaceutical Science | 25 |
| | Public Health | 7 |
| **Sports** | Team Sports | 147 |
| | Individual Sports | 50 |
| | Racket Sports | 15 |
| | General Sports | 28 |
| **Generalist** | Daily Life | 31 |
| | Transportation | 38 |
| | Game & Entertainment | 86 |
| | Problem Solving | 86 |
| | Support & Services | 4 |
| **Total** | — | **1,245** |

Table 14: Task Distribution Across Domains and Subdomains in VISTOOLBENCH

---

Single-Turn, Generalist, Hybrid-Reasoning

**Prompt:** Check out the menu from this new coffee shop I'm meeting a date at. I want to have a hot coffee ready for him when he gets here. Give me a suggestion for someone who doesn't like steamed milk or sweet-tasting drinks. I don't want it to be the cheapest type of coffee on the menu.

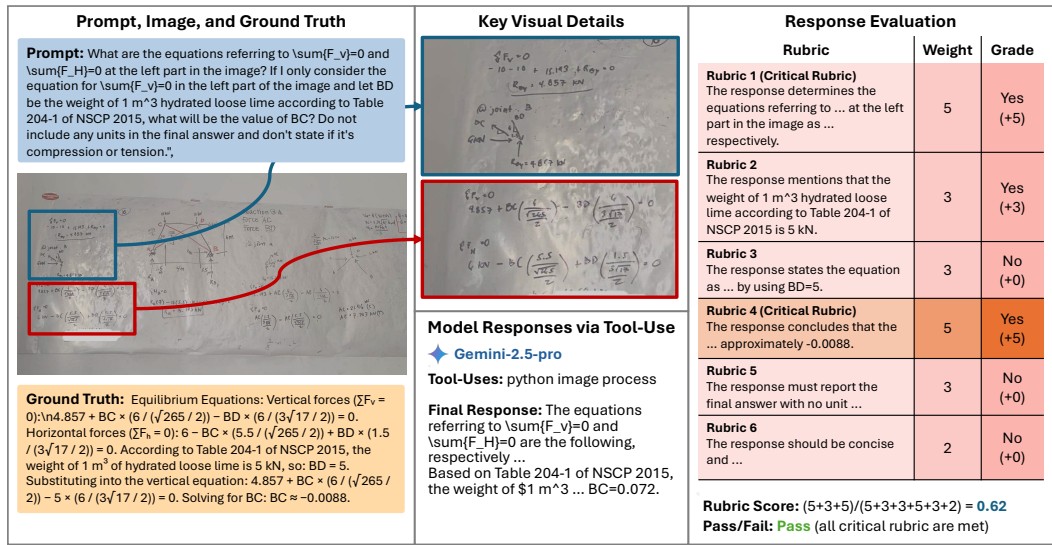

Figure 9: Demonstration example from VISTOOLBENCH (single-turn, STEM domain, region switch Q&A).

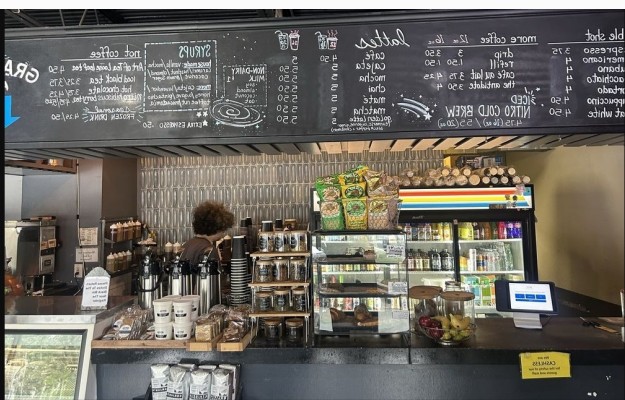

**Golden Answer:** That's a great way to start your date! Looking at the menu, an Americano would meet your requirements. It doesn't contain steamed milk, and it's not sweet. At $4, it's not the cheapest item on the menu, either. Let me know if you need any other suggestions!

**Reference Tool Trajectory:**

1. `python_image_processing`: Load image to enhance it by flipping horizontally.

2. `python_image_processing`: Crop the image for items that qualify as coffee and are not the cheapest on the menu.

3. `web_search`: Perform a web search to identify which types of coffee contain steamed milk and are sweet-tasting.

**Rubrics:**

| Description | Weight | Type | Category | Critical |
|---|---|---|---|---|
| Suggests Americano coffee as the beverage. | 5 | Objective | Reasoning, Instruction following, Truthfulness, Visual Understanding | Yes |
| States that the Americano is not sweet and does not contain milk. | 5 | Objective | Instruction following, Truthfulness | Yes |
| States that the Americano is not the cheapest coffee on the menu | 4 | Objective | Visual Understanding | No |

### Single-Turn, STEM, Region Switch Q&A

**Prompt:** For the top-left flowchart, what operation updates 'S' inside the loop, and what condition terminates it? Also, for the top-right flowchart, what are the two possible calculations for 'y', and what condition decides which one is performed?

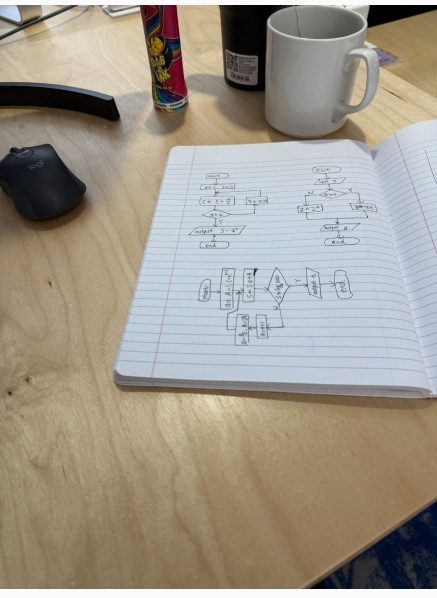

**Golden Answer:** The flowchart on the top left defines a loop. The process inside this loop updates the variable 'S' with the specific mathematical operation $S \leftarrow S + x/2$. The loop is controlled by the diamond shape, which sets the exact termination condition to $x \geq 4$. The flowchart on the top-right describes a conditional process based on the input 'x.'. The diamond shape, $x > 0$, is the condition that determines which path is taken. If the condition is true, the calculation performed is $y \leftarrow x + 1$; if it is false, the alternate calculation performed is $y \leftarrow 2^x$.

**Reference Tool Trajectory:**

1. `python_image_processing`: Crop the top-left flowchart and read the text inside the loop to identify the iterative operation on the 'S' variable ($S \leftarrow S + x/2$) and the text in the diamond shape to identify the loop's termination condition ($x \geq 4$).

2. `python_image_processing`: Crop the the top-right flowchart and read the text in the diamond to find the deciding condition ($x > 0$) and trace its "Yes" and "No" paths to find the two possible calculations for the output 'y' ($y \leftarrow x + 1$ and $y \leftarrow 2^x$).

**Rubrics:**

| Description | Weight | Type | Category | Critical |
|---|---|---|---|---|
| States that the operation updating $S$ inside the loop is $S \leftarrow S + x/2$. | 5 | Objective | Reasoning, Instruction following, Truthfulness, Visual Understanding | Yes |
| States that the loop terminates when $x > 4$. | 5 | Objective | Reasoning, Instruction following, Truthfulness, Visual Understanding | Yes |
| The response states both possible calculations for the output $y$ as $y \leftarrow x + 1$ and $y \leftarrow 2^x$. | 4 | Objective | Reasoning, Instruction following, Truthfulness, Visual Understanding | Yes |
| The response states that the deciding condition from the top-right flowchart is $x > 0$. | 3 | Objective | Reasoning, Instruction following, Truthfulness, Visual Understanding | No |

## Single-Turn, Generalist, Region Switch Q&A

**Prompt:** I need to get to Crossroads Mall entrance from the Central Transfer station by 8:30. When do I need to be ready to be picked up? Also, which road will the route follow for most of the trip?

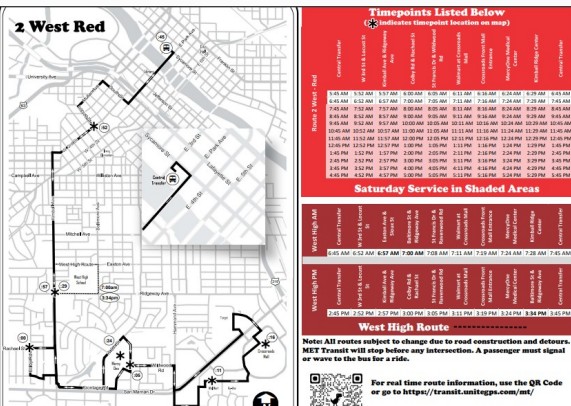

**Golden Answer:**

- You will need to be picked up at the Central Transfer station at 7:45 AM, since this bus arrive at Crossroads Mall by 8:16.
- The route mainly follows Kimball Avenue from the image.

**Reference Tool Trajectory:**

1. `python_image_processing`: Zoom-in the bus schedule area, find a bus leaving the Central Transfer station to arrive at Crossroads Mall by 8:30. Also use this tool to identify the main road the route follows..

2. `python_image_processing`: Crop the map area to find the main road name for the bus route.

**Rubrics:**

| Description | Weight | Type | Category | Critical |
|---|---|---|---|---|
| The model identifies the main road the route follows as Kimball Avenue. | 5 | Objective | Reasoning, Instruction following, Truthfulness, Visual Understanding | Yes |
| The response states that the user should be picked up at 7:45 AM to catch the bus arriving at Crossroads Mall entrance by 8:30 AM. | 5 | Objective | Reasoning, Instruction following, Truthfulness, Visual Understanding | Yes |

## Multi-Turn, Medical, Temporal Sequential Reasoning

### Turn 1

**Prompt:** The medical team is treating a young patient admitted to the emergency room with a diagnosis of pneumothorax after a MVA. During surgery, a chest drain is placed under fluoroscopic guidance. Based on this image, describe the correct trajectory and placement of the instruments. Also, describe the criteria for confirming the safety of the procedure, as well as the possible complications of improper drain placement.

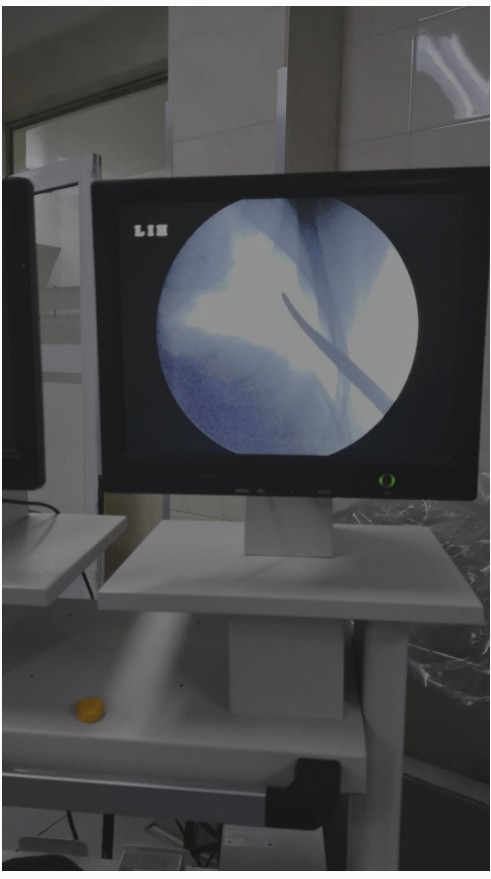

**Golden Answer:** The image shows the introduction of scissors into the thoracic cavity of a patient with pneumothorax. The instrument's path is carefully directed toward the pleural area, taking great care to avoid direct contact with central structures such as the heart. The

orientation is appropriate, as it follows a lateral and slightly superior path, allowing safe access to the pleural space. Correct placement and safety criteria:

- The instrument is directed toward the pleural cavity and not the mediastinum.
- It is kept away from the cardiac silhouette and large vessels.
- The progressive opening of the tract facilitates subsequent placement of the drainage tube.

Possible complications of incorrect placement:

1. Cardiac or large vessel injury (cardiac tamponade, massive hemothorax).
2. Lung injury with subcutaneous emphysema or hemothorax.
3. Ineffective extrapleural placement.

In conclusion, the image shows a technically correct procedure in the scissors dissection phase, maintaining safety by avoiding vital structures before definitive placement of the chest drain.

**Reference Tool Trajectory:**

1. `python_image_processing`: Apply a brightness and contrast adjustment to clearly display the image.
2. `python_image_processing`: Apply a crop operation to extract only the area of the monitor containing the image.
3. `web_image_processing`: Search for information about the pneumothorax intervention procedure and its potential complications from improper drain placement.

**Rubrics:**

| Description | Weight | Type | Category | Critical |
| --- | --- | --- | --- | --- |
| Identify that the chest drain orientation follows a lateral and slightly superior path to safely access the pleural space. | 5 | Objective | Instruction following, Truthfulness, Visual Understanding | Yes |
| Identify safety criteria: instrument trajectory into the pleural cavity while avoiding the cardiac silhouette and large vessels. | 5 | Objective | Reasoning, Instruction following, Truthfulness | Yes |
| Identify possible complications of improper drain placement, including cardiac or vessel injury, lung injury, and ineffective extrapleural placement. | 4 | Objective | Reasoning, Instruction following, Truthfulness | Yes |

=======================================================================

**Turn 2**

**Prompt:** The patient also has a fracture due to the MVA, which was treated along with the pneumothorax. The patient returns for routine follow-up after surgical fixation of a fracture. The radiograph shows the surgical site with the internal fixation hardware in place. Identify the position of the fracture and the hardware holding it. Also, identify the progress of bone healing and if there are any potential complications, such as infection, loosening, hardware migration, or malunion.

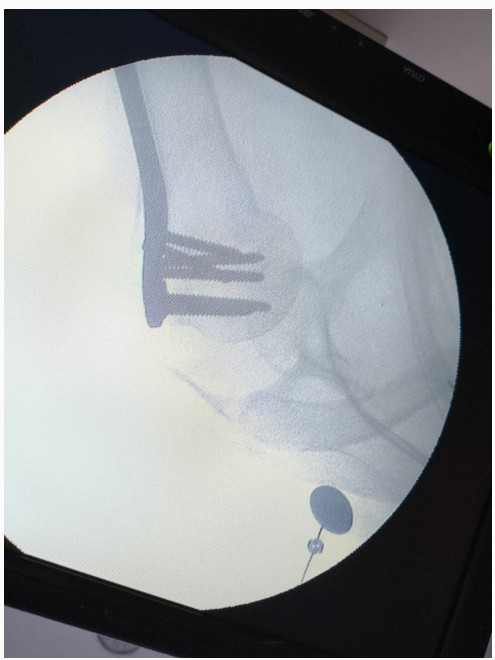

**Golden Answer:** The follow-up radiograph of the proximal humerus shows internal fixation with a plate and multiple screws appropriately placed in the fracture. The hardware is intact, with no signs of loosening, migration, or fracture. The fracture is still slightly appreciable but shows early union with callus formation, consistent with a normal healing process at this stage. There is no evidence of infection, osteolysis, or malunion. Overall, the postoperative outcome of the proximal humerus fracture is favourable, with correct alignment and satisfactory progression.

**Reference Tool Trajectory:**

1. `python_image_processing`: Apply a flip operation to flip it vertically and extract it in the correct orientation.

2. `python_image_processing`: Apply a rotation operation using the estimated coordinates to extract only the laptop screen area with the imaging findings.

**Rubrics:**

| Description | Weight | Type | Category | Critical |
|---|---|---|---|---|
| Identifies absence of infection and other complications. | 5 | Objective | reasoning, truthfulness, visual understanding | Yes |
| Identify that the proximal humerus fracture is treated with internal fixation hardware (a plate and multiple screws) appropriately placed. | 5 | Objective | visual understanding | Yes |
| Identifies that callus is forming at the fracture site. | 4 | Objective | Reasoning, Visual Understanding | Yes |
| Identify that this is a shoulder radiograph. | 2 | Objective | Visual Understanding | No |
| Identifies that the radiograph image quality is sufficient for evaluation | 1 | Objective | Visual Understanding | No |

Multi-Turn, Generalist, Follow-up Test

**Turn 1**

**Prompt:** Shown is a map of multiple train lines throughout a city, with each station/stop marked by a white dot (with intersecting lines indicating where one can transfer lines). Suppose I start at the northmost station of Line Seven; then, without using external sources, what is the least number of required stops to reach the southmost station of Line One? Give all the reasoning necessary to solve the question.

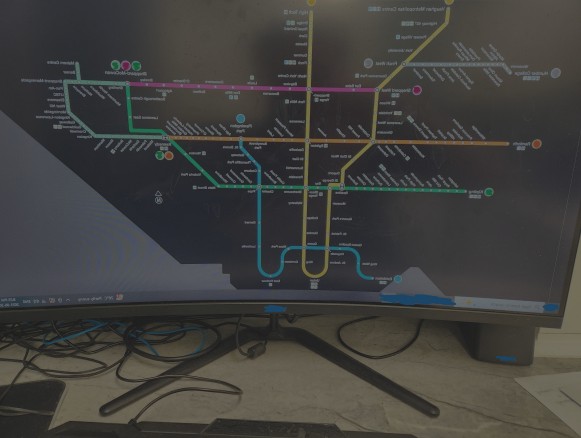

**Golden Answer:** I am sorry, but it appears the numbers corresponding to each train line are not listed in the image given. To determine both the northmost station of Line Seven, as well as the southmost station of Line One, the line number corresponding to each color (or at least the colors associated with Lines One and Seven) must be given. Can you please provide this information?

**Reference Tool Trajectory:**

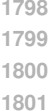
1. `python_image_processing`: Flip the image horizontally, and increase brightness and contrast as needed.

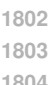
2. `python_image_processing`: Locate the train line map on the screen of the computer in the upper/middle portion of the image. Identify that the line numbers are not listed/are missing from the picture.

**Rubrics:**

| Description | Weight | Type | Category | Critical |
|---|---|---|---|---|
| The model identifies that the line numbers corresponding to each color are not provided in the image and are necessary to solve the problem. | 5 | Objective | Visual Understanding | Yes |
| The model asks a follow-up clarification question requesting the mapping between line numbers and their colors for Lines One and Seven. | 4 | Objective | Instruction following | Yes |

==================================================================

**Turn 2**

**Prompt:** That's my bad, I forgot to provide you with the line numbers on the map. Can you now please solve the same question? Shown is a map of multiple train lines throughout a city, with each station/stop marked by a white dot (with intersecting lines indicating where one can transfer lines). Suppose I start at the northmost station of Line Seven; then, without using external sources, what is

the least number of required stops to reach the southmost station of Line One? Give all the reasoning necessary to solve the question.

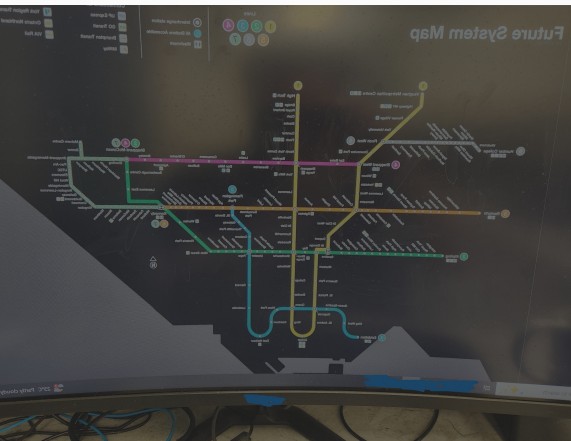

**Golden Answer:** Using the compass at bottom right (north is up), identify Line 1 (yellow) and Line 7 (light green). The northernmost Line 7 station is Malvern Centre and the southernmost Line 1 station is Union. The minimum-stop route is: Malvern Centre → Neilson → west via Washburn → Sheppard McCowan → Kennedy → south to Warden → Pape → Garrard → Queen → Union, avoiding longer branches via Brenyon or Ionview. Counting stops on this path (excluding the starting station) gives 25.

**Reference Tool Trajectory:**

1. `python_image_processing`: Flip the image horizontally, and increase brightness and contrast as needed.

2. `python_image_processing`: Locate the train line map on the computer screen in the middle/upper portion of the image. Further, zoom into the upper portion of the screen to identify the train line color pairing as: light green (Line Seven) and yellow (Line One). Next, locate the compass symbol near the bottom right portion of the screen, indicating that the top of the image is the northmost portion. Finally, identify all possible station stops (marked as white dots) between the northmost station of Line Seven (Malvern Centre) and the southmost station of Line One (Union).

**Rubrics:**

| Description | Weight | Type | Category | Critical |
|---|---|---|---|---|
| Model correctly identifies the northmost station of Line Seven as Malvern Centre. | 5 | Objective | Reasoning, Truthfulness, Visual Understanding | Yes |
| Model correctly identifies the southmost station of Line One as Union Station. | 5 | Objective | Reasoning, Instruction following, Truthfulness, Visual Understanding | Yes |
| The model states that the minimum number of stops required when travelling from Malvern Centre to Union Station is 25. | 4 | Objective | Reasoning, Instruction following, Truthfulness, Visual Understanding | Yes |
| The model outlines a path listing stations from Malvern Centre to Union Station that follows the specified route, allowing equivalent station names. | 3 | Objective | Reasoning, Instruction following, Truthfulness, Visual Understanding | No |

Multi-Turn, Sports, Progressive Visual Reasoning

**Turn 1**

**Prompt:** A badminton tournament during the 2016 Rio Olympics between Lin Dan (in red) and Lee Chong Wei (in blue) is shown, which is a best-of-three-game series. The middle of a round is currently in play. Who is currently hitting/about to hit the shuttlecock, and how do you know? Further, if the shuttlecock is missed by the opponent after being hit by the player from the answer in the first question, what numbered point on the ground labelled in red will be the farthest point away from the receiver (player who missed), such that the shuttlecock is not considered out?

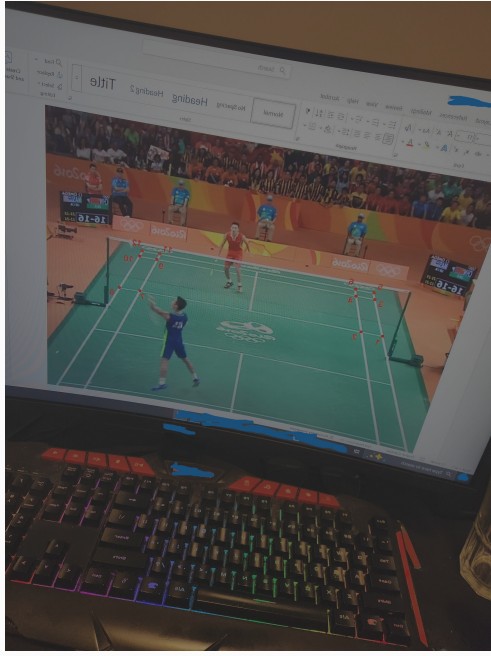

**Golden Answer: Who is hitting/about to hit the shuttlecock?** Since the player in blue is jumping, this is a clear indication that he is currently hitting/about to hit the shuttlecock. Further, if one looks more carefully, a white shuttlecock can be identified near the top right of the blue player's racket. Thus, the blue player (Lee Chong Wei) is currently hitting/about to hit the shuttlecock.

**What point labelled in red is farthest away from the receiving player if he misses the shot made by Lee Chong Wei, not considered out?** Since the game shown is between two players (singles), any shots where the shuttlecock lands in the farthest left or farthest right rectangles are considered out (everything else is considered in, including the backmost rectangle). Thus, this eliminates the points numbered 1, 3, 5, 8, 10, and 12 as possible answers. Of the remaining points, the point numbered 6 is furthest away from Lin Dan (red player). Thus, the point labelled 6 is the farthest point away from the receiving player (Lin Dan) considered in, assuming he misses the shot made by Lee Chong Wei.

**Reference Tool Trajectory:**

1. `python_image_processing`: Flip the image horizontally.
2. `python_image_processing`: Adjust the brightness and contrast of the image.

**Rubrics:**

| Description | Weight | Type | Category | Critical |
|---|---|---|---|---|
| The model correctly identifies Lee Chong Wei (in blue) as the player currently hitting or about to hit the shuttlecock. | 5 | Objective | Instruction following, Truthfulness, Visual Understanding, Reasoning | Yes |
| The model justifies its identification by referencing either the player's posture/position or the visible shuttlecock location near the racket. | 4 | Objective | Visual Understanding, Reasoning | Yes |
| The model must reason that the game shown is a singles match between two players. | 3 | Objective | Reasoning | No |
| The model must state that the farthest most left and farthest most right rectangles are considered out if the shuttlecock lands there. | 3 | Objective | Reasoning | No |
| Identify the 12 points labelled in red (1 through 12) on Lin Dan's side of the court. | 1 | Objective | instruction following, Visual Understanding | No |
| State that the shuttlecock is out if it lands in any of the labelled points 1, 3, 5, 8, 10, or 12. | 2 | Objective | instruction following, reasoning, truthfulness | No |
| State that point number 6 is the farthest allowed point from Lin Dan still considered in.. | 5 | Objective | Instruction following, Truthfulness, Visual Understanding, Reasoning | Yes |
| Response includes explicit reasoning rather than only providing a ground truth answer. | 3 | Objective | Reasoning, Presentation | No |

================================================================

**Turn 2**

**Prompt:** Suppose now, instead of hitting the shuttlecock to the point labelled 6 on the ground, Lee Chong Wei instead hits the shuttlecock to the point labelled 8 on the ground. Assuming Lin Dan does not receive (hit) the shuttlecock and lets it hit the ground, what will the scoreboard read after this point? State the player's name associated with the number of points as well.

**Golden Answer: Current score according to the scoreboard:** From the scoreboard, it reads 16-16, where the left number is the score associated with the team from China (from the flag shown), whereas the right number is the score associated with the team from Malaysia (from the flag shown).
**New score, assuming the shuttlecock lands at the point labelled 8:** Since the game is a singles match, the outer left and outer right-most rectangles are considered out. Since the point labelled 8 is in the outer right-most rectangle, this shot made by Lee Chong Wei would be considered out. Since Lin Dan did not receive the shot as stated by the prompt, Lin Dan will win this round, increasing the score to 17-16. (Here, it is important to note that the left number on the scoreboard has increased, and not the right). Further, Lin Dan will have 17 points, while Lee Chong Wei will still have 16 points.

**Reference Tool Trajectory:**

1. `python_image_processing`: Identify the current match score as 16-16 (middle top right of the image), where the left number is associated with the score of the team from China, whereas the right number is associated with the score of the team from Malaysia, according to the flags.

2. `web_search`: Look up and determine that Lin Dan represents the team from China, whereas Lee Chong Wei represents the team from Malaysia.

**Rubrics:**

| Description | Weight | Type | Category | Critical |
|---|---|---|---|---|
| Identify the current score as 16-16. | 3 | Objective | Truthfulness, Visual Understanding | Yes |
| Associate the left number with the team from China and the right number with the team from Malaysia based on the flags. | 1 | Objective | Truthfulness, Visual Understanding | No |
| Model correctly assigns Lin Dan to the left scoreboard number (team from China) and Lee Chong Wei to the right scoreboard number (team from Malaysia). | 2 | Objective | Reasoning, Visual Understanding | No |
| Model correctly reasons that because the shuttlecock landing point labelled 8 is out in singles matches and Lin Dan does not hit it back, Lin Dan receives the point | 5 | Objective | Reasoning, Visual Understanding | Yes |
| The model states the new score after Lin Dan receives the point as: 17-16 (17 for Lin Dan, and 16 for Lee Chong Wei). | 5 | Objective | Reasoning | Yes |
| The model states the new score after Lin Dan receives the point as: 17-16 (17 for Lin Dan, and 16 for Lee Chong Wei). | 5 | Objective | Truthfulness, Instruction Following | Yes |
| The reasoning explains why point 8 is considered out. | 4 | Objective | Reasoning, Presentation | Yes |
| The reasoning explains why the left number on the scoreboard is associated with Lin Dan and not Lee Chong Wei. | 1 | Objective | Reasoning, Presentation | No |

========================================================================

**Turn 3**

**Prompt:** Now, since the tournament is a best-of-three-game series, if Lee Chong Wei in fact does hit the shuttlecock out at the point labelled 8 on the ground as discussed, how many additional points will he need to win the match (assuming the score does not tie at 20-20)? If Lee Chong Wei wins this match/game, is he guaranteed to win the series? If so, what were the scores of the previous two matches played against each other?

**Golden Answer: How many points does Lee Chong Wei need to win the match after shooting the shuttlecock out?** The new score after Lee Chong Wei hits the shuttlecock out is 17-16 (17 for Lin Dan, 16 for Lee Chong Wei). Since a standard badminton game is up to 21 points, and assuming the score does not tie to 20-20 (in which case you can only win if you get two points back-to-back), Lee Chong Wei thus needs 21-16=5 points more to win.

**Scores of the previous two matches:** One can deduce that the previous two games have already been finished from the scoreboard, where it states that Lin Dan won one game with a final score of 21-15, whereas Lee Chong Wei won the other game with a score of 11-21. Thus, the game being played currently is the final game to determine the winner of the three-game series. Hence, if Lee Chong Wei wins this match, he is guaranteed to win the series.

**Reference Tool Trajectory:**

1. `python_image_processing`: Identify the scores of the previous two matches/games in the middle top right of the image as: 21-15 (for Lin Dan), and 11-21 (for Lee Chong Wei).

2. `web_search`: Lookup the amount of points needed to win a standard badminton game as 21 (assuming a 20-20 point tie does not occur).

**Rubrics:**

| Description | Weight | Type | Category | Critical |
|---|---|---|---|---|
| Identifies the final scores of the previous two matches as 21-15 (Lin Dan) and 11-21 (Lee Chong Wei). | 3 | Objective | Instruction following, Truthfulness, Visual Understanding | No |
| The model reasons that Lee Chong Wei must reach a total point count of 21 to win the match (assuming a 20-20 tie does not occur). | 4 | Objective | Reasoning | Yes |
| The model correctly calculates that Lee Chong Wei needs 5 additional points to win the current match. | 5 | Objective | Reasoning, Instruction following, Truthfulness | Yes |
| The model correctly states that if Lee Chong Wei wins this match, he is guaranteed to win the best-of-three series and provides the scores of the previous two matches. | 5 | Objective | Reasoning, Presentation | Yes |
| The response provides intermediate reasoning that addresses each question posed by the prompt: (1) how many additional points Lee Chong Wei needs to win the match, and (2) whether he is guaranteed to win the series and what the scores of the previous two matches were. | 2 | Objective | Reasoning, Presentation | No |

### D.3 MORE IMAGES FROM VISTOOLBENCH

In this section, we present additional benchmark images across five categories: STEM (Figure 10), Medical (Figure 11), Finance (Figure 12), Sports (Figure 13), and Generalist (Figure 14). These examples highlight the diverse visual elements contained in VISTOOLBENCH and demonstrate that its tasks are genuinely visually challenging, often requiring the use of vision tools to be solved effectively. In addition, we provide the images type categories within VISTOOLBENCH in Table 15.

## E MORE ON TOOLS ANALYSIS

In this section, we provide more details on the tools supported by VISTOOLBENCH and more examples on Tool-use of the evaluated models.

### E.1 TOOL DESCRIPTION

Table 16 provides tool description supported by VISTOOLBENCH.

### E.2 ABLATION ON ATOMIC VISION TOOLS AND GENERAL VISION TOOL

Our evaluation toolkit includes a general-purpose vision tool, `python_image_processing`, which executes arbitrary image-processing operations based on model-generated code. To empirically study the effect of decoupling coding ability from visual reasoning, we perform an ad-

| Image Type | Count | Description |
|---|---|---|
| Real-World Photos | 1108 | Natural photographic images of real objects, scenes, people, or environments. |
| Screenshots | 945 | Captured computer or mobile screens, including UIs, apps, code editors, and web pages. |
| Charts and Plots | 492 | Line charts, bar charts, scatter plots, scientific graphs, and other quantitative figures. |
| Medical Imagery | 348 | Clinical or biomedical images such as X-rays, MRIs, microscopy, or diagnostic visuals. |
| Tables (Rendered Images) | 328 | Images of tabular data from reports, spreadsheets, or financial summaries. |
| Diagrams | 283 | Flowcharts, block diagrams, circuit diagrams, system schematics, and conceptual visuals. |
| Documents | 239 | Scanned or photographed text documents, forms, notes, receipts, or printed pages. |
| Scientific Content Images | 232 | Equations, geometric figures, annotations, physics setups, or mathematical visuals. |
| Maps | 57 | Geographical, navigation, or floor-plan style maps. |
| Synthetic / Rendered Images | 72 | 2D/3D computer-generated graphics, CAD images, or simulation renderings. |
| Others | 12 | Images that do not fit into the defined categories or contain mixed content. |

Table 15: Distribution of Image Types in VISTOOLBENCH with Descriptions

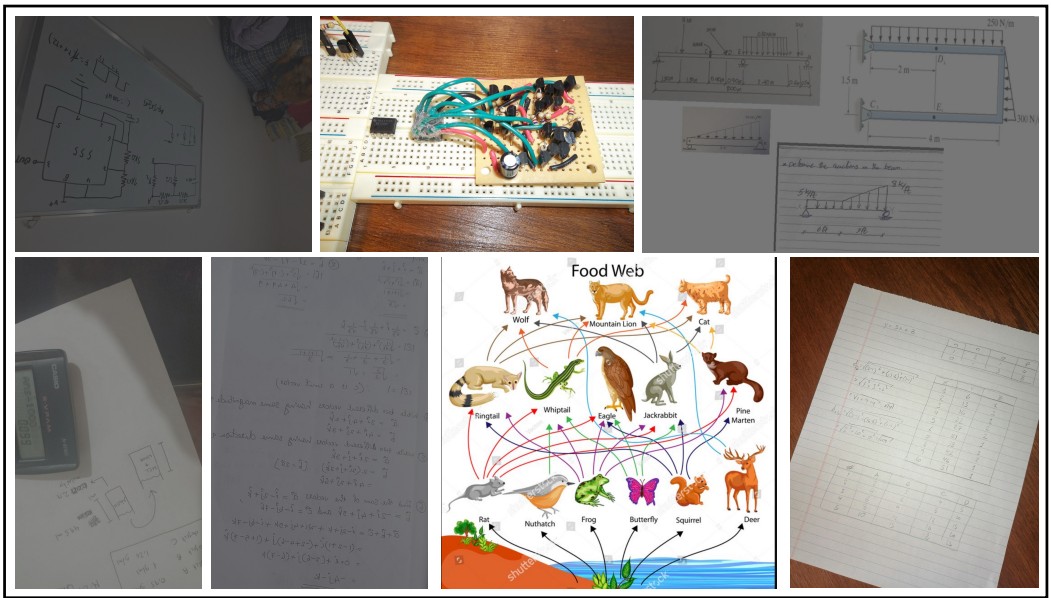

Figure 10: Selected STEM images from VISTOOLBENCH.

ditional ablation experiment in which `python_image_processing` is replaced by an atomic toolset composed of common, predefined image-processing operations, including:

- `crop_image`

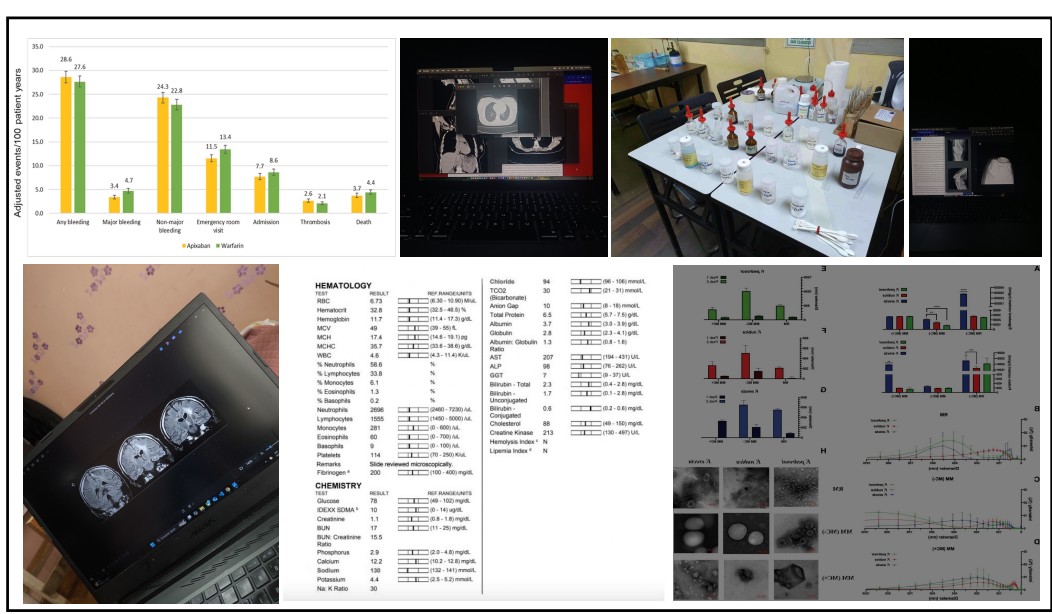

Figure 11: Selected medical images from VISTOOLBENCH.

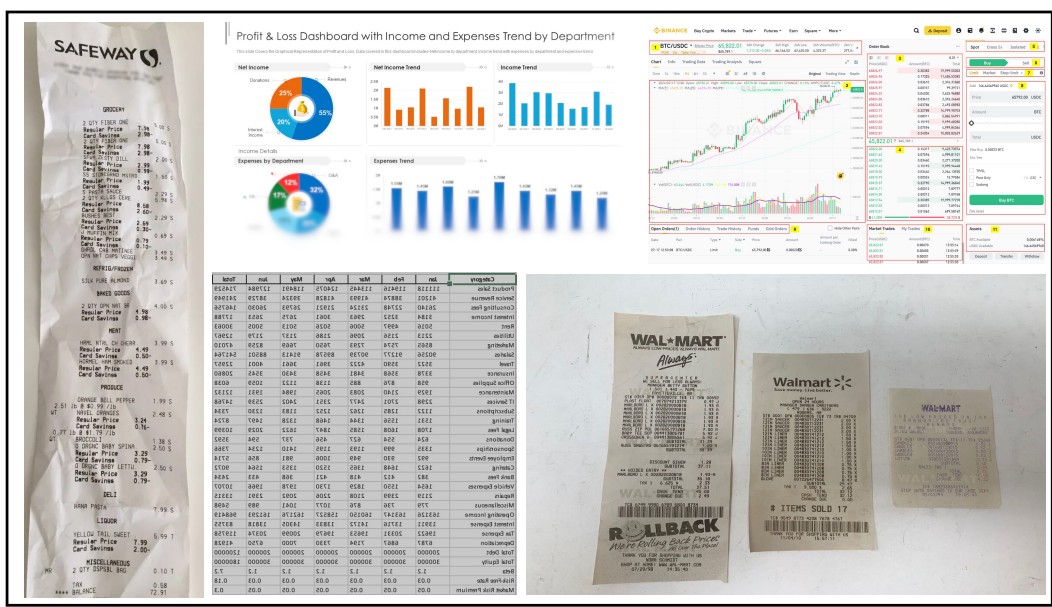

Figure 12: Selected finance images from VISTOOLBENCH.

- `rotate_image`
- `adjust_brightness`
- `adjust_contrast`
- `adjust_saturation`
- `adjust_sharpness`
- `resize_image`
- `flip_image`
- `convert_to_grayscale`

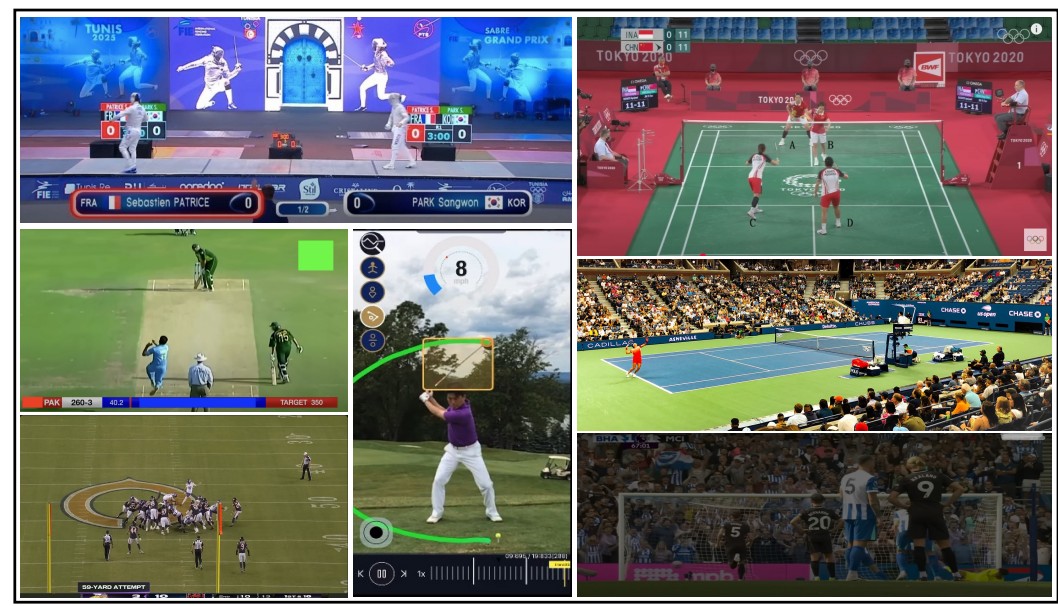

Figure 13: Selected sports images from VISTOOLBENCH.

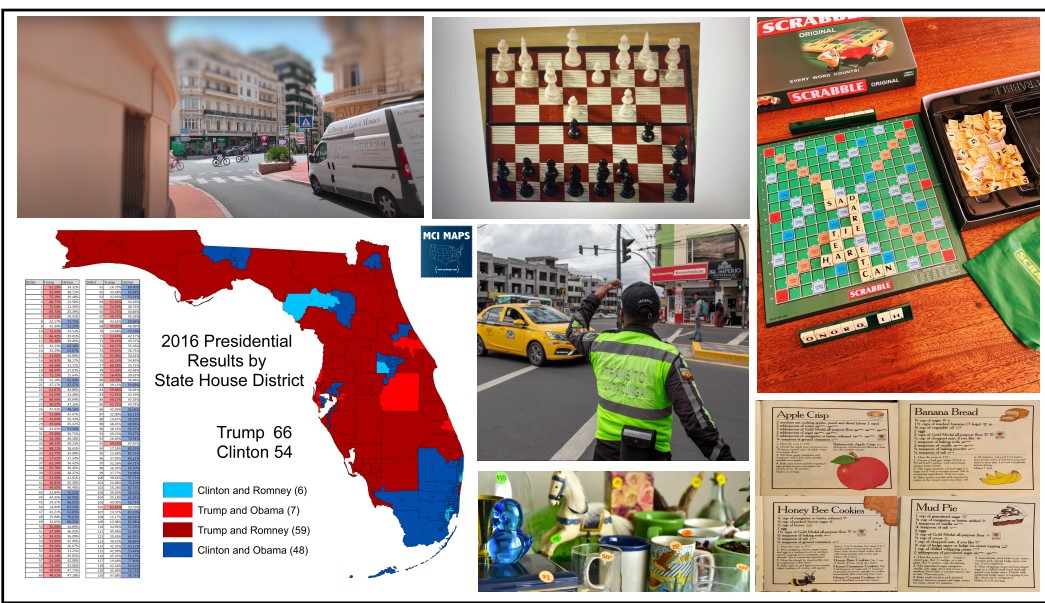

Figure 14: Selected generalist images from VISTOOLBENCH.

- `apply_filter`

We evaluated GPT-5 and GPT-4.1 on single turn tasks under two settings. The results are shown in Table 17. We make the following key observations:

1. For GPT-4.1, the atomic tool setup provides a small improvement. However, the gain is marginal, suggesting that the model's failures stem mainly from visual reasoning rather than from coding issues.

2. On the other hand, GPT-5 shows a performance decline under atomic tools, which we hypothesize arises because atomic tools restrict the model's expressive image manipulation

| Tool | Functionality |
|------|---------------|
| `python_image_processing` | Vision-specific manipulation, including image cropping, editing, rotation, brightness/contrast adjustment, and enhancement. Enables iterative refinement of visual inputs. |
| `python_interpreter` | General code execution. |
| `web_search` | Open-domain information retrieval from the web. |
| `browser-get-page-text` | Extraction of textual content from online sources. |
| `historical_weather` | Weather records for temporal and geographic look-ups. |
| `calculator` | Arithmetic operations for quick computations. |

Table 16: Supported tools supported in VISTOOLBENCH.

| Model | Setup | Overall | STEM | Medical | Finance | Sports | Generalist |
|-------|-------|---------|------|---------|---------|--------|------------|
| **GPT-5** | General vision tool | 23.88 | 29.31 | 23.14 | 24.39 | 18.49 | 24.19 |
| | Atomic vision tools | 18.15 | 20.69 | 14.05 | 17.07 | 18.49 | 20.95 |
| **GPT-4.1** | General vision tool | 5.97 | 6.03 | 10.74 | 1.63 | 8.40 | 3.23 |
| | Atomic vision tools | 6.30 | 6.90 | 11.57 | 1.63 | 7.56 | 4.03 |

Table 17: APR Results Across Models and Domains

capabilities. GPT-5 frequently composes multi-step and creative operations in the general setting; atomic tools constrain this flexibility and can reduce its performance.

### E.3 IMAGE MANIPULATION OPERATIONS

In this section, we present the representative o3 generated python code snippets, where the model invokes the `python_image_processing` tool to perform diverse image manipulations in support of problem solving. Lines that execute the manipulations are highlighted in yellow.

---

**Image Operations: Crop and Resize**

**Task ID: 68658a711603983919432615**

- line 9-15: image cropping
- line 17-18: image resize

```python
from PIL import Image
import os
import numpy as np
import cv2

img_path = '/mnt/efs/.../vision_tool_use/Kb2JElJ87npHXeG.jpg'
img = Image.open(img_path)

# Crop bottom middle area where Dixit might be
w, h = img.size
left = int(w * 0.35)
top = int(h * 0.45)
right = int(w * 0.65)
bottom = int(h * 0.8)
crop = img.crop((left, top, right, bottom))

# Enlarge cropped region
crop_large = crop.resize((crop.size[0]*2, crop.size[1]*2))
```

---

```
20  # Save
21  os.makedirs('eval_results_single_turn/...', exist_ok=True)
22  crop_large.save('eval_results_single_turn/...', 'PNG')
```

Original Image → Cropping → Transformed Image

## Image Operations: Rotation, Brightness and Contrast Adjustment

**Task ID: 68658a711603983919432613**

- line 5-6: image rotation
- line 7-9: brightness adjustment
- line 10-12: contrast adjustment

```python
1   from PIL import Image, ImageEnhance
2   import os, numpy as np
3   img_path = '/mnt/.../vision_tool_use/PtZEjeYmkRmW-Lf.jpg'
4   img = Image.open(img_path)
5   # rotate 180 degrees
6   img_rot = img.rotate(180, expand=True)
7   # enhance brightness
8   enhancer = ImageEnhance.Brightness(img_rot)
9   img_enh = enhancer.enhance(1.8)
10  # increase contrast
11  enhancer_c = ImageEnhance.Contrast(img_enh)
12  img_enh2 = enhancer_c.enhance(1.5)
13  # save
14  save_dir = 'eval_results_single_turn/...'
15  os.makedirs(save_dir, exist_ok=True)
16  save_path = os.path.join(save_dir, 'transformed_image_0.png')
17  img_enh2.save(save_path, 'PNG')
```

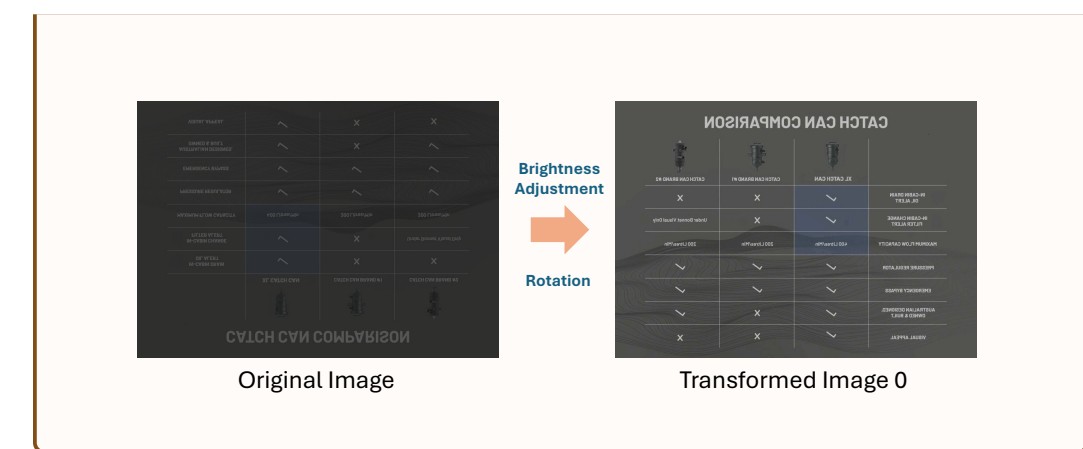

Original Image      Transformed Image 0

## Image Operations: Flip

**Task ID: 68658a711603983919432613**

- line 4-5: image flip

```
1  from PIL import Image
2  import os, numpy as np
3  img = Image.open('/.../transformed_image_0.png')
4  # flip horizontally to correct mirroring
5  img_flipped = img.transpose(Image.FLIP\_LEFT\_RIGHT)
6  # maybe crop to table area? Keep full for readability
7  save_dir='eval_results_single_turn/...'
8  img_flipped.save(save_dir,'PNG')
```

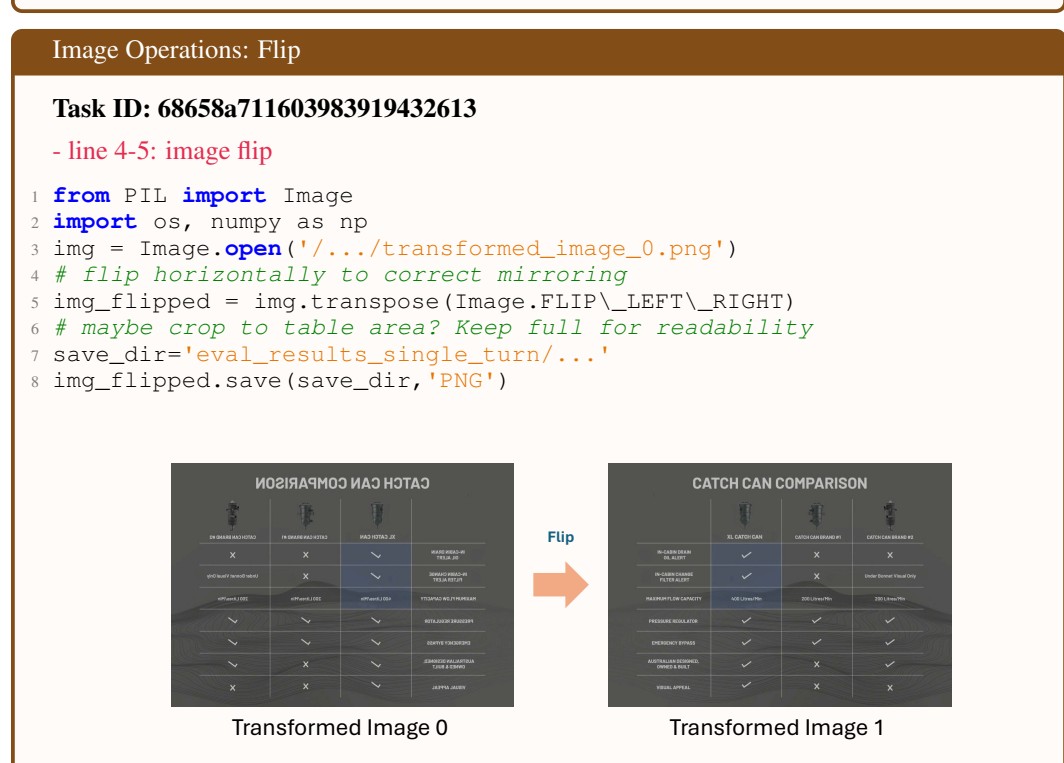

Transformed Image 0      Transformed Image 1

## Image Operations: Editing, Others

**Task ID: 68757ac94c044bac23c9bd8b**

- line 23-26: image editing

```
1  from PIL import Image
2  import numpy as np
3  import cv2
4  import os
5  # Load image
6  img_path='/mnt/.../SZJPcfAUQgyEVGe.jpg'
7  img=cv2.imread(img_path)
8  # Convert to HSV
9  hsv=cv2.cvtColor(img, cv2.COLOR_BGR2HSV)
10 # Define green range (for tree foliage), adjust
11 lower=np.array([35,40,40])
12 upper=np.array([85,255,255])
```

```
13 mask=cv2.inRange(hsv, lower, upper)
14 # Morphological operations to remove noise
15 kernel=np.ones((5,5),np.uint8)
16 mask=cv2.morphologyEx(mask, cv2.MORPH_OPEN, kernel)
17 # Find contours
18 contours,_=cv2.findContours(mask, cv2.RETR_EXTERNAL,
19 cv2.CHAIN_APPROX_SIMPLE)
20 # Filter contours by area to avoid tiny bits
21 tree_contours=[cnt for cnt in contours if cv2.contourArea(cnt)>500]
22 print('Totalcontourslarge', len(tree_contours))
23 # Draw bounding boxes to visualize
24 for cnt in tree_contours:
25     x,y,w,h=cv2.boundingRect(cnt)
26     cv2.rectangle(img,(x,y),(x+w,y+h),(0,0,255),2)
27 # Save transformed
28 out_dir='/...'
29 os.makedirs(out_dir, exist_ok=True)
30 Image.fromarray(cv2.cvtColor(img, cv2.COLOR_BGR2RGB)).
31 save(out_dir+'/transformed_image_0.png','PNG')"
```

Original Image          Transformed Image 0

### E.4 VISION TOOL CALL FOR GPT-5

From the first column of Table 8 and Figure 6, we observe that although GPT-5 and GPT-5-think invoke the vision tool less frequently than o3, they perform more image manipulations overall. A closer inspection shows that GPT-5 and GPT-5-think often execute multiple image-processing operations within a single vision-tool call, thereby achieving higher tool-call efficiency than o3. We provide two demonstration examples below.

---

**GPT-5 Vision Tool Call Output**

**Task ID: 68658a711603983919432613**

-line 12-14, 21-26: saved three plus transformed images

```
1 from PIL import Image, ImageEnhance, ImageFilter
2 import os
3 base='eval_results_single_turn/.../transformed_image_0.png'
4 img=Image.open(base)
5 # Let's crop central table area
6 w,h=img.size
7 # Crop left column, middle, right ... for readability
8 # First overall crop removing margins
9 crop=img.crop((int(w*0.06), ..., int(h*0.92)))
10 # enhance
11 crop=ImageEnhance.Contrast(crop).enhance(1.4)
12 #Save
13 out_dir=os.path.dirname(base)
14 crop.save(out_dir+'/transformed_image_1.png','PNG')
15 # Crop each column region roughly
16 # Determine thirds
```

```
17 cw=crop.size[0]
18 ch=crop.size[1]
19 col_width=cw//3
20 cols=[]
21 for i in range(3):
22   c=crop.crop((i*col_width,0,(i+1)*col_width,ch))
23   c=ImageEnhance.Brightness(c).enhance(1.2)
24   c=c.filter(ImageFilter.UnsharpMask(radius=1.5,percent=180,
       threshold=2))
25   c.save(out_dir+f'/transformed_image_{2+i}.png','PNG')
26   print("saved 3+ images")
```

### GPT-5-think Vision Tool Call Output

**Task ID: 68658a711603983919432616**

-line 24, 27, 32, 35, 38, 41: saved six transformed images in one tool call

```
1  from PIL import Image, ImageEnhance, ImageFilter, ImageOps
2  import numpy as np
3  import os
4
5  # Load the source image
6  src_path = '/mnt/efs/.../ymHX_cj7Vg3lDk8.jpg'
7  im = Image.open(src_path)
8  # Step 1: Auto-orient and slightly crop borders
9  im = ImageOps.exif_transpose(im)
10 # Step 2: Enhance brightness and contrast
11 bright = ImageEnhance.Brightness(im).enhance(1.8)
12 contrast = ImageEnhance.Contrast(bright).enhance(2.0)
13 # Step 3: Convert to grayscale for clarity
14 gray = ImageOps.grayscale(contrast)
15 # Step 4: Increase sharpness
16 sharp = ImageEnhance.Sharpness(gray).enhance(2.5)
17 # Step 5: Apply slight denoise via median filter, then unsharp mask
18 median = sharp.filter(ImageFilter.MedianFilter(size=3))
19 unsharp = median.filter(ImageFilter.UnsharpMask(radius=2, percent
     =200, threshold=3))
20
21 # Save the enhanced full image
22 save_dir = 'eval_results_single_turn/...'
23 os.makedirs(save_dir, exist_ok=True)
24 unsharp.save(os.path.join(save_dir, 'transformed_image_0.png'), '
     PNG')
25 # Also create a higher-contrast inverted version which sometimes
     helps
26 inv = ImageOps.invert(unsharp)
27 inv.save(os.path.join(save_dir, 'transformed_image_1.png'), 'PNG')
28 # Crop likely totals area: lower half of the long receipt\nw, h =
     im.size
29 # Long receipt is on right; crop right area
30 right_x0 = int(w*0.35)
31 right = unsharp.crop((right_x0, 0, w, h))
32 right.save(os.path.join(save_dir, 'transformed_image_2.png'), 'PNG'
     )
33 # Crop bottom of right receipt where totals usually appear
34 bottom_right = unsharp.crop((right_x0, int(h*0.55), w, h))
35 bottom_right.save(os.path.join(save_dir, 'transformed_image_3.png')
     , 'PNG')
36 # Crop mid section (items + discounts)
37 mid_right = unsharp.crop((right_x0, int(h*0.30), w, int(h*0.60)))
```

```
38 mid_right.save(os.path.join(save_dir, 'transformed_image_4.png'), '
      PNG')
39 # Crop the smaller left slip (maybe PWD details)
40 left = unsharp.crop((0, int(h*0.1), int(w*0.38), int(h*0.7)))
41 left.save(os.path.join(save_dir, 'transformed_image_5.png'), 'PNG')
42 print('Saved enhanced and cropped images.')"
```

## E.5 TOOL APIS

In this section, we provide more details on the tool API description.

### Tool Description: python_image_processing

```
{
    "type": "function",
    "function": {
        "name": "python_image_processing",
        "description": (
            "Generate arbitrary Python code for image manipulation
            ↪  and save the transformed image as PNG.\n"
            f"-Read one source image (your choice) from the
            ↪  working-directory file list: {image_list}.\n"
            f"-Perform any image processing with PIL, NumPy, or
            ↪  OpenCV. You cannot use matplotlib to show the
            ↪  image.\n"
            f"-You **must save** the transformed image as PNG to
            ↪  {processed_image_save_path} using the filename
            ↪  pattern "
            "\"transformed_image_i.png\", where the counter **i
            ↪  starts at 0 and increments on each invocation** "
            "so files are never overwritten. Example:\n"
            f"    img.save(f\"{processed_image_save_path}/transform↵
            ↪  med_image_{{i}}.png\",
            ↪  \"PNG\")\n"
        ),
        "parameters": {
            "type": "object",
            "properties": {
                "code": {
                    "type": "string",
                    "description": "Python code to run.",
                    "minLength": 1,
                    "maxLength": 5000
                }
            },
            "required": ["code"]
        }
    }
}
```

### Tool Description: python_processing

```
{
    "type": "function",
    "function": {
        "name": "python_interpreter",
        "description": (
            "General-purpose Python interpreter. Run arbitrary
            ↪  Python code and capture stdout via print(). "
```

```
                "Any exceptions are returned in stderr.\n\n"
                "Pre-installed packages:\n"
                "  • numpy\n"
                "  • pandas\n"
                "  • requests\n"
                "  • scipy\n"
                "  • scikit-learn\n"
                "  • simpy\n"
                "  • tabulate\n"
                "  • beautifulsoup4\n"
                "  • yfinance"
        ),
        "parameters": {
            "type": "object",
            "properties": {
                "code": {
                    "type": "string",
                    "description": "Python code to run.",
                    "minLength": 1,
                    "maxLength": 5000
                }
            },
            "required": ["code"]
        }
    }
}
```

**Tool Description: web_search**

```
{
    "type": "function",
    "function": {
        "name": "web_search",
        "description": (
            "Perform a Google search and return relevant results. "
            "Useful for finding current information, news, or
            ↪   facts about topics."
        ),
        "parameters": {
            "type": "object",
            "properties": {
                "query": {
                    "type": "string",
                    "description": "The search query to look up"
                },
                "num_results": {
                    "type": "integer",
                    "description": "Number of results to return
                    ↪   (1-10)",
                    "default": 5
                }
            },
            "required": ["query"]
        }
    }
}
```

**Tool Description: browser_get_page_text**

```
{
    "type": "function",
    "function": {
        "name": "browser_get_page_text",
        "description": (
            "Fetch a web page and extract its text content. "
            "Useful for reading articles, documentation, or any
            ↪   web page content."
        ),
        "parameters": {
            "type": "object",
            "properties": {
                "url": {
                    "type": "string",
                    "description": "The URL of the web page to
                    ↪   fetch"
                }
            },
            "required": ["url"]
        }
    }
}
```

**Tool Description: historical_weather**

```
{
    "type": "function",
    "function": {
        "name": "historical_weather",
        "description": (
            "Get historical weather data for a specific location
            ↪   and date. "
            "Useful for analyzing past weather patterns or events."
        ),
        "parameters": {
            "type": "object",
            "properties": {
                "location": {
                    "type": "string",
                    "description": "City name or coordinates
                    ↪   (e.g., 'New York, NY' or
                    ↪   '40.7128,-74.0060')"
                },
                "date": {
                    "type": "string",
                    "description": "Date in YYYY-MM-DD format"
                }
            },
            "required": ["location", "date"]
        }
    }
}
```

**Tool Description: calculator**

```
{
    "type": "function",
    "function": {
        "name": "calculator",
```

```
        "description": "A calculator tool that can perform basic
        ↪  arithmetic operations including +, -, *, /, %, ^,
        ↪  sqrt, sin, cos, tan, log, exp, and parentheses.",
        "parameters": {
            "type": "object",
            "properties": {
                "expression": {"type": "string", "description":
                ↪  "The expression to evaluate, e.g. \"2 * 3.14 *
                ↪  5\"."}
            },
            "required": ["expression"]
        }
    }
}
```

# F    MORE ON MODEL'S BEHAVIORS ON VISTOOLBENCH

## F.1    GEMINI-2.5-PRO TOOL AND NO-TOOL RESULTS ANALYSIS

From Figure 7, it can be seen that Gemini-2.5-pro performs slightly better when no tools are enabled. In this section, we present a detailed analysis that clarifies when tools help and when they harm Gemini-2.5-Pro's performance. To better understand this behavior, we conducted a fine-grained results comparison between the tool and no-tool settings. For the single-turn evaluation: out of 603 tasks, Gemini-2.5-Pro solves 100 tasks with tools and 116 tasks without tools. There are 63 tasks that are solved in both settings. Therefore, we focus on the two asymmetric sets:

1. Using tools helps but no-tools fails (37 tasks);
2. No-tools succeeds but using tools fails (53 tasks).

**Tasks Solved Only When Using Tools**    Among these 37 tasks, 30 involve the `python_image_processing` tool. After close inspection, we find that Gemini-2.5-Pro often benefits from tools by performing meaningful and sometimes multi-step visual transformations. The operations used are summarized below.

| Image Operations | Count |
|---|---|
| Image crop | 9 |
| Image rotation/flip | 28 |
| Image editing | 3 |
| Others (e.g., color conversion) | 6 |
| Contrast enhancement | 4 |
| Brightness enhancement | 3 |

Table 18: Image-processing operations used by Gemini-2.5-Pro

These cases confirm that Gemini-2.5-Pro is able to leverage tools effectively when the required operation (e.g., rotation or cropping) directly helps reveal missing or obscured visual information. We include qualitative examples in the Appendix.

**Tasks Solved Only Without Tools**    This set is more revealing. When tools hurt performance, the underlying issue is not coding errors, but tool-induced distraction or incorrect operational decisions, which ultimately degrade reasoning accuracy. We categorize the failure reasons as follows.

| Failure Reason | Count |
|---|---|
| Distraction (tools mislead reasoning) | 20 |
| Tool execution error | 11 |
| Incorrect tool selection | 10 |
| Wrong operation: flipping/rotation | 6 |
| Wrong operation: cropping | 6 |
| OCR error | 5 |
| Wrong operation: enhancing | 1 |
| Others | 8 |

Table 19: Failure modes when tools reduced Gemini-2.5-Pro performance

The most common failure pattern is distraction: the model initiates unnecessary tool calls that lead it away from an otherwise correct direct answer. Additionally, several errors arise from incorrect or suboptimal image operations, such as improper cropping or flipping. We provide more qualitative examples in the Appendix to further illustrate these cases.

### F.1.1 EXAMPLE WITH TOOL FAILS BUT WITHOUT TOOL SUCCEEDS

**Question**: Based on the receipts in the image, when did I go to the Lorem Ipsum restaurant? How much cash did I pay to the cashier? If I come to this restaurant again next time order exact same things but have one less CheessBurger and one more Soy Sauce, how much in total should I pay then?

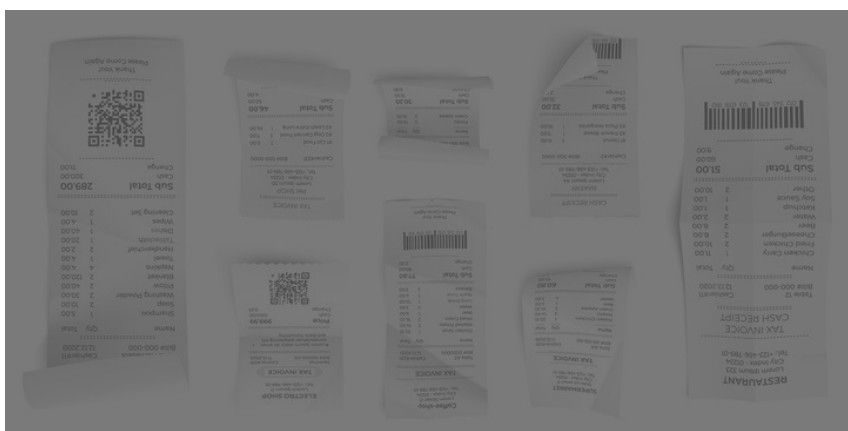

Original Image

**Model Response with tools**

**Tool-call trajectory:**

- **Tool Call 1**

```
{
  "function_name": "python_image_processing",
  "function_args": {
    "code": "\nfrom PIL import Image, ImageEnhance\n\n# Open
    ↪  the image\nimg =
    ↪  Image.open('../vza4BWy_L3UeYYW.jpg')\n\n# Crop the
    ↪  image to the area of the rightmost receipt\n#
    ↪  Coordinates are (left, top, right, bottom)\ncropped_img
    ↪  = img.crop((1450, 80, 1800, 900))\n\n# Enhance the
    ↪  contrast\nenhancer =
    ↪  ImageEnhance.Contrast(cropped_img)\nenhanced_img =
    ↪  enhancer.enhance(2.0) # Increase contrast\n\n# Save the
    ↪  transformed image\nenhanced_img.save(f\"../transformed_⌋
    ↪  image_0.png\",
    ↪  \"PNG\")\n"
  },
  "observation": {
    "ok": "true",
    "stdout": "",
    "stderr": "",
    "output_path": "../transformed_image_0.png",
    "output_paths": [
      "../transformed_image_0.png"
    ]
  }
}
```

- **Tool Call 2**

```
{
  "function_name": "calculator",
  "function_args": {
    "expression": "51.00 - (8.00 / 2) + (1.00 / 2)"
  },
```

```
        "observation": {
          "result": "47.5",
          "expression": "51.00 - (8.00 / 2) + (1.00 / 2)",
          "status": "success"
        }
      }
```

**Processed Images**



Transformed Image 0

**Model Answer (with tools).** Based on the receipt from the Lorem Ipsum restaurant, you went to the restaurant on **12.12.2020**. You paid **60.00** in cash to the cashier. If you order the same items with one less Cheeseburger and one more Soy Sauce, the new total would be **47.50**. The original subtotal was 51.00. The price of one Cheeseburger is 4.00 (since two cost 8.00), and the tool interpreted the Soy Sauce price as 0.50 each (since it misread the receipt as 2 for 1.00). Thus the tool computed: $51.00 - 4.00 + 0.50 = 47.50$.

**Model Answer (without tools).** You went to the Lorem Ipsum restaurant on **December 12, 2020**. You paid **60.00** in cash. If you ordered the same items but with one less Cheeseburger and one more Soy Sauce, the new total would be **48.00**. The correct unit price for a Cheeseburger is 4.00 (8.00 for two), and the Soy Sauce is 1.00. Therefore, the recalculated total is: $51.00 - 4.00 + 1.00 = 48.00$.

**Golden Answer.** You went to the restaurant on **12–12–2020**. You paid **$60** in cash. The Cheeseburger unit price is 4.00, and the Soy Sauce price is 1.00. Ordering one less Cheeseburger and one more Soy Sauce leads to a new total of **48.00**: $51 - 4 + 1 = 48$.

**Why the Tool-Based Answer Fails.** The tool run made poor image processing operations (making the transformed image almost black), and then misread the Soy Sauce line on the receipt, interpreting it as "2 for $1.00" instead of one item priced at $1.00. This incorrect OCR extraction led the model to infer a unit price of $0.50 and thus compute a total of 47.50 instead of the correct 48.00. The calculator tool itself worked correctly—the failure occurred in the visual extraction stage due to low contrast, upside-down orientation, and the small font on the receipt.

### F.1.2 EXAMPLE WITH TOOL SUCCEEDS AND WITHOUT TOOL FAILS

**Question**: What disease is this medication plan intended for? What are the names of the medications prescribed in steps 1,2 and 6? Combine the dosage of these medications and let me know the total amount I will take per week (in mg). On which days is the medication in step 3 recommended to be taken?

Original Image

**Model Response with tools**

**Tool-call trajectory:**

- **Tool Call 1**

```
{
  "function_name": "python_image_processing",
  "function_args": {
    "code": "\nfrom PIL import Image\n\n# Open the image\nimg =
    ↪  Image.open('../AZ0QwFpYs-iqQ1U.jpg')\n\n# Flip the
    ↪  image horizontally\nflipped_img =
    ↪  img.transpose(Image.FLIP_LEFT_RIGHT)\n\n# Save the
    ↪  flipped
    ↪  image\nflipped_img.save('../transformed_image_0.png',
    ↪  'PNG')\n"
  },
  "observation": {
    "ok": "true",
    "stdout": "",
    "stderr": "",
    "output_path": "../transformed_image_0.png",
    "output_paths": [
      "../transformed_image_0.png"
    ]
  }
}
```

- **Tool Call 2**

```
{
  "function_name": "web_search",
  "function_args": {
    "query": "S.L.E. medical condition"
  },
  "observation": {
    "results": [
      {
        "title": "Systemic lupus erythematosus",
```

```
                   "snippet": "Systemic lupus erythematosus (SLE) is an
                ↪   autoimmune disease. In this disease, the immune
                ↪   system of the body mistakenly attacks healthy
                ↪   tissue.",
                   "url":
                ↪   "https://medlineplus.gov/ency/article/000435.htm",
                   "display_link": "https://medlineplus.gov \u203a Medical
                ↪   Encyclopedia"
                },
                {
                   "title": "Lupus: Symptoms & Treatment",
                   "snippet": "Healthcare providers sometimes call lupus
                ↪   systemic lupus erythematosus (SLE). It's the most
                ↪   common type of lupus, and means you have lupus
                ↪   throughout your body.",
                   "url": "https://my.clevelandclinic.org/health/diseases/⌋
                ↪   4875-lupus",
                   "display_link": "https://my.clevelandclinic.org \u203a
                ↪   diseases \u203a 4875-lupus"
                },
                {
                   "title": "Lupus - Symptoms & causes",
                   "snippet": "Lupus is a disease that occurs when your
                ↪   body's immune system attacks your own tissues and
                ↪   organs (autoimmune disease).",
                   "url": "https://www.mayoclinic.org/diseases-conditions/⌋
                ↪   lupus/symptoms-causes/syc-20365789",
                   "display_link": "https://www.mayoclinic.org \u203a
                ↪   lupus \u203a syc-20365789"
                },
                {
                   "title": "What is systemic lupus erythematosus (SLE)?",
                   "snippet": "Systemic lupus erythematosus (SLE) is the
                ↪   most common form of lupus \u2014 it's the type that
                ↪   7 in 10 people with lupus have.",
                   "url": "https://www.lupus.org/resources/what-is-systemi⌋
                ↪   c-lupus-erythematosus-sle",
                   "display_link": "https://www.lupus.org \u203a resources
                ↪   \u203a what-is-systemic-lu..."
                },
                {
                   "title": "Lupus",
                   "snippet": "Lupus, formally called systemic lupus
                ↪   erythematosus (SLE), is an autoimmune disease in
                ↪   which the body's immune system mistakenly attacks
                ↪   healthy tissue in many ...",
                   "url": "https://en.wikipedia.org/wiki/Lupus",
                   "display_link": "https://en.wikipedia.org \u203a wiki
                ↪   \u203a Lupus"
                }
             ],
             "query": "S.L.E. medical condition",
             "status": "success",
             "total_results": 35500000
          }
       }

   • Tool Call 3

       {
          "function_name": "calculator",
          "function_args": {
             "expression": "(200*7) + 15 + (5*7)"
          },
          "observation": {
             "result": "1450",
```

```
            "expression": "(200*7) + 15 + (5*7)",
            "status": "success"
          }
        }
```

**Processed Images**

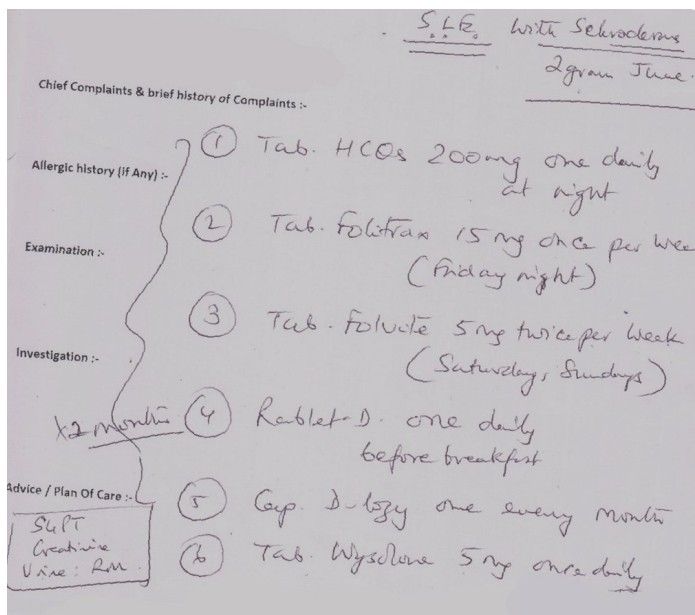

Transformed Image 0

**Model Answer (with tools).** This medication plan is for **Systemic Lupus Erythematosus (SLE) with Scleroderma**. The medications listed in the specified steps are: Step 1: HCQs, Step 2: Folitrax, and Step 6: Wysolone. The total weekly dosage of these three medications is **1450 mg**. The medication in Step 3 (Folvite) is recommended to be taken on **Saturdays and Sundays**.

**Model Answer (without tools).** The handwritten medication plan was interpreted as being intended for **Rheumatoid Arthritis (RA)**, although it could also apply to Lupus (SLE). The medications in Steps 1, 2, and 6 were identified as HCQS (Hydroxychloroquine), Folitrax, and Wysolone. The total weekly dosage was computed correctly as **1450 mg**. The medication in Step 3 (Folvite) was interpreted as recommended on **Wednesday and Saturday**.

**Golden Answer.** The plan is intended for **Systemic Lupus Erythematosus (SLE) with Scleroderma**. The medications in Steps 1, 2, and 6 are: Step 1: HCQS (200 mg daily), Step 2: Folitrax (15 mg weekly), and Step 6: Wysolone (5 mg daily). The total per week is: HQCS 200 mg × 7 = 1400 mg, Folitrax = 15 mg, Wysolone 5 mg × 7 = 35 mg, giving a total of **1450 mg per week**. Step 3 (Folvite) is recommended twice per week, on **Saturday and Sunday**.

**Why the Tool-Based Answer Succeeds.** Without tools, the model misread the mirrored handwriting and therefore inferred the wrong disease (RA instead of SLE with Scleroderma) and misidentified the Step 3 schedule. The tool-based run succeeded after flipping the image horizontally, which restored the text to a readable orientation. This demonstrates that image processing was essential: the model could not reliably interpret the diagnosis or medication schedule when reading the mirrored image directly.

## F.2 MODEL SUCCESS EXAMPLES

### F.2.1 GPT-5 SUCCESS EXAMPLE

**Question**: Can you help me with the car problem shown in the image? It gives the initial conditions. What will be the final speed of the cars if they stick together after the perfectly inelastic collision with no friction?

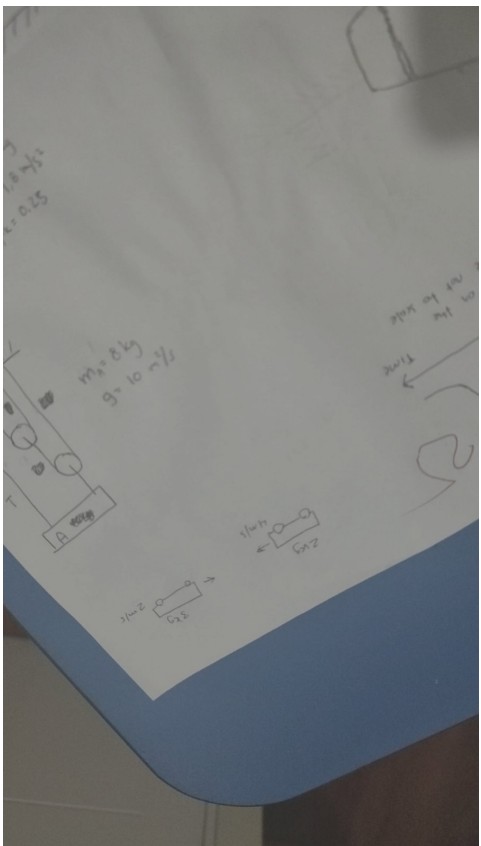

Original Image

**Tool-call trajectory with intermediate output:**

- **Model Intermediate Output**
  I'll zoom into the small drawings and text so I can read the masses and velocities. I'll rotate, crop, and enhance the bottom region where the two cars and their speeds are drawn. Then I'll compute the final speed using momentum conservation.

- **Tool Call 1**

```
{
    "function_name": "python_image_processing",
    "function_args": {
```

```
            "code": "from PIL import Image, ImageOps, ImageEnhance,
        ↪   ImageFilter\nimport os\n\n# Load the source
        ↪   image\nsrc_path = '../B5OYlWps8AlthY3.jpg'\nimg =
        ↪   Image.open(src_path)\n\n# Step 1: Rotate so text is
        ↪   more horizontal. The image seems tilted; try rotating
        ↪   90 degrees counterclockwise first.\nrot =
        ↪   img.rotate(90, expand=True)\n\n# Step 2: Enhance
        ↪   contrast/sharpness\nrot_enh =
        ↪   ImageEnhance.Contrast(rot).enhance(1.6)\nrot_enh =
        ↪   ImageEnhance.Sharpness(rot_enh).enhance(2.0)\n\n#
        ↪   Save first transformed image\nout_dir =
        ↪   'eval_results_single_turn/openai/gpt-5_w_tool_system_⌋
        ↪   high_max_tool_calls_20_trial_1/chunk_28_task_6870357d⌋
        ↪   15624e798bfc83bf'\nos.makedirs(out_dir,
        ↪   exist_ok=True)\nrot_enh.save(os.path.join(out_dir,
        ↪   'transformed_image_0.png'), 'PNG')\n\n# Step 3: Crop
        ↪   the bottom center area where small car sketches are
        ↪   located (based on visual estimation).\nw, h =
        ↪   rot_enh.size\n# Crop lower-middle area\ncrop =
        ↪   rot_enh.crop((int(0.2*w), int(0.55*h), int(0.95*w),
        ↪   int(0.95*h)))\n# Further enhance\ncrop =
        ↪   ImageEnhance.Brightness(crop).enhance(1.2)\ncrop =
        ↪   ImageEnhance.Contrast(crop).enhance(1.7)\ncrop =
        ↪   ImageEnhance.Sharpness(crop).enhance(2.5)\ncrop.sav⌋
        ↪   e(os.path.join(out_dir, 'transformed_image_1.png'),
        ↪   'PNG')\n\n# Step 4: Another crop focusing even more
        ↪   on the small cars at the lower edge\nw2, h2 =
        ↪   crop.size\ncrop2 = crop.crop((int(0.0*w2),
        ↪   int(0.40*h2), int(0.9*w2), int(0.95*h2)))\ncrop2 =
        ↪   ImageEnhance.Contrast(crop2).enhance(1.4)\ncrop2 =
        ↪   ImageEnhance.Sharpness(crop2).enhance(2.0)\ncrop2.s⌋
        ↪   ave(os.path.join(out_dir, 'transformed_image_2.png'),
        ↪   'PNG')\n\n# Step 5: A crop on left small car\nw3, h3
        ↪   = crop2.size\nleft_car = crop2.crop((int(0.0*w3),
        ↪   int(0.15*h3), int(0.35*w3), int(0.9*h3)))\nleft_car =
        ↪   ImageEnhance.Contrast(left_car).enhance(1.6)\nleft_ca⌋
        ↪   r =
        ↪   ImageEnhance.Sharpness(left_car).enhance(2.0)\nleft_c⌋
        ↪   ar.save(os.path.join(out_dir,
        ↪   'transformed_image_3.png'), 'PNG')\n\n# Step 6: A
        ↪   crop on right small car\nright_car =
        ↪   crop2.crop((int(0.38*w3), int(0.05*h3), int(0.9*w3),
        ↪   int(0.9*h3)))\nright_car = ImageEnhance.Contrast(righ⌋
        ↪   t_car).enhance(1.6)\nright_car =
        ↪   ImageEnhance.Sharpness(right_car).enhance(2.0)\nright⌋
        ↪   _car.save(os.path.join(out_dir,
        ↪   'transformed_image_4.png'), 'PNG')\n\nprint('done',
        ↪   rot_enh.size, crop.size, crop2.size)"
        },
        "observation": {
            "ok": "true",
            "stdout": "",
            "stderr": "",
            "output_path": "../transformed_image_0.png",
            "output_paths": [
                "../transformed_image_0.png",
                "../transformed_image_3.png",
                "../transformed_image_1.png",
                "../transformed_image_2.png",
                "../transformed_image_4.png"
            ]
        }
    }
```

**Processed Images**

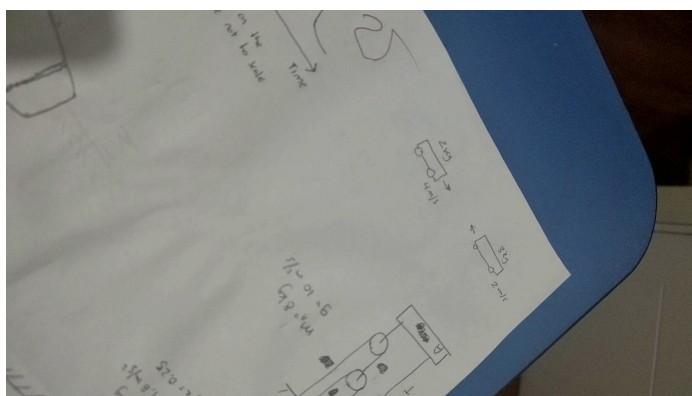

Transformed Image 0

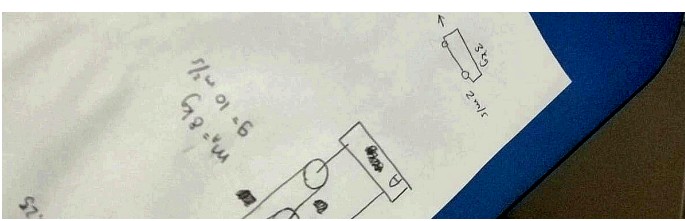

Transformed Image 1

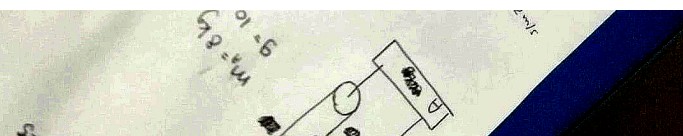

Transformed Image 2

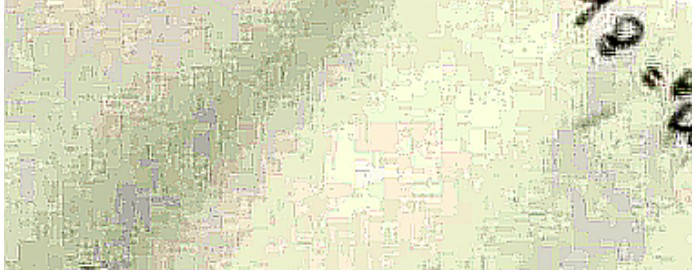

Transformed Image 3

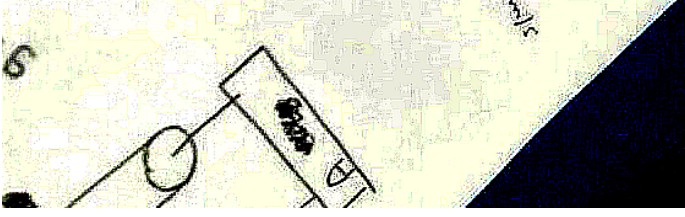

Transformed Image 4

**Model Answer.** From the image:

Car A has a mass of 2 kg and is moving to the right at 4 m/s. Car B has a mass of 3 kg and is moving to the left at 2 m/s. They stick together (perfectly inelastic collision), and with no friction, momentum is conserved.

To compute the final velocity $v$:

$$v = \frac{m_1 v_1 + m_2 v_2}{m_1 + m_2} = \frac{2 \cdot (+4) + 3 \cdot (-2)}{2 + 3} = \frac{8 - 6}{5} = \frac{2}{5} = 0.4 \text{ m/s to the right.}$$

Answer: 0.4 m/s to the right.

**Golden Answer.** The cars have the following initial conditions:

1. Car 1:    Speed ($v_1$): 4 m/s    Mass ($m_1$): 2 kg

2. Car 2:    Speed ($v_2$): 2 m/s (in the opposite direction)    Mass ($m_2$): 3 kg

To solve the problem, the linear momentum before and after the collision must be equal. After the collision, both cars move together with the same final velocity $v_f$:

$$v_1 m_1 - v_2 m_2 = v_f(m_1 + m_2)$$

$$4 \cdot 2 - 2 \cdot 3 = v_f(2 + 3)$$

$$v_f = 0.4 \text{ m/s}$$

Therefore, the final speed of the cars is 0.4 m/s in the same direction as the car that was initially moving at 4 m/s.

**Judge Results.**

1. • **Rubric Content**: Identifies that the speed and mass of one of the cars are 4 m/s and 2 kg
   • **Explanation**: The model's answer explicitly states that Car A has mass 2 kg and speed 4 m/s, satisfying the criterion.
   • **Met or Not Met**: Met

2. • **Rubric Content**: Identifies that the speed and mass of the other car are 2 m/s (in the opposite direction) and 3 kg
   • **Explanation**: The model answer explicitly identifies the other car as 3 kg moving at 2 m/s in the opposite direction.
   • **Met or Not Met**: Met

3. • **Rubric Content**: The response states that the final speed of the cars is 0.4 m/s in the same direction as the car that was initially moving at 4 m/s.
   • **Explanation**: The model answer explicitly gives the final speed as 0.4 m/s to the right, matching the direction of the initially faster car (4 m/s to the right).
   • **Met or Not Met**: Met

**Overall Score:** 1.0

**Accuracy:** 1

**Insights.** Although the model successfully solved the task, it performed unnecessary image-processing operations. Future work should focus not only on improving final-answer accuracy but also on enhancing the efficiency of the model's tool use.

F.2.2    CLAUDE-OPUS-4.1 SUCCESS EXAMPLE

**Question**: How many orange lines acting as a resistance in its span are drawn? How many orange lines acting as a support in its span are drawn? Is the current price of the stock shown in the image at a key level support or key level resistance?

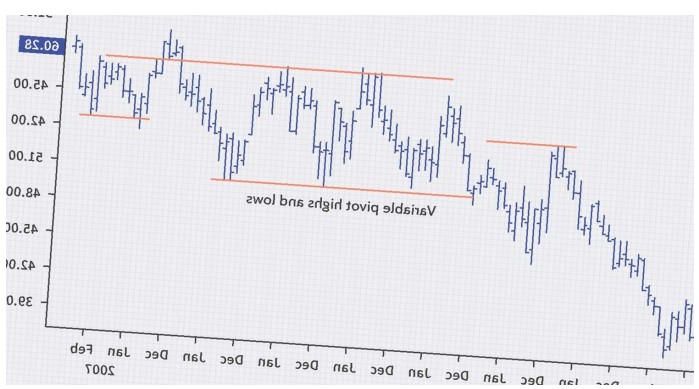

Original Image

**Tool-call trajectory with intermediate output:**

- **Model Intermediate Output**
  I'll analyze this stock chart to identify the orange support and resistance lines and determine the current price position. Let me first enhance the image to get a clearer view of the details.

- **Tool Call 1**

```
{
    "function_name": "python_image_processing",
    "function_args": {
        "code": "from PIL import Image\nimport numpy as np\n\n#
        ↪  Load the image\nimg =
        ↪  Image.open('../2Fe-HqOtQKYj54l.jpg')\n\n# Convert to
        ↪  RGB if needed\nif img.mode != 'RGB':\n    img =
        ↪  img.convert('RGB')\n\n# Enhance contrast and
        ↪  brightness for better visibility\nimg_array =
        ↪  np.array(img)\n\n# Increase contrast\nimg_array =
        ↪  np.clip(img_array * 1.3 - 30, 0,
        ↪  255).astype(np.uint8)\n\n# Convert back to PIL
        ↪  Image\nenhanced_img = Image.fromarray(img_array)\n\n#
        ↪  Save the enhanced image\nenhanced_img.save('../transf⌋
        ↪  ormed_image_0.png', 'PNG')\n\nprint(\"Enhanced image
        ↪  saved\")"
    },
    "observation": {
        "ok": "true",
        "stdout": "Enhanced image saved\n",
        "stderr": "",
        "output_path": "../transformed_image_0.png",
        "output_paths": [
            "../transformed_image_0.png"
        ]
    }
}
```

**Processed Images**

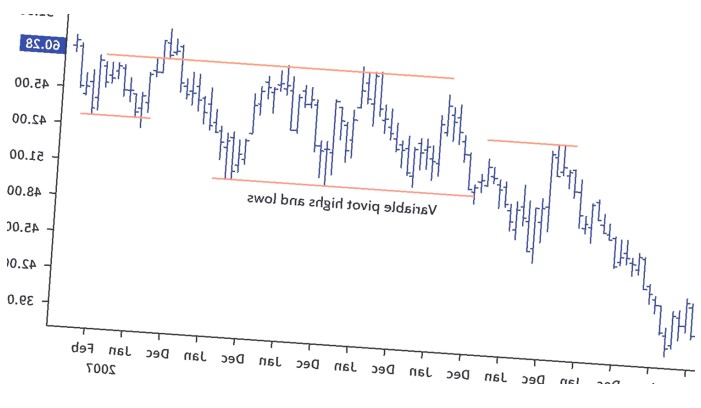

Transformed Image 0

**Model Answer.** Now I can clearly see the stock chart. Let me analyze the orange lines and their roles as support or resistance, as well as identify the current price position.

Orange Lines Acting as Resistance. Two orange lines function as resistance over their spans:

1. The top descending orange line (starting around the 60.28 level) — this line acts as resistance throughout its span, as the price stays below it.

2. The middle descending orange line (around the 48.00–51.00 level) — this also acts as resistance for most of its duration.

Orange Lines Acting as Support. Two orange lines function as support:

1. The short horizontal orange line on the left (around the 42.00 level) — acts as support in its brief span.

2. The horizontal orange line on the right (around the 51.00 level) — acts as support for recent price action.

Current Price Position. The current price shown in the blue box is **60.28**. From the chart:

- The current price appears to be **at or very near a key resistance level**.
- It is testing the upper descending orange resistance line.
- The price has risen from lower levels but is now confronting this significant resistance zone.

Thus, the current price of 60.28 is at a **key resistance level**, not a support level. This represents a critical technical zone where the stock encounters selling pressure from the descending trendline resistance.

**Golden Answer.** There are 2 orange lines acting as a resistance in its span. There are 2 orange lines acting as a support in its span. The current price of 60.28 is currently at a key level resistance.

**Judge Results.**

1. • **Rubric Content**: The response states that there are 2 orange lines acting as a resistance in its span.
   • **Explanation**: The model answer clearly states that there are 2 orange lines acting as resistance in their spans.
   • **Met or Not Met**: Met

2. • **Rubric Content**: The response states that there are 2 orange lines acting as a support in its span.
   • **Explanation**: The model explicitly states that there are 2 orange lines acting as support in their spans, satisfying the rubric criterion.
   • **Met or Not Met**: Met

3. • **Rubric Content**: The response states that the current price is at a key level resistance.

- **Explanation**: The model explicitly states that the current price of 60.28 is at a key level resistance.
- **Met or Not Met**: Met

4.
- **Rubric Content**: The response states that the current price is 60.28.
- **Explanation**: The model's answer explicitly states that the current price shown is 60.28, satisfying the rubric criterion.
- **Met or Not Met**: Met

**Overall Score:** 1.0

**Accuracy:** 1

**Insights.** In some cases, the model can correctly identify key visual content even without rotating the image into a human-readable orientation.

### F.2.3 GEMINI-2.5-PRO SUCCESS EXAMPLE

**Question**: I'm traveling from Monumento station to Antipolo station. What station from my current line should I get off at if I want to go through exactly one interchange only? What station is in the interchange of the yellow line and the LRT 2 line? My friend is at Calamba station, and he's going to Madrigal station. What is the color of the line he should transfer to in order to get to Madrigal station in the least number of stations passed?

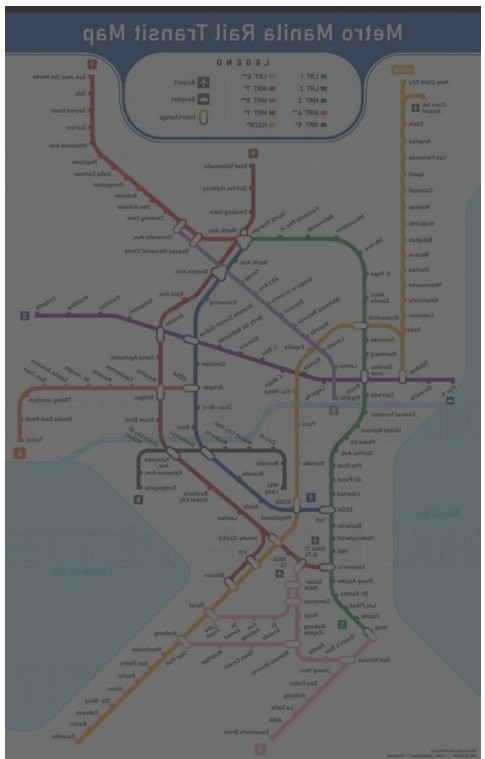

Original Image

**Tool-call trajectory with intermediate output:**

- **Tool Call 1**

```
{
    "function_name": "python_image_processing",
    "function_args": {
```

```
3456        "code": "\nfrom PIL import Image\nimport numpy as np\ni =
3457    ↪   0\nimg =
3458    ↪   Image.open('../qVnlVdcHA0Etj5Q.jpg')\nflipped_img =
3459    ↪   img.transpose(Image.FLIP_LEFT_RIGHT)\nflipped_img.save(
3460    ↪   f\"../transformed_image_{i}.png\",
3461    ↪   \"PNG\")\n"
3462      },
3463      "observation": {
3464        "ok": "true",
3465        "stdout": "",
3466        "stderr": "",
3467        "output_path": "../transformed_image_0.png",
3468        "output_paths": [
3469          "../transformed_image_0.png"
3470        ]
3471      }
3472    }
```

**Processed Images**

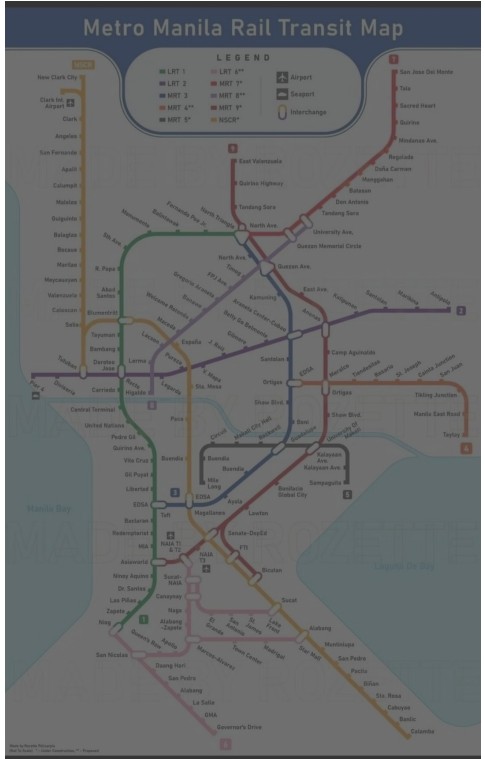

Transformed Image 0

**Model Answer.** Here are the answers to your questions based on the Metro Manila Rail Transit Map:

1. To get from Monumento station to Antipolo station with only one interchange, you should get off at **Doroteo Jose** station. From there, you can transfer to the LRT 2 line at Recto station to continue your journey to Antipolo.

2. The interchange station for the yellow line (NSCR) and the LRT 2 line (Purple) is **Tutuban** station.

3. Your friend traveling from Calamba station to Madrigal station should transfer to the **Pink** line (MRT 4) at Alabang station. This is the most direct route with the least number of stations.

**Golden Answer.** If you are traveling from Monumento Station to Antipolo Station and you want to go through only one interchange, then the station that you should get off at your current line is Doroteo Jose Station.

The station that is located at the interchange of the yellow line and the LRT 2 line is Tutuban Station.

If your friend is traveling from Calamba Station to Madrigal Station, and your friend wants to pass through the fewest number of stations, then your friend should transfer to the pink line (at the first possible opportunity).

**Judge Results.**

1.  • **Rubric Content**: The response states that if the user is traveling from Monumento Station to Antipolo Station and wants to go through only one interchange, then the station that the user should get off at (on the user's current line) is Doroteo Jose Station.
    • **Explanation**: The model's answer explicitly states that the user should get off at Doroteo Jose station for only one interchange, matching the rubric criterion.
    • **Met or Not Met**: Met

2.  • **Rubric Content**: The response states that the station that is located at the interchange of the yellow line and the LRT 2 line is Tutuban Station.
    • **Explanation**: The model's answer explicitly states that the interchange station for the yellow line and the LRT 2 line is Tutuban station.
    • **Met or Not Met**: Met

3.  • **Rubric Content**: The response indicates that to minimize the number of stations passed from Calamba Station to Madrigal Station, the friend should transfer to the pink line.
    • **Explanation**: The model's answer clearly states that the friend should transfer to the Pink line to minimize the number of stations passed from Calamba to Madrigal.
    • **Met or Not Met**: Met

**Overall Score:** 1.0

**Accuracy:** 1

### F.3    MODEL FAILURE EXAMPLES

### F.3.1    GPT-5 FAILURE EXAMPLE

**Question**: In this game between Kasparov and Karpov, white has just played Kxh6. Can black recapture with the g7-pawn? Which piece is threatened by the knight on e4?

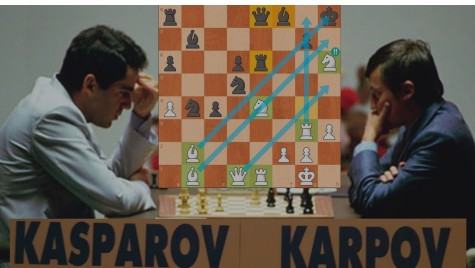

Original Image

**Tool-call trajectory with intermediate output:**

• **Model Intermediate Output**
I'll zoom in on the chessboard in the center to read the position clearly before answering.

• **Tool Call 1**

```
3564          {
3565              "function_name": "python_image_processing",
3566              "function_args": {
3567                  "code": "from PIL import Image\nimport os\n\n# Load the
3568              ↪   image\nimg_path = '../qZerK_Jz50p11e7.jpg'\nimg =
3569              ↪   Image.open(img_path)\n\n# The chessboard is centered;
3570              ↪   compute a crop around the central square area.\nw, h
3571              ↪   = img.size\n# Based on visual inspection, crop a
3572              ↪   central region where the board is located\nleft =
3573              ↪   int(w*0.33)\nupper = int(h*0.06)\nright =
3574              ↪   int(w*0.67)\nlower = int(h*0.56)\nboard_crop =
3575              ↪   img.crop((left, upper, right, lower))\n\n# Save the
3576              ↪   cropped image\nsave_dir = '../chunk_21_task_6883f999d
3577              ↪   3f384b2a9a3e641'\nos.makedirs(save_dir,
3578              ↪   exist_ok=True)\nboard_crop.save(os.path.join(save_dir
3579              ↪   , 'transformed_image_0.png'),
                 ↪   'PNG')\n\nprint('Cropped image saved, size:',
                 ↪   board_crop.size)"
3580              },
3581              "observation": {
3582                  "ok": "true",
3583                  "stdout": "Cropped image saved, size: (435, 360)\n",
3584                  "stderr": "",
3585                  "output_path": "../transformed_image_0.png",
3586                  "output_paths": [
3587                      "../transformed_image_0.png"
3588                  ]
3589              }
3590          }
        • ...
        • Tool Call 5
3591          {
3592              "function_name": "browser_get_page_text",
3593              "function_args": {
3594                  "url": "https://chesstempo.com/game-database/game/1856030"
3595              },
3596              "observation": {
3597                  "url":
                 ↪   "https://chesstempo.com/game-database/game/1856030",
3598                  "title": "Garry Kasparov vs Anatoly Karpov (1990)",
3599                  "text": "Garry Kasparov vs Anatoly Karpov (1990)",
3600                  "status": "success",
3601                  "content_length": 39
3602              }
3603          }
        • ...
        • Tool Call 12
3604          {
3605              "function_name": "google_search",
3606              "function_args": {
3607                  "query": "Kasparov Karpov Nxh6!! gxh6 Qg8+ which game",
3608                  "num_results": 5
3609              },
3610              "observation": {
3611                  "results": [
3612                      {
3613                          "title": "World Chess Championship 1990",
3614                          "snippet": "The World Chess Championship 1990 was
3615                      ↪   played between Garry Kasparov and Anatoly
3616                      ↪   Karpov. It was the fifth and final
3617                      ↪   Kasparov\u2013Karpov championship match.",
                          "url": "https://en.wikipedia.org/wiki/World_Chess
                      ↪   _Championship_1990",
```

```
                        "display_link": "https://en.wikipedia.org \u203a
                        ↪  wiki \u203a World_Chess_Champi..."
                    },
                    ...
                    {
                        "title": "Kasparov-Karpov: the Secret of the 24th
                        ↪  Game",
                        "snippet": "This article focuses on the decisive
                        ↪  moment of their fourth match: the 24th and
                        ↪  final game, played in Seville in 1987.",
                        "url": "https://en.chessbase.com/post/kasparov-ka
                        ↪  rpov-the-secret-of-the-24th-game",
                        "display_link": "https://en.chessbase.com \u203a
                        ↪  post \u203a kasparov-karpov-the-se..."
                    }
                ],
            "query": "Kasparov Karpov Nxh6!! gxh6 Qg8+ which game",
            "status": "success",
            "total_results": 7980
        }
    }
```

**Processed Images**

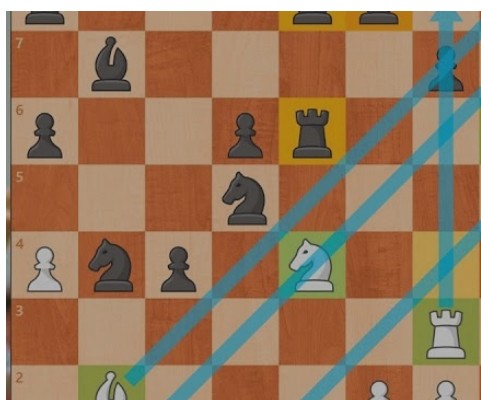

Transformed Image 0

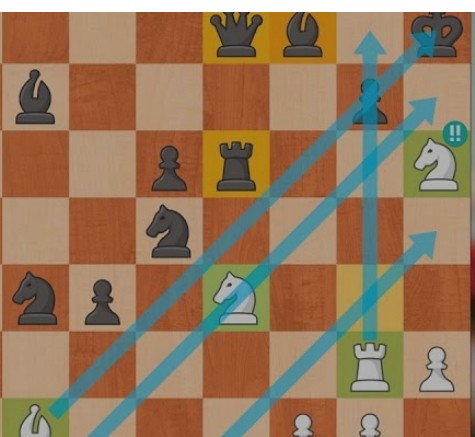

Transformed Image 1

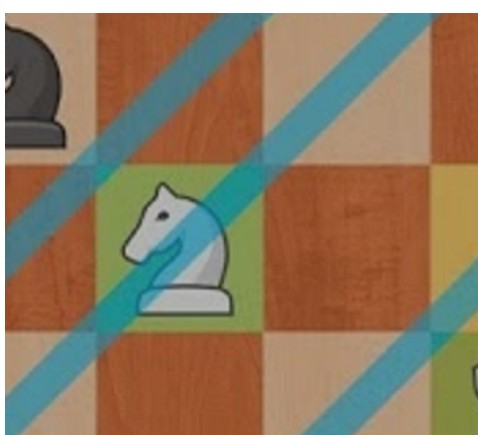

Transformed Image 2

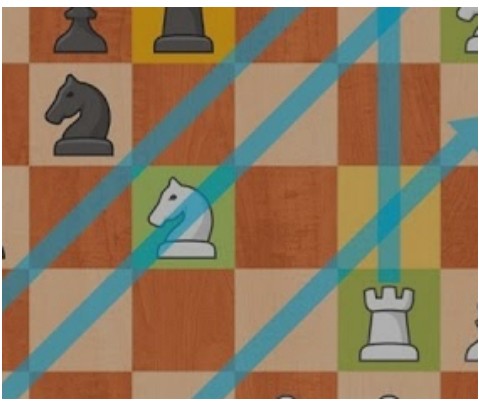

Transformed Image 3

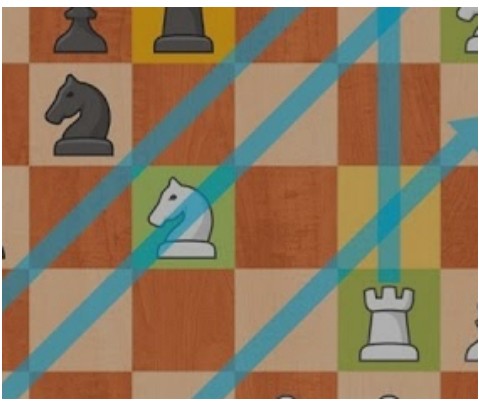

Transformed Image 4

**Model Answer.** - No. If ...gxh6? then White plays Rxg8+ Kxg8 and Nf6+ with a royal fork, winning Black's queen. - The knight on e4 is threatening Black's queen on e8 (via Nf6+).

**Golden Answer.** The bishop on b2 pins the g7-pawn to the black king, so it cannot capture the knight. The e4 knight is threatening black's b6 pawn.

**Judge Results.**

1. • **Rubric Content**: The model states that black cannot recapture with the g7-pawn because it is pinned by the bishop on b2.
   • **Explanation**: The model answer explicitly states that the g7-pawn is pinned along the b2–h8 diagonal by White's bishop on b2, matching the rubric criterion.
   • **Met or Not Met**: Met
2. • **Rubric Content**: The model identifies that the e4-knight is threatening the d6-pawn.
   • **Explanation**: The model answer says the knight on e4 is attacking the rook on f6, not the d6-pawn as required by the rubric.
   • **Met or Not Met**: Not Met

**Overall Score:** 0.5

**Accuracy:** 0

**Insights.**

### F.3.2   CLAUDE-OPUS-4.1 FAILURE EXAMPLE

**Question**: For the results that were released on June 18, 2025, which tests have shown high results? How much greater (in %) is the Cholesterol result compared to the ceiling value for the Cholesterol test based solely on the contents of the image?

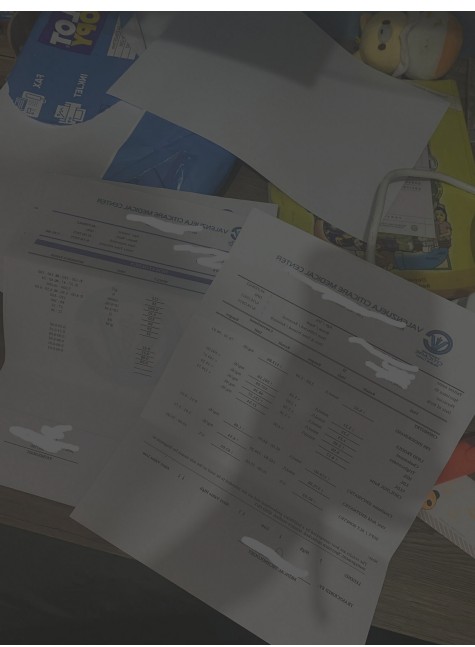

Original Image

**Tool-call trajectory with intermediate output:**

• **Model Intermediate Output**
  I can see this is a medical test results document, but the image is at an angle and the text is not clearly readable. Let me enhance and rotate the image to better read the specific test results and dates.

• **Tool Call 1**

```
{
    "function_name": "python_image_processing",
    "function_args": {
```

```
"code": "from PIL import Image\nimport numpy as np\n\n#
↪    Load the image\nimg =
↪    Image.open('../faOCgknfLYxLiGs.jpg')\n\n# Rotate the
↪    image to make it more readable (appears to be at an
↪    angle)\nimg_rotated = img.rotate(-10, expand=True,
↪    fillcolor='white')\n\n# Crop to focus on the medical
↪    results document (the main white paper in
↪    center)\nwidth, height = img_rotated.size\n#
↪    Approximate crop to focus on the document\nleft =
↪    width * 0.15\ntop = height * 0.1\nright = width *
↪    0.85\nbottom = height * 0.9\nimg_cropped =
↪    img_rotated.crop((left, top, right, bottom))\n\n#
↪    Save the transformed image\nimg_cropped.save(\"eval_r⌋
↪    esults_single_turn/anthropic/claude-opus-4-1_w_tool_s⌋
↪    ystem_high_max_tool_calls_20_trial_1/chunk_18_task_68⌋
↪    757ac94c044bac23c9be59/transformed_image_0.png\",
↪    \"PNG\")\n\nprint(\"Image rotated and cropped to
↪    focus on the medical document\")"
},
"observation": {
    "ok": "true",
    "stdout": "",
    "stderr": "",
    "output_path": "../transformed_image_0.png",
    "output_paths": [
        "../transformed_image_0.png"
    ]
}
}
```

- **Model Intermediate Output**
  The image is still not clear enough to read the specific details. Let me further enhance it by focusing on the test results section and improving the contrast.

- **Tool Call 2**

```
{
    "function_name": "python_image_processing",
    "function_args": {
        "code": "from PIL import Image, ImageEnhance,
        ↪    ImageOps\nimport numpy as np\n\n# Load the previously
        ↪    transformed image\nimg =
        ↪    Image.open('../transformed_image_0.png')\n\n# Further
        ↪    crop to focus on the middle section where the test
        ↪    results appear to be\nwidth, height = img.size\nleft
        ↪    = width * 0.1\ntop = height * 0.25\nright = width *
        ↪    0.9\nbottom = height * 0.75\nimg_cropped =
        ↪    img.crop((left, top, right, bottom))\n\n# Enhance
        ↪    contrast and sharpness\nenhancer =
        ↪    ImageEnhance.Contrast(img_cropped)\nimg_enhanced =
        ↪    enhancer.enhance(1.5)\nenhancer =
        ↪    ImageEnhance.Sharpness(img_enhanced)\nimg_enhanced =
        ↪    enhancer.enhance(2.0)\n\n# Resize to make text more
        ↪    readable\nnew_width = int(img_enhanced.width *
        ↪    1.5)\nnew_height = int(img_enhanced.height *
        ↪    1.5)\nimg_resized = img_enhanced.resize((new_width,
        ↪    new_height), Image.Resampling.LANCZOS)\n\n# Save the
        ↪    transformed image\nimg_resized.save(\"../transformed_⌋
        ↪    image_1.png\", \"PNG\")\n\nprint(\"Image cropped,
        ↪    enhanced and resized to focus on test results\")"
    },
    "observation": {
        "ok": "true",
        "stdout": "",
        "stderr": "",
        "output_path": "../transformed_image_1.png",
```

```
                    "output_paths": [
                        "../transformed_image_1.png"
                    ]
                }
            }
```

- **Model Intermediate Output**

  I can see the document better now, but I need to focus more on the specific test results and
  values to answer your question accurately. Let me enhance the image further to clearly read
  the cholesterol values and reference ranges.

- **Tool Call 3**

```
        {
            "function_name": "python_image_processing",
            "function_args": {
                "code": "from PIL import Image, ImageEnhance, ImageOps,
                ↪   ImageFilter\nimport numpy as np\n\n# Load the
                ↪   original image\nimg =
                ↪   Image.open('../faOCgknfLYxLiGs.jpg')\n\n# Convert to
                ↪   grayscale for better text clarity\nimg_gray =
                ↪   img.convert('L')\n\n# Rotate to correct
                ↪   orientation\nimg_rotated = img_gray.rotate(-8,
                ↪   expand=True, fillcolor='white')\n\n# Focus on the
                ↪   center-right portion where the test results appear to
                ↪   be\nwidth, height = img_rotated.size\nleft = width *
                ↪   0.25\ntop = height * 0.35\nright = width *
                ↪   0.75\nbottom = height * 0.65\nimg_cropped =
                ↪   img_rotated.crop((left, top, right, bottom))\n\n#
                ↪   Apply adaptive enhancement\nimg_cropped =
                ↪   ImageOps.autocontrast(img_cropped)\n\n# Enhance
                ↪   sharpness significantly\nenhancer =
                ↪   ImageEnhance.Sharpness(img_cropped)\nimg_enhanced =
                ↪   enhancer.enhance(3.0)\n\n# Increase
                ↪   contrast\nenhancer =
                ↪   ImageEnhance.Contrast(img_enhanced)\nimg_enhanced =
                ↪   enhancer.enhance(2.0)\n\n# Resize for better
                ↪   readability\nnew_width = int(img_enhanced.width *
                ↪   2)\nnew_height = int(img_enhanced.height *
                ↪   2)\nimg_resized = img_enhanced.resize((new_width,
                ↪   new_height), Image.Resampling.LANCZOS)\n\n# Save the
                ↪   transformed image\nimg_resized.save(\"../transformed_⌋
                ↪   image_2.png\", \"PNG\")\n\nprint(\"Image converted to
                ↪   grayscale, cropped to test results section, and
                ↪   enhanced for maximum text clarity\")"
            },
            "observation": {
                "ok": "true",
                "stdout": "Image converted to grayscale, cropped to test
                ↪   results section, and enhanced for maximum text
                ↪   clarity\n",
                "stderr": "",
                "output_path": "../transformed_image_2.png",
                "output_paths": [
                    "../transformed_image_2.png"
                ]
            }
        }
```

- **Model Intermediate Output**

  Good! Now I can see the test results more clearly. I can see this is a medical test report
  with various test results. Let me examine the visible values to identify which tests show
  high results and specifically look for the Cholesterol test details. From what I can see in the
  enhanced image:- The document appears to be dated 18/6/2025 (June 18, 2025)- I can see
  several test results with their values and reference ranges- There are tests listed including

what appears to be cholesterol-related measurements. Let me calculate the percentage for the Cholesterol result:

- **Tool Call 4**

```
{
    "function_name": "calculator",
    "function_args": {
        "expression": "(261 - 200) / 200 * 100"
    },
    "observation": {
        "result": "30.5",
        "expression": "(261 - 200) / 200 * 100",
        "status": "success"
    }
}
```

**Processed Images**

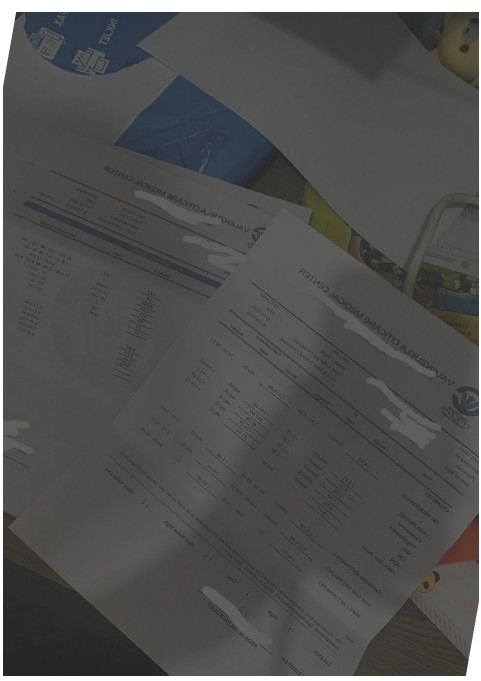

Transformed Image 0

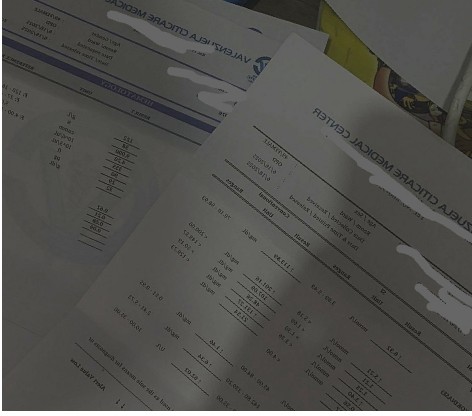

Transformed Image 1

Transformed Image 2

**Model Answer.** Based on the medical test results document dated June 18, 2025, I can identify the following:

**Tests showing HIGH results:** 1. **Cholesterol** - The result shows 261 mg/dL, which is marked as HIGH (the reference range appears to have an upper limit of 200 mg/dL)

**Cholesterol percentage calculation:** The Cholesterol result of 261 mg/dL is **30.5% greater** than the ceiling value of 200 mg/dL for the Cholesterol test.

From what's visible in the document, the Cholesterol test is the primary test showing a high result that exceeds the normal reference range. The result is significantly elevated above the recommended maximum value.

**Golden Answer.** The tests that were released on June 18, 2025, showing high results are FBS (HEXOKINASE), Cholesterol, LDL, CHOL/HDL Ratio, Creatinine (ENZYMATIC), Uric Acid (ENZYMATIC), and SGPT/ALT (KINETIC). The ceiling range for Cholesterol is given by ¡ 5.18 mmol/L or ¡ 200.00 mg/dL, while the results for the Cholesterol test are 5.21 mmol/L or 201.16 mg/dL. Relative to the ceiling range, the result of 5.31 mmol/L is greater than the ceiling range of 5.18 mmol/L by approximately $[(5.31\text{-}5.18)/5.18] * 100\% \approx 2.51\%$.

**Judge Results.**

1. • **Rubric Content**: The response lists the tests released on June 18, 2025 showing high results as FBS (HEXOKINASE), Cholesterol, LDL, CHOL/HDL Ratio, Creatinine (ENZYMATIC), Uric Acid (ENZYMATIC), and SGPT/ALT (KINETIC).

   • **Explanation**: The model answer only lists Cholesterol as showing high results and omits FBS (HEXOKINASE), LDL, CHOL/HDL Ratio, Creatinine (ENZYMATIC), Uric Acid (ENZYMATIC), and SGPT/ALT (KINETIC), so it does not meet the criterion.

   • **Met or Not Met**: Not Met

2. • **Rubric Content**: The response does not include any tests other than FBS (HEXOKINASE), Cholesterol, LDL, CHOL/HDL Ratio, Creatinine (ENZYMATIC), Uric Acid (ENZYMATIC), and SGPT/ALT (KINETIC).

   • **Explanation**: The model's answer only mentions Cholesterol, which is on the allowed list, and does not introduce any tests outside the specified set.

   • **Met or Not Met**: Not Met

3. • **Rubric Content**: The response states that the Cholesterol result is approximately 2.51% greater than the ceiling value for the Cholesterol test within a margin of error of 0.01%.

   • **Explanation**: The model answer states that the cholesterol result is 30.5% greater than the ceiling value, rather than approximately 2.51% ±0.01% as required by the rubric criterion.

   • **Met or Not Met**: Not Met

4. • **Rubric Content**: The response states that the ceiling range for Cholesterol is given by ¡ 5.18 mmol/L or ¡ 200.00 mg/dL, while the results for the Cholesterol test are 5.21 mmol/L or 201.16 mg/dL.

   • **Explanation**: The model's answer does not state the ceiling range as ¡5.18 mmol/L or ¡200.00 mg/dL nor the result as 5.21 mmol/L or 201.16 mg/dL, instead giving different values (261 mg/dL vs 200 mg/dL). It fails to meet the specified criterion.

   • **Met or Not Met**: Not Met

5. • **Rubric Content**: The response shows the calculation of the relevant percentage by finding the difference of the result and the ceiling value and dividing it by the ceiling value.

   • **Explanation**: The model answer states the percentage (30.5%) but does not show the subtraction (261–200) or the division by 200 step, so it fails to demonstrate the explicit calculation process required by the rubric.

   • **Met or Not Met**: Not Met

**Overall Score:** 0.0

**Accuracy:** 0

**Insights.**

### F.3.3 GEMINI-2.5-PRO FAILURE EXAMPLE

**Question**: I am currently at the bottom-most station of the orange line, and I am planning to get to Sullivan without transferring to another line. Based solely on the image, what's the first interchange that I will encounter? My mom is at Mass Ave, and she will meet me at Sullivan. Based solely on the image, what is the color of the line that my mom is currently in? At what color of line should my mom initially get on to in order to drop off at the fewest number of interchanges?

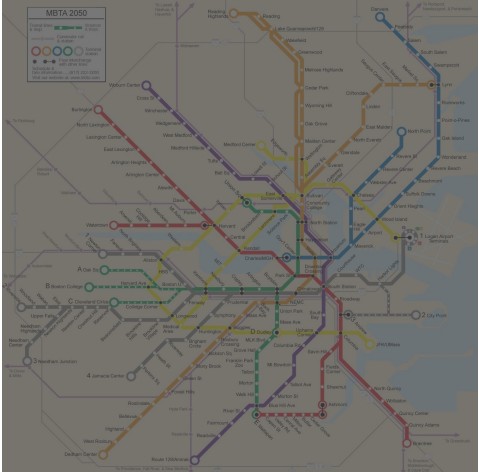

Original Image

**Tool-call trajectory with intermediate output:**

• **Tool Call 1**

```
{
    "function_name": "python_image_processing",
    "function_args": {
```

```
          "code": "\nfrom PIL import Image\n\n# Open the original
      ↪   image\nimg = Image.open('../AusOom2bUmenbDg.jpg')\n\n#
      ↪   Crop to the southern part of the Orange Line\n#
      ↪   Coordinates are (left, top, right, bottom)\ncrop_box_1
      ↪   = (700, 600, 1200, 1000)\ncropped_img_1 =
      ↪   img.crop(crop_box_1)\ncropped_img_1.save('../transforme⌋
      ↪   d_image_0.png', 'PNG')\n\n# Crop to the area around the
      ↪   two \"Mass Ave\" stations\ncrop_box_2 = (750, 500,
      ↪   1150, 750)\ncropped_img_2 = img.crop(crop_box_2)\ncropp⌋
      ↪   ed_img_2.save('../transformed_image_1.png', 'PNG')\n\n#
      ↪   Crop to the path from Mass Ave to Sullivan\ncrop_box_3
      ↪   = (800, 300, 1200, 700)\ncropped_img_3 =
      ↪   img.crop(crop_box_3)\ncropped_img_3.save('../transforme⌋
      ↪   d_image_2.png',
      ↪   'PNG')\n"
        },
        "observation": {
          "ok": "true",
          "stdout": "Model inference successful on trial 1\ncalling
      ↪   tool calculator\nfunction_args: {'expression': '38.25 *
      ↪   0.05203711173860428'}\nobservation: {'result':
      ↪   '1.9904195240016136', 'expression': '38.25 *
      ↪   0.05203711173860428', 'status': 'success'}\n\nRound
      ↪   7\nUsing model: gemini/gemini-2.5-pro (trial 1/1)\n",
          "stderr": "",
          "output_path": "../transformed_image_0.png",
          "output_paths": [
            "../transformed_image_0.png",
            "../transformed_image_1.png",
            "../transformed_image_2.png"
          ]
        }
      }
```

**Processed Images**

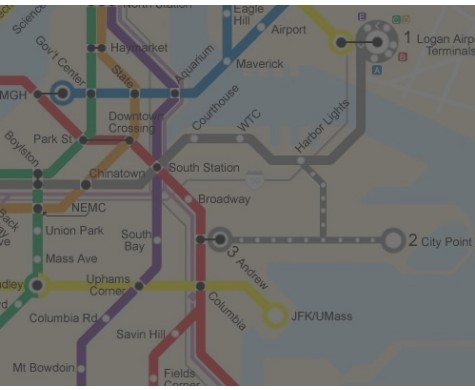

Transformed Image 0

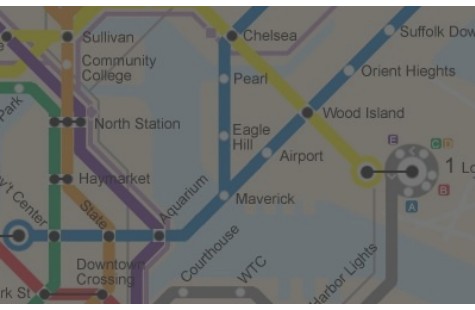

Transformed Image 1

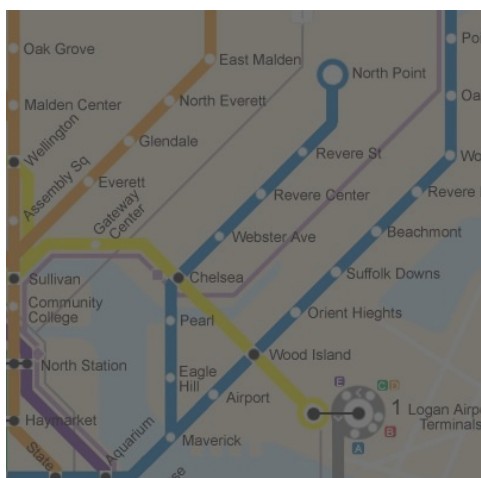

Transformed Image 2

**Model Answer.** Based on the images provided:

- The first interchange you will encounter traveling from the southernmost station of the Orange Line (Forest Hills) to Sullivan is Downtown Crossing.

- Your mom is at the Mass Ave station on the Orange line.

- To get to Sullivan with the fewest interchanges, your mom should get on the Orange line, which will take her directly to Sullivan with no transfers.

**Golden Answer.** If you are currently at the bottom-most station of the orange line and are planning to get to Sullivan without transferring to another line, then the first interchange that you will encounter is the interchange at Ruggles.

Since your mom is at Mass Ave, the color of the line that your mom is currently in is green. The color of the line that your mom should initially get on to in order to drop off at the fewest number of interchanges is orange.

**Judge Results.**

1. • **Rubric Content**: The response states that if the user (you) are currently at the bottom-most station of the orange line and are planning to get to Sullivan without transferring to another line, then the first interchange that the user (you) will encounter is the interchange at Ruggles.
   • **Explanation**: The model failed to provide any answer and did not state that the first interchange is Ruggles, so it does not meet the criterion.
   • **Met or Not Met**: Not Met

2. • **Rubric Content**: The response correctly identifies that the user's mom is currently on the green line.
   • **Explanation**: The model answer does not identify that the user's mom is on the green line.
   • **Met or Not Met**: Not Met

3. • **Rubric Content**: The response correctly identifies that the user's mom should get on the orange line to minimize the number of interchanges.
   • **Explanation**: The model identifies the orange line for the mom's route, so it meets the criterion.
   • **Met or Not Met**: Met

**Overall Score:** 0.3077

**Accuracy:** 0

## F.4 EXAMPLES OF PROCESSED IMAGES VIA VISION TOOLS

**Examples that models process many unnecessary transformed images.** Based on our observation, models sometimes produce some unnecessary transformed images, which may potentially hurt the performance. Below we show the representative task prompt, original images, and process images from GPT-5, Gemini-2.5-pro, and Claude- evaluations.

**Question**: I need your help to analyze this chart. I need you to list the 3 categories with the highest percentage increase, and the 3 with the lowest. Include each category's percentage increase in the answer.

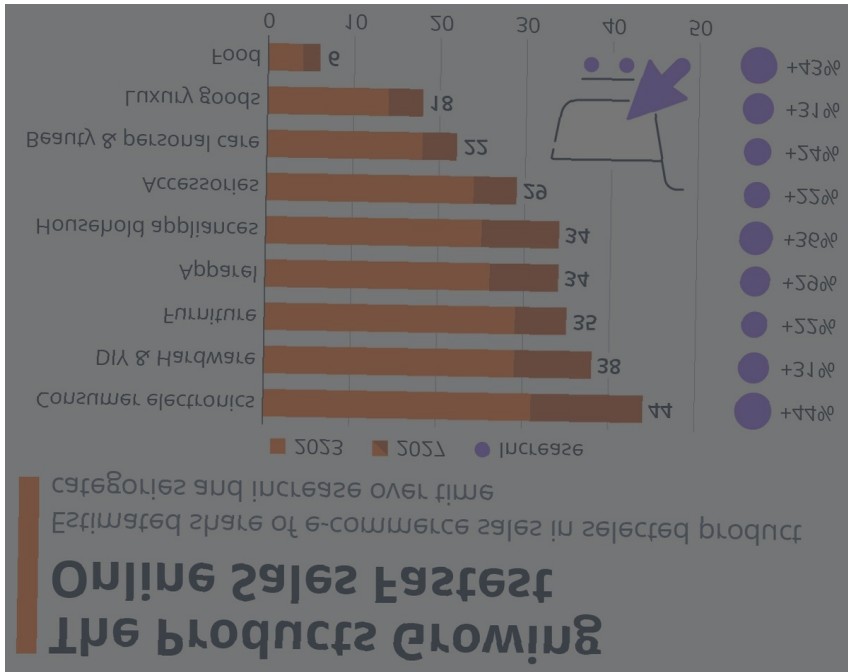

Original Image

**Processed Images**

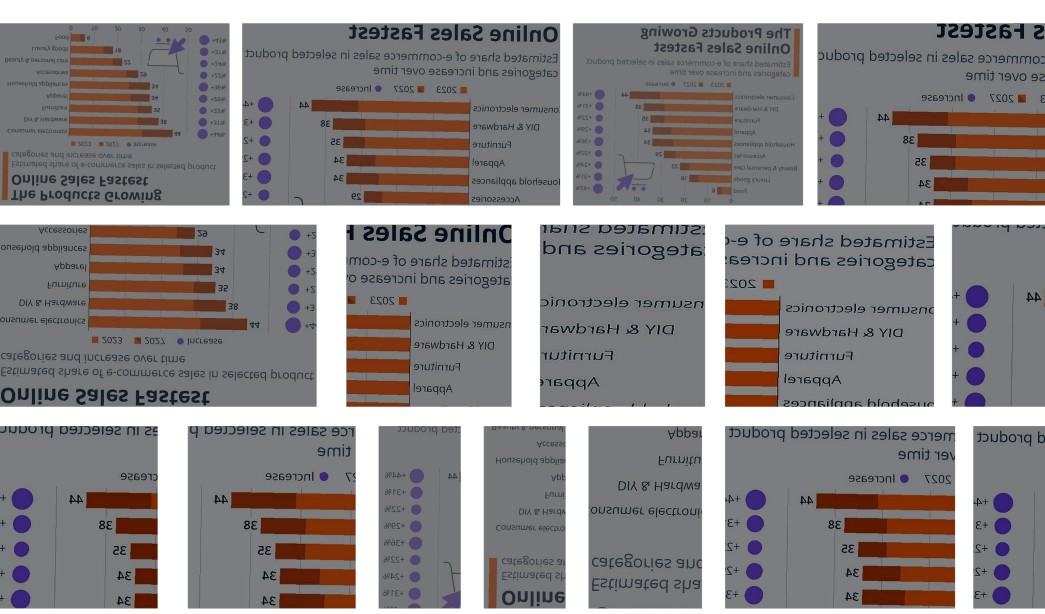

Transformed Images by GPT-5.

**Question**: If the player nearest to the goalkeeper in the image was moved to the 18-yard line instead of his current position, how many opposing players would be in an offside position? Assume that the black team is on defense.

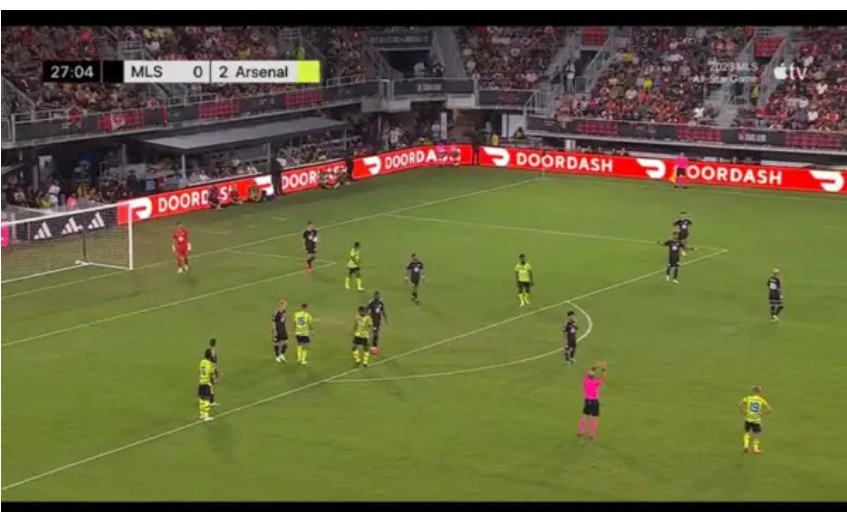

Original Image

**Processed Images**

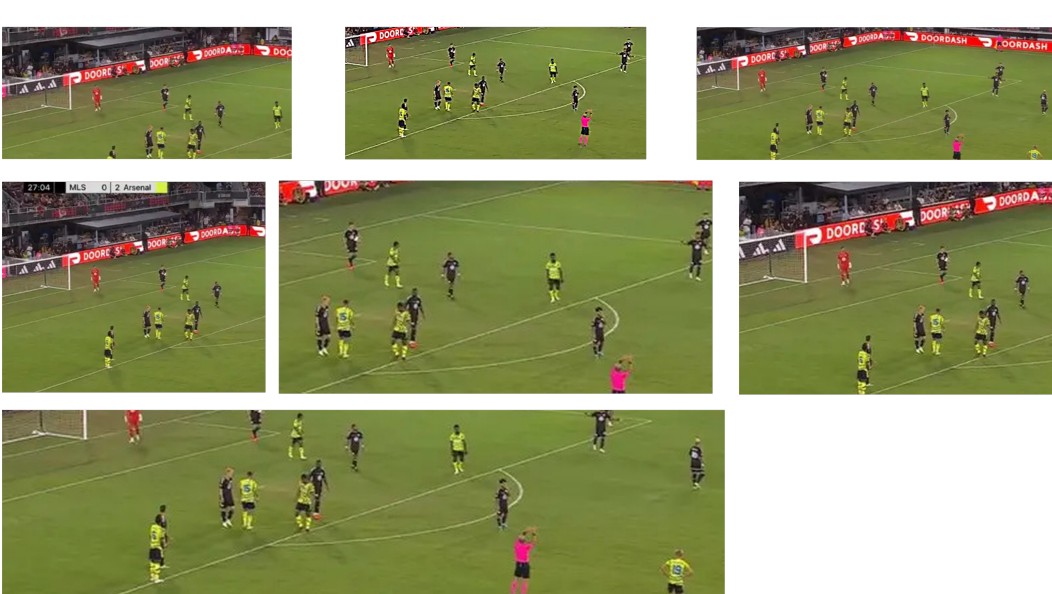

Transformed Images by GPT-5.

**Question**: A young male patient has an abnormal chest deformity. The patient is asymptomatic apart from the chest wall deformity, which has been present since birth. He has no family history of a similar condition. Various radiological imaging techniques were performed to assess the patient's chest deformity. Can you identify the abnormalities seen in the chest X-ray and the 3D volume rendering findings? Also, using the chest CT slides, identify the muscles and structures that are absent.

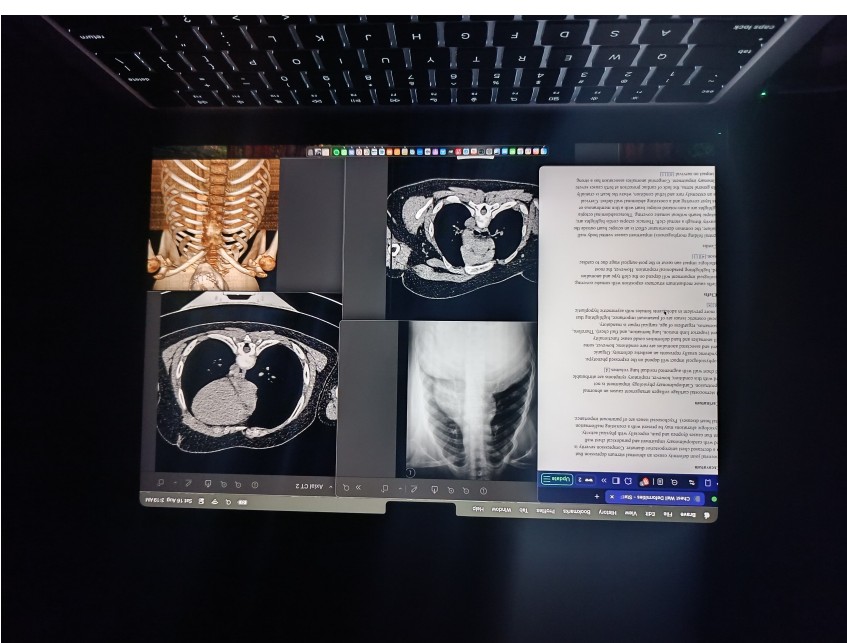

Original Image

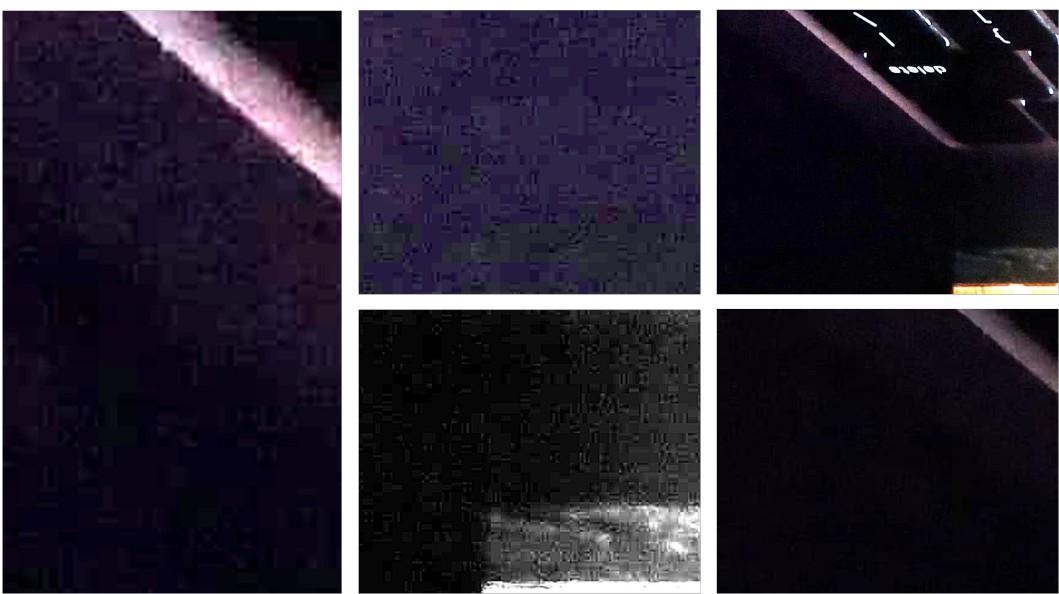

Transformed Images by Claude-opus-4.1.

**Question**: What is the molecular formula of the final product of the reaction? Also, what is the molecular formula of the trifluorinated compound employed in the initial reaction step?

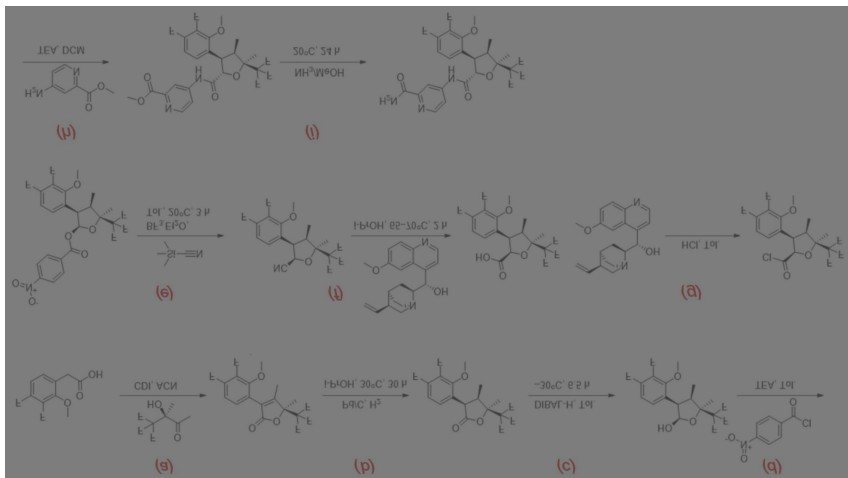

Original Image

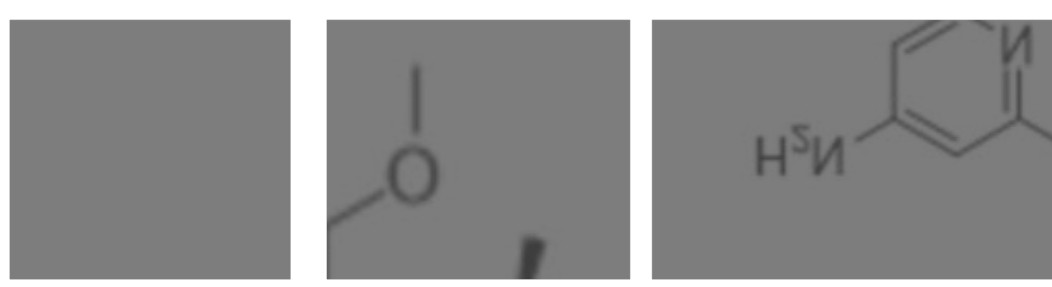

Transformed Images by Gemini-2.5-pro.

