# OpenReview forum: "Beyond Seeing: Evaluating Multimodal LLMs on Tool-Enabled Image Perception, Transformation, and Reasoning"
_ICLR.cc/2026/Conference — Submitted to ICLR 2026_

### Official Review · Reviewer_64fz · 2025-10-26

**Soundness:** 2
**Presentation:** 3
**Contribution:** 2
**Rating:** 4
**Confidence:** 3

**Summary:**

This paper introduces VISTOOLBENCH, a novel benchmark designed to evaluate MLLMs under the think with images paradigm. The benchmark comprises 1,204 open-ended vision tasks across five domains. Through rigorous evaluation of 16 representative MLLMs, the paper shows that current models struggle with integrating vision and tools and identifies divergent tool-use behaviors across models.

**Strengths:**

1. The paper is well-written and presented. The questions in VISTOOLBENCH are well-designed and carefully collected.
2. The evaluation is both comprehensive and detailed. VISTOOLBENCH devises a nuanced, rubric-based evaluation framework that moves beyond simple binary accuracy. This method enables fine-grained assessment across multiple dimensions and successes of each model's reasoning process.
3. The empirical results and analysis are valuable for the tool use of MLLMs. The paper benchmarks the performance of 16 SOTA models and uncovers critical tool-use behaviors between different models. These findings offer actionable insights for improving MLLMs' ability to strategically manipulate visual information and reason effectively in complex, real-world applications.

**Weaknesses:**

1.	The paper emphasizes the importance of tool use. However, the error analysis section 3.3 identifies model weaknesses such as visual perception errors, reasoning mistakes and calculation errors that do not appear to directly stem from tool usage. These error types are generally applicable to almost all multimodal evaluation tasks, regardless of whether tools are involved or successfully utilized.
2.	In Figure 7, the experiments show that Gemini 2.5 Pro actually performs 2.7% better after the tool is removed. The paper attributes this counterintuitive result to its “stronger native visual capabilities” and “possibly limited exposure to tool-use training.” This intriguing anomaly warrants deeper investigation. Did the model make incorrect tool calls, or did tool integration interfere with its otherwise strong visual reasoning process? Furthermore, since the effectiveness seems heavily influenced by the strength of the system prompt, are the experimental results sufficiently generalizable and robust?
3.	It would be valuable to include results from more recent open-source models such as Qwen3-VL, as well as tool-augmented open-source models like DeepEyes [1] et al.
4.	The paper selects tasks that models like o3 and o4-mini cannot handle, which may raise potential concerns about selection bias when arguing that GPT-5-think and other models exhibit lower success rates.

[1] DeepEyes: Incentivizing “Thinking with Images” via Reinforcement Learning, 2025.

**Questions:**

see weaknesses

---

> ### Author Response · Authors · 2025-11-21
>
> We thank the reviewer for the thoughtful comments and we address the reviewer's comment below:
>
> ### Fine-grained Vision Tool Failure Analysis
>
> We thank the reviewer for raising this point. In response, we conducted a fine-grained analysis of vision–tool-use failure cases. The details are provided in General Response Part I, and the full breakdown is included in Appendix B.7 of the revised paper (highlighted in blue).
>
> ### Ablation Study on Gemini 2.5 Pro
>
> We thank the reviewer for highlighting this intriguing observation regarding Gemini-2.5-Pro. In Figure 7, Gemini-2.5-Pro performs 2.7% better without tool use, a counterintuitive result given the expectation that tools should enhance visual processing. Below, we present a detailed analysis that clarifies when tools help and when they harm Gemini-2.5-Pro’s performance. To better understand this behavior, we conducted a fine-grained comparison between the tool and no-tool settings. For the single-turn evaluation: out of 603 tasks, Gemini-2.5-Pro solves 100 tasks with tools and 116 tasks without tools. There are 63 tasks that are solved in both settings. Therefore, we focus on the two asymmetric sets: 1. **Using tools helps but no-tools fails** (37 tasks); 2. **No-tool using succeeds but using tools fails** (53 tasks) Below we analyze both cases in depth.
>
> - Tasks Solved Only When Using Tools (37 tasks). Among these 37 tasks, 30 involve the python_image_processing tool. After close inspection, we find that Gemini-2.5-Pro often benefits from tools by performing meaningful and sometimes multi-step visual transformations. The operations used are summarized below:
> | **Image Operations**                         | **Count** |
> |----------------------------------------------|-----------|
> | Image crop                                   | 9         |
> | Image rotation / flip                        | 28        |
> | Image editing                                | 3         |
> | Others (e.g., image color conversion)        | 6         |
> | Contrast enhancement                         | 4         |
> | Brightness enhancement                       | 3         |
>
>
> - Tasks solved Only Without Tools (53 tasks). This set is more revealing. When tools hurt performance, the underlying issue is not coding errors but tool-induced distraction or incorrect operational decisions, which ultimately degrade reasoning accuracy. We categorize the failure reasons as follows:
> | **Failure Reason**                                                  | **Count** |
> |---------------------------------------------------------------------|-----------|
> | Distraction (tools distracted model from correct reasoning)         | 20        |
> | Tool execution error                                                | 11        |
> | incorrect_tool_selection                                            | 10        |
> | wrong_operation_flipping/rotation                                   | 6         |
> | wrong_operation_cropping                                            | 6         |
> | OCR_error                                                           | 5         |
> | wrong_operation_enhancing                                           | 1         |
> | others                                                              | 8         |
>
> The most common failure pattern is distraction, the model initiates unnecessary tool calls that lead it away from an otherwise correct direct answer. Additionally, several errors arise from incorrect or suboptimal image operations, such as improper cropping or flipping. We have provided more qualitative examples in the Appendix F.1 to further illustrate these cases.

---

> ### Author Response · Authors · 2025-11-21
>
> ### Add More Recent Open Source Models
>
> We appreciate the reviewer’s suggestion. We have expanded our baselines to include the latest frontier models: Gemini-3-pro, Claude-sonnet-4.5, and Claude-sonnet-4.5-thinking, as well as mid-sized models (Qwen3-VL-30B, Qwen3-VL-235B) and smaller models such as Thyme-7B and Deepeyes-7B. Model performances and more discussions are provided in General Response Part II.
>
> ### Potential Selection Bias
>
> We appreciate the reviewer’s concern about possible selection bias arising from choosing tasks that current frontier models such as o3 and o4-mini struggle with. We would like to first highlight that task construction was **model-agnostic**. The contributors designed tasks based on real-world, tool-requiring scenarios, not on observing any particular model’s failures. We just use o3, o4-mini, and Gemini-2.5-pro as a preliminary stage to select the most challenging tasks. More importantly, to reduce selection bias, each task within VisToolBench went through a **multi-stage human pipeline** including
>
> (i) rigorous annotation of ground-truth answers and reference tool-using trajectories, and
>
> (ii) multiple rounds of review and revision by independent annotators and core organizers to filter out ill-posed, ambiguous, or overly adversarial instances.
>
> This process mitigates selection bias by ensuring that tasks are challenging yet solvable for a careful human with access to the same tools, rather than cherry-picked failure cases for specific models. It is also worth noting that **all evaluated models perform poorly on VisToolBench**, with GPT-5-think only relatively better than others; our claim is not that GPT-5-think is uniquely good, but that current tool-using MLLMs as a whole have substantial headroom on these carefully curated and human-validated tasks.

---

> > ### Author Response · Authors · 2025-11-24
> > **Request to review response**
> >
> > Dear reviewer 64fz,
> >
> > Thank you for taking the time to review our paper. Your comments have helped us improve the quality of the paper and we truly appreciate your feedback. This message is to request you to review our response and let us know if you have any further queries about our paper. Thank you once again for your valuable feedback!
> >
> > Best,
> > Authors of Submission1127

---

### Official Review · Reviewer_dBhG · 2025-10-27

**Soundness:** 3
**Presentation:** 2
**Contribution:** 3
**Rating:** 4
**Confidence:** 4

**Summary:**

The paper introduces VisToolBench, a benchmark designed to evaluate Multimodal Large Language Models (MLLMs) in active visual reasoning in terms of perceiving, transforming, and reasoning with images rather than treating them as static inputs. The benchmark includes 1.2k complex, open-ended tasks across five domains, with detailed evaluation rubrics. Experiments show that current MLLMs struggle with such tasks, achieving low performance. The results reveal significant variation in tool-use behavior across models, highlighting the need for better integration of visual perception and tool-based reasoning in future MLLMs.

**Strengths:**

1. The paper presents a large collection of complex VQA tasks with high-quality annotations spanning diverse topics. This dataset will serve as a valuable resource for future evaluations of perception and visual reasoning models.

2. The human-annotated process scores provides an excellent foundation for assessing the faithfulness of reasoning trajectories generated by multimodal LLMs.

3. The introduction of the “proactive” rubric is particularly valuable, as it captures a key aspect of tool-using and agentic LLM behavior that is essential for evaluating real-world applications.

**Weaknesses:**

1. The paper, "VisToolBench," lacks clarity on how tools are actually called, particularly regarding the “reference-tool-use chain” described in Section 2.2. This crucial implementation detail is also missing from the accompanying Hugging Face page. Based on the system prompts shown in Appendix C.2, the tool usage appears to rely solely on plain text prompts without any explicit tool-calling format or structured invocation instructions. Furthermore, Appendix D.1 is insufficient for understanding the system’s behavior — it only presents text-based tool trajectories, without showing concrete results or demonstrations of the tool operations.

2. The core motivation of the paper is to evaluate tool-using multimodal LLMs, yet the proposed dataset mainly consists of visual detail search tasks with perturbed images (flipped, etc). The paper does not sufficiently justify why tool use is necessary for these tasks. Tools like historical_weather may be essential for answering a question but there lacks a quantitative results. To strengthen the argument, it would be valuable to provide baseline comparisons where tool use is explicitly disabled (e.g., prompting closed-source MLLMs with “please do not call any tools”) or by evaluating vanilla models such as Qwen, LLaMA, or Gemma. This would better demonstrate the actual benefit and necessity of tool usage for the benchmark.

3. The two main criteria, APR and ARS, primarily evaluate the model's accuracy and the faithfulness of its CoT trajectory. They do not directly assess the quality or correctness of the tool-using process itself (e.g., tool selection, argument formulation, or interpreting tool output). The evaluation is thus mismatched with the paper's stated goal of evaluating tool-using MLLMs.

**Questions:**

My concerns on the main topic of tool-using is elaborated in Weakness.

---

> ### Author Response · Authors · 2025-11-21
>
> We thank the reviewer for the thoughtful comments and we address the reviewer's comment below:
>
> ### How the Tools are Actually Called
>
> We thank the reviewer for raising this important clarification. Below, we provide detailed responses addressing the concerns.
>
> **Clarifying the “Reference Tool-Use Chain”**
>
> We apologize for the confusion. The reference tool-use chain is not how models call tools. Instead, it is authored by human contributors to illustrate how a human would likely use tools to solve the visual reasoning task. We emphasize that this reference chain is never exposed to models during evaluation. We will revise the paper to make this distinction clearer.
>
> **Tool-Call Implementation Details**
>
> Model tool-use is executed through **structured function-callings, not through plain text prompts**. As described in Section 3  Evaluation Setup paragraph, we use the function-calling interfaces provided by modern MLLMs, implemented through LiteLLM [1]. Specifically: Each tool is registered as a JSON-schema function, then the model receives the full tool schema in the system message so that the model will know which tools are available.
>
> The model outputs a structured function-call if it selects a tool as below:
>
> ```json
> {
>   "tool": "python_image_processing",
>   "tool_input": {
>     "code": "{the image processing code written by the model}"
>   }
> }
> ```
> Then the function callings are executed, producing: transformed images (for vision tools), or textual information (for non-vision tools, such as web_search). The tool result will be appended to the model's input messages. Then models will decide whether to issue another function call, or produce its final answer. This process repeats until the model stops calling tools or reaches a pre-defined maximum number of tool calls. Therefore, our tool-use implementation is not plain-text prompting, but a formal and reproducible function-call mechanism. We have included several **models' full responses including tool call, tool-call observations, and final answers** in Appendix F.
>
> [1] LiteLLM Documentation: Function Calling Interface. https://docs.litellm.ai/docs/completion/function_call.
>
> ### Why Tool Use is Necessary for VisToolBench
>
> We thank the reviewer for raising this important question regarding the necessity of tool use in our benchmark. VisToolBench indeed requires visual tools to solve the task. In particular:
>
> - We explicitly instruct all the contributors to design tasks for such that visual tool use is inherently required from a human problem-solving standpoint. As a result, we have collected a large amount of images that are information dense, or with awkward angles, or under poor lighting conditions, etc. Therefore, the images contained in VisToolBench are more realistic, visually challenging, and necessary vision tool-use is necessary to reveal the key visual details. We refer the reviewer for more benchmark images in Appendix D.2 and D.3 in the revised paper.
>
> - Our experimental results show that frontier models **actively choose to use vision tools** towards solving the tasks. Specifically, Figure 5 in the main paper shows that three frontier MLLMs (GPT-5, Gemini-2.5-Pro, and o3) heavily invoke python_image_processing tool. Since the tool invocation is optional for these models, this consistent behavior indicates that the models themselves *believe* that using vision tool is needed to solve the tasks, reinforcing that VisToolBench indeed targets vision tool requiring reasoning.
>
> - To directly evaluate the necessity of tool use, we indeed have conducted an ablation where tool usage is completely disabled. As shown in Figure 7, GPT-5 exhibits a **14.4% performance drop** without access to visual tools. This experiment directly addresses the reviewer’s suggestion: disabling tool use results in worse performance, demonstrating that tools are essential for solving many benchmark tasks.
>
> ### Fine-grained Vision Tool Analysis
>
> We thank the reviewer for raising this point. In response, we conducted a fine-grained analysis of vision–tool-use failure cases. The details are provided in General Response Part I, and the full breakdown is included in Appendix B.7 of the revised paper (highlighted in blue).

---

> > ### Author Response · Authors · 2025-11-24
> > **Request to review response**
> >
> > Dear reviewer dBhG,
> >
> > Thank you for taking the time to review our paper. Your comments have helped us improve the quality of the paper and we truly appreciate your feedback. This message is to request you to review our response and let us know if you have any further queries about our paper. Thank you once again for your valuable feedback!
> >
> > Best,
> > Authors of Submission1127

---

### Official Review · Reviewer_5rAQ · 2025-10-30

**Soundness:** 3
**Presentation:** 3
**Contribution:** 3
**Rating:** 4
**Confidence:** 4

**Summary:**

This paper presents **VISTOOLBENCH**, a novel benchmark designed to evaluate Multimodal Large Language Models (MLLMs) under the "think with images" paradigm—an important shift from existing benchmarks that treat images as static inputs ("think about images"). The work addresses a critical gap in real-world MLLM deployment: dynamic, tool-augmented reasoning over imperfect or complex visual inputs. The benchmark’s design (task diversity, rigorous data collection, nuanced evaluation metrics) and comprehensive experimental analysis (16 models, tool-use behavior, error diagnosis) make it a valuable contribution to multimodal AI research.

**Strengths:**

1.  Most existing benchmarks are limited to "static image input + passive reasoning". However, in real-world scenarios, user images often have issues such as rotation and underexposure, requiring dynamic tool operations (cropping, enhancement, etc.) to extract key information. The "think with images" paradigm proposed in the paper transforms visual input from a "passive context" into an "operable cognitive space", addressing the core pain points in the practical application of MLLMs.

2. The tasks span 5 domains: STEM, healthcare, finance, sports, and general use, encompassing both single-turn (603) and multi-turn (601) interactions. Moreover, the task design simulates real user needs (such as menu price calculation and medical image interpretation), avoiding the limitations of synthetic scenarios.

3. High-quality tasks are selected through a multi-stage process of "contributor training → initial design → model pre-evaluation (to ensure task difficulty) → two rounds of review".

**Weaknesses:**

1. The "specific scenario coverage" of tasks in various domains is not clearly defined (for example, in the medical field, only "medical-related tasks" are mentioned, without specifying whether sub-scenarios such as medical image interpretation and medical record image analysis are included), which may affect the evaluation of the benchmark's adaptability to different scenarios.
2. The distribution of "image types" in the task (such as the proportion of photos, charts, screenshots, etc.) is not specified. If a certain type of image accounts for a too high proportion, it may cause the model evaluation results to be biased towards the processing ability of that type of image rather than the general visual tool ability.
3. The paper points out that "visual perception errors account for the highest proportion," but it does not further break them down into subcategories such as "regional positioning errors (incorrect ROI cropping)," "insufficient image enhancement (failure to brighten key information)," and "OCR text extraction errors," making it impossible to accurately identify the shortcomings in the model's use of visual tools.
4. The human-LLM consistency for subjective indicators (such as "answer clarity") is only 73.96%-81.77%, and there is no explanation on how to handle such inconsistencies (e.g., whether to adopt multi-LLM voting or optimize prompts), which may affect the credibility of the scoring results.

**Questions:**

The VISTOOLBENCH benchmark proposed in this paper fills the gap in evaluating the "thinking with images" capability of MLLMs. The research design is innovative and practical, and the experimental analysis is in-depth. If the transparency of tasks, evaluation metrics, and experimental fairness can be optimized according to the above suggestions, the academic value and persuasiveness of the paper will be further enhanced.

---

> ### Author Response · Authors · 2025-11-21
>
> We thank the reviewer for the encouraging review and we address reviewer's concern below:
>
> ### The "Specific Scenario Coverage" of Tasks is not Clearly Defined
>
> We thank the reviewer for raising this important point. In Figure 1, we have shown our domain taxonomy with five general domains and the corresponding sub-domains. Due to the space limitation, the full results are not missing. We thank the reviewer for pointing this out and to address reviewer’s concern, we provide the full distribution of tasks across all subdomains below (this table is now available in Appendix D.1 in the revised paper):
>
> | **Domain**     | **Subdomain**             | **Number of Tasks** |
> |----------------|---------------------------|----------------------|
> | **STEM**       | Engineering               | 44                   |
> |                | Physics                   | 49                   |
> |                | Maths                     | 49                   |
> |                | Biology                   | 52                   |
> |                | Chemistry                 | 44                   |
> | **Finance**    | Market Analysis           | 77                   |
> |                | Personal Finance          | 90                   |
> |                | Corporate Finance         | 50                   |
> |                | Currency Trends           | 14                   |
> |                | Taxation Analysis         | 12                   |
> | **Medical**    | Clinical Specialties      | 42                   |
> |                | Laboratory Medicine       | 51                   |
> |                | Medical Imaging           | 113                  |
> |                | Pharmaceutical Science    | 25                   |
> |                | Public Health             | 7                    |
> | **Sports**     | Team Sports               | 147                  |
> |                | Individual Sports         | 50                   |
> |                | Racket Sports             | 15                   |
> |                | General Sports            | 28                   |
> | **Generalist** | Daily Life                | 31                   |
> |                | Transportation            | 38                   |
> |                | Game & Entertainment      | 86                   |
> |                | Problem Solving           | 86                   |
> |                | Support & Services        | 4                    |
> | **Total**      | ---                       | **1,245**            |
>
> ### The Distribution of Image Types
>
> Below we show the distribution of image types of VisToolBench. It can be seen that our benchmark contains diverse image types and majority of them are comes from real-world photos. We have also included representative images from each domain in Appendix D.3 (Figure 10 to Figure 14) in the revised paper.
>
> | **Image Type**               | **Count** | **Description** |
> |------------------------------|-----------|------------------|
> | **Real-World Photos**        | 1108      | Natural photographic images of real objects, scenes, people, or environments. |
> | **Screenshots**              | 945       | Captured computer or mobile screens, including UIs, apps, code editors, and web pages. |
> | **Charts and Plots**         | 492       | Line charts, bar charts, scatter plots, scientific graphs, and other quantitative figures. |
> | **Medical Imagery**          | 348       | Clinical or biomedical images such as X-rays, MRIs, microscopy, or diagnostic visuals. |
> | **Tables (Rendered Images)** | 328       | Images of tabular data from reports, spreadsheets, or financial summaries. |
> | **Diagrams**                 | 283       | Flowcharts, block diagrams, circuit diagrams, system schematics, and conceptual visuals. |
> | **Documents**                | 239       | Scanned or photographed text documents, forms, notes, receipts, or printed pages. |
> | **Scientific Content Images**| 232       | Equations, geometric figures, annotations, physics setups, or mathematical visuals. |
> | **Maps**                     | 57        | Geographical, navigation, or floor-plan style maps. |
> | **Synthetic / Rendered Images** | 72    | 2D/3D computer-generated graphics, CAD images, or simulation renderings. |
> | **Others**                   | 12        | Images that do not fit into the defined categories or contain mixed content. |

---

> ### Author Response · Authors · 2025-11-21
>
> ### Fine-grained Vision Tool Analysis
>
> We thank the reviewer for raising this point. In response, we conducted a fine-grained analysis of vision–tool-use failure cases. The details are provided in General Response Part I, and the full breakdown is included in Appendix B.7 of the revised paper (highlighted in blue).
>
> ### LLM-as-Judge for Subjective Rubrics
>
> We thank the reviewer for the constructive feedback. Across all rubrics, over 92% are objective and only 8% are subjective, as VisToolBench is primarily designed to evaluate MLLMs’ vision reasoning ability, and most rubrics focus on whether the model answers the question correctly or obtain the correct visual details, which are all belong to objective rubrics.
>
> As the reviewer suggested, to further strengthen the LLM-as-judge protocol on subjective rubrics, we compared our primary judge model, o4-mini, with a majority-vote ensemble consisting of three judge models: o4-mini, GPT-4.1, and GPT-4o. The results show that majority voting provides a meaningful gain, improving human–model alignment on subjective evaluations by 5.32%. The detailed objective and overall rubric results are summarized below:
>
> | **Judging Strategy** | **Overall** | **Objective** | **Subjective** |
> |----------------------|-------------|---------------|----------------|
> | **o4-mini**          | 0.8807      | 0.9170        | 0.7396         |
> | **Majority vote**    | 0.8860      | 0.9149        | 0.7928 (**↑ 5.32%**) |

---

> > ### Author Response · Authors · 2025-11-24
> > **Request to review response**
> >
> > Dear reviewer 5rAQ,
> >
> > Thank you for taking the time to review our paper. Your comments have helped us improve the quality of the paper and we truly appreciate your feedback. This message is to request you to review our response and let us know if you have any further queries about our paper. Thank you once again for your valuable feedback!
> >
> > Best,
> > Authors of Submission1127

---

### Official Review · Reviewer_97Na · 2025-10-31

**Soundness:** 3
**Presentation:** 3
**Contribution:** 3
**Rating:** 6
**Confidence:** 3

**Summary:**

This paper focus on accessing the ability of Large Vision-Language Models (LVLMs) to "think with images"—that is, to actively manipulate images (e.g., crop, edit, enhance) and integrate other tools to solve complex problems. The authors introduce a new benchmark named VISTOOLBENCH, specifically designed to evaluate MLLMs under the "think with images" paradigm. Tasks are designed to necessitate the simultaneous use of "vision tools" (like image processing libraries) and "general-purpose tools" (like web search, calculators, and Python interpreters) for their solution. It employs a detailed, rubric-based evaluation system instead of simple binary (correct/incorrect) judgments to provide more nuanced diagnostics.

**Strengths:**

1. The paper accurately identifies a critical gap in current MLLM evaluation. The distinction between "thinking about" vs. "thinking with" images is clear and crucial, as the latter is vital for the real-world deployment of MLLMs (handling imperfect, user-provided photos).
2. The design of VISTOLLBENCH is comprehensive. It contains 1,204 tasks across 5 professional domains, providing sufficient volume and broad coverage.
3. Moving beyond binary accuracy, the use of weighted rubrics (assessing understanding, truthfulness, reasoning, etc.) allows for a deeper diagnosis of where models fail.
4. The experimental results (SOTA model at 18.44% pass rate) are striking and clearly expose the shortcomings of current models. The divergence between GPT and Gemini in tool-use benefit is a particularly interesting and thought-provoking insight.

**Weaknesses:**

1. **Confation in Visual Tool Design:** The core "vision tool" is python_image_processing, which requires the MLLM to generate Python (PIL/OpenCV) code to perform image operations. This design conflates two different capabilities: A) The model's "reasoning ability" for visual operations (knowing what to do, e.g., "crop the menu area") and B) The model's "programming ability" (being able to write correct Python code, e.g., img.crop((x1, y1, x2, y2))). A model might be strong in A but weak in B, leading to task failure. A cleaner design would provide "atomic" tools (e.g., crop(region_name) or enhance(type)) to more purely evaluate visual reasoning, not programming.

2. **Vagueness in Error Attribution:** The paper attributes 70%-80% of failures to "Visual Perception Error," which is an overly broad and potentially misleading conclusion. Combined with point 1, this "perception error" is not fine-grained. Does it mean:

- (a) The model failed to realize it needed a tool? (This is a reasoning failure.)

- (b) The model realized it, but wrote incorrect code for the tool? (This is a tool-use failure.)

- (c) The model successfully executed the tool (e.g., correct crop), but still couldn't see the details in the transformed image? (This is a true perception failure.)

The paper does not (or cannot) clearly decouple these scenarios, making it difficult to pinpoint the true bottleneck.

3. **Lack of Comparison to Relevant SOTA Models:** The paper claims to evaluate the tool-use capabilities of MLLMs, but the main results lack an evaluation of several SOTA models that have already explored this domain (tool-calling and active perception).
- Thyme: Think Beyond Images
- Reinforced Visual Perception with Tools
- Qwen3-VL
- Seed1.6

**Questions:**

See above

---

> ### Author Response · Authors · 2025-11-21
>
> We sincerely thank the reviewer for the positive review and thoughtful comments, we address the reviewer's concerns below:
>
> ### Conflation in Visual Tool Design
>
> Thank you for the insightful comment. We address the reviewer’s concern by explaining our choice of the general python_image_processing tool and by validating this design through a new ablation study, as suggested by the reviewer.
>
> **Why do we use a general python_image_processing tool?**
>
> Our benchmark is intentionally designed to challenge frontier MLLMs such as GPT, Claude, and Gemini models, which are extensively trained on code and have demonstrated strong coding capabilities in their previous evaluations. In addition, these models also achieve very high tool-use success rates in VisToolBench, suggesting that failure cases rarely stem from Python syntax issues. Instead, the primary difficulty lies in reasoning about what visual transformation is needed, not in writing the code to execute it.
>
> We fully agree with the reviewer that atomic tools can reduce dependence on coding ability. However, they restrict the space of permissible image manipulations and risk biasing the evaluation toward a fixed set of predefined operations as well. This, in turn, limits the potential expressiveness of the frontier MLLMs. In contrast, by providing a general python_image_processing tool, the models can be creative and to compose multi-step or complex image manipulations. This indeed occurs in practice: top-performing models (e.g., GPT-5, o3) frequently perform diverse, multi-stage, and sophisticated operations that go far beyond typical atomic-tool functionality (see quantitative results in Figure 6 in the main paper and qualitative examples of this in Appendix E.4 and Appendix F.2).
>
> **New ablation study (requested by reviewer)**
>
> We greatly appreciate the reviewer’s suggestion to empirically evaluate the effect of decoupling coding ability from visual reasoning. Following the feedback, we conducted a new ablation comparing:
> - an atomic tool setting (with ten predefined tools such as crop_image, rotate_image, etc.), and
> - the original general Python image-processing setting
> on GPT-5 and GPT-4.1 on single turn tasks. The ARP results (%) are shown below:
>
> | Model  | Setup        | Overall | STEM   | Medical | Finance | Sports | Generalist |
> |--------|--------------|---------|--------|---------|---------|--------|------------|
> | **GPT-5** | General vision tool | 23.88  | 29.31 | 23.14  | 24.39  | 18.49 | 24.19     |
> |        | Atomic vision tools  | 18.15  | 20.69 | 14.05  | 17.07  | 18.49 | 20.95     |
> | **GPT-4.1** | General vision tool | 5.97   | 6.03  | 10.74  | 1.63   | 8.40  | 3.23      |
> |        | Atomic vision tools  | 6.30   | 6.90  | 11.57  | 1.63   | 7.56  | 4.03      |
>
> Key Observations
>
> - For GPT-4.1, the atomic tool setup provides a small improvement. However, the gain is marginal, suggesting that the model’s failures stem mainly from visual reasoning rather than from coding issues.
>
> - On the other hand, GPT-5 shows a performance decline under atomic tools, which we hypothesize arises because atomic tools restrict the model’s expressive image manipulation capabilities. GPT-5 frequently composes multi-step and creative operations in the general setting; atomic tools constrain this flexibility and can reduce its performance.
>
> Detailed results are now included in Appendix E.2 in the revised paper. We agree that supporting both atomic and general vision tools enables different evaluation goals and can be benefit for smaller models. Our evaluation toolkit is highly modular, and we will extend it to include the proposed atomic tools. Evaluators or model developers can easily enable or disable specific tools to serve their purpose.
>
> ### Vagueness in Error Attribution
>
> We thank the reviewer for raising this point. In response, we conducted a fine-grained analysis of vision–tool-use failure cases. The details are provided in General Response Part I, and the full breakdown is included in Appendix B.7 of the revised paper (highlighted in blue).
>
> ### Lack of Comparison to Relevant SOTA Models
>
> We appreciate the reviewer’s suggestion. We have expanded our baselines to include the latest frontier models: Gemini-3-pro, Claude-sonnet-4.5, and Claude-sonnet-4.5-thinking, as well as mid-sized models (Qwen3-VL-30B, Qwen3-VL-235B) and smaller models such as Thyme-7B and Deepeyes-7B. Model performances and more discussions are provided in General Response Part II.

---

> > ### Author Response · Authors · 2025-11-24
> > **Request to review response**
> >
> > Dear reviewer 97Na,
> >
> > Thank you for taking the time to review our paper. Your comments have helped us improve the quality of the paper and we truly appreciate your feedback. This message is to request you to review our response and let us know if you have any further queries about our paper. Thank you once again for your valuable feedback!
> >
> > Best,
> > Authors of Submission1127

---

> ### Comment · Reviewer_97Na · 2025-11-25
> **Reply to authors**
>
> Thanks for your response. My concerns are addressed. I will keep my score.

---

> > ### Author Response · Authors · 2025-11-25
> > **Thank you for your continued support!**
> >
> > Thanks a lot your continued positive support of VisToolBench. We’re delighted that our rebuttal has resolved all of your concerns. We would be truly grateful if you could reevaluate our work based on the clarifications we provided. Please let us know if you have further questions!

---

### Author Response · Authors · 2025-11-21
**General Response Part I**

We thank the reviewers for the thoughtful and constructive feedback. Reviewers highlighted that our work clearly identifies an important gap in current MLLM evaluation, distinguishing between **thinking about images** and **thinking with images**, and emphasized the value of enabling models to actively manipulate imperfect, real-world visual inputs [97Na, 5rAQ]. We also appreciate the recognition of the breadth and realism of VisToolBench, which contains 1,204 high-quality tasks across five domains and is supported by a careful, multi-stage human-annotated data collection and review process [97Na, 5rAQ, dBhG, 64fz]. We also appreciate the reviewers recognition of rubric-based grading that help reveal more detailed model behaviors beyond simple accuracy [97Na, 5rAQ, dBhG, 64fz]. Reviewers also found the empirical results informative, especially the consistently low pass rates of state-of-the-art models and the clear differences in tool-use patterns across systems, which point to concrete directions for improving MLLMs [97Na, 5rAQ, dBhG, 64fz]. We are grateful that the research design and experimental analysis were viewed as both innovative and practical [5rAQ]. Below we first address two main comments mentioned by multiple reviewers, and we will address each point raised by the reviewers separately.

### Fine-grained vision tool-use error analysis

We thank the reviewers for encouraging a more fine-grained examination of tool-use behaviors. Our primary metrics, APR and ASR, evaluate the correctness and faithfulness of the model’s final answers, and we additionally report three high-level tool-use metrics: tool-call success rate, tool-call proactivity, and tool-call volume. We fully agree that a deeper analysis of vision tool-use failures can offer more actionable insights for model development.

To this end, we have incorporated a detailed tool-use failure analysis covering **eleven categories** of vision tool-related errors, including visual perception mistakes (after vision tool call), incorrect cropping, incorrect rotation/flip operations, OCR errors, distractions, and incorrect tool selection, etc. To support large-scale evaluation, we prompt GPT-5 with the question, the model’s tool-use chain, the golden answer, and the model’s actual answer to generate an initial failure-category prediction. Human supervision is then applied to filter out any inappropriate or incorrect categories. As shown in the table below, visual perception remains the dominant failure mode across all models (GPT-5: 34.40%, Gemini-2.5-Pro: 32.56%, Claude-Opus-4.1: 39.46%), and errors such as incorrect cropping and execution failures also occur frequently (e.g., GPT-5: 14.68% cropping errors and 15.14% execution errors).

| Failure Type                 | GPT-5 (%) | Gemini-2.5-pro (%) | Claude-opus-4.1 (%) |
|-----------------------------|-----------|----------------------|-----------------------|
| Visual Misinterpretation    | 34.40    | 32.56              | 39.46               |
| OCR Error                   | 11.01    | 12.21               | 7.62                |
| Distraction                 | 8.26     | 4.07               | 10.76               |
| Tool Execution Error        | 15.14    | 12.79               | 6.28               |
| Incorrect Tool Selection    | 2.75     | 3.49               | 5.83                |
| Incorrect Cropping          | 14.68    | 12.21               | 17.94              |
| Incorrect Rotation/Flipping | 5.05     | 5.23                | 5.38                |
| Incorrect Enhancing         | 1.83     | 0.58                | 1.35               |
| Incorrect Editing           | 0.00     | 0.00               | 0.45                |
| Incorrect Resizing          | 0.46     | 0.00                | 0.00                |
| Other                       | 6.42     | 16.86               | 4.93            |
| **Total Count**  | **218**  |**172** | **223**    |

We emphasize that reasonably good visual perception is essential for enabling effective tool use in our benchmark. For example, some tasks require the model to first localize a small object or region. However, as our qualitative examples in show, models often fail to propose accurate cropping coordinates in one shot, leading to zooming into irrelevant regions, and this makes models to perform multiple unnecessary tools calls (see examples in Appendix F.3 in our revised paper), which may propagate downstream errors. We also included several model success examples, failure examples, and transformed images using vision tools in Appendix F.2 to F.4 in the revised paper.

Overall, we believe that this fine-grained error analysis provides process-level insight into where current MLLMs struggle, complementing our existing metrics and strengthening the evaluation framework. We have included this analysis in Appendix B.8 in the revised paper (highlighted in blue).

---

> ### Author Response · Authors · 2025-11-21
> **General Response Part II**
>
> ### More Baseline Models
>
> We appreciate the reviewer’s suggestion to include more baseline models. In response, we report results for a broader set of MLLMs below, including the most recent frontier models Gemini-3-pro, Claude-sonnet-4.5 and Claude-sonnet-4.5-thinking, mid-sized models Qwen3-VL-30B and Qwen3-VL-235B, and smaller models such as Thyme-7B and Deepeyes-7B, covering both commercial and open-source vision-language models as the reviewers suggested. The results for single-turn tasks are shown below (we include the results of GPT-5-think for comparison purposes).
>
> | Model                     | Overall | STEM   | Medical | Finance | Sports | Generalist |
> |---------------------------|---------|--------|---------|---------|--------|------------|
> | GPT-5-think               | 26.04   | 26.96  | 24.79   | 24.39   | 24.37  | 29.84      |
> | Gemini-3-pro              | **39.64** | **41.37** | **34.71** | **48.78** | **38.66** | **34.68** |
> | Claude-sonnet-4.5         | 7.40    | 5.26   | 10.99   | 3.70    | 7.59   | 8.70       |
> | Claude-sonnet-4.5-thinking| 8.68    | 9.47   | 12.12   | 5.68    | 4.55   | 10.99      |
> | Qwen3-VL-30B              | 1.00    | 0.00   | 1.65    | 0.81    | 1.68   | 0.81       |
> | Qwen3-VL-235B             | 1.66    | 0.00   | 5.79    | 0.00    | 2.52   | 0.00       |
> | Thyme-7B                  | 1.82    | 2.59   | 2.48    | 0.81    | 0.00   | 3.23       |
> | Deepeyes-7B               | 3.32    | 2.59   | 3.31    | 4.07    | 3.36   | 3.23       |
>
> We make the following observations:
>
> - The most recent model **Gemini-3-pro** achieves the current SOTA on VisToolBench, which exceed the GPT-5-think by a large margin on single-turn tasks (we have updated Table 3 and Table 6 with multi-turn results of Gemini-3-pro in the revised paper).
>
> - Models still achieve only modest performance: Claude-sonnet-4.5 models have overall APRs below 10%, and all mid-size to small-size models perform below 5%. Notably, some open-source models (Thyme-7B and Deepeyes-7B) that are trained specifically for the *think-with-image* paradigm also show poor performance, which is expected given their limited model size.
>
> - It is worth noting that Deepeyes-7B achieves a 3.32% overall APR, outperforming several larger models such as Llama-Maverick, Pixtral-large, Qwen3-VL-30B, and Qwen3-VL-235B, indicating that targeted training for vision-tool-use can be beneficial.
>
> Nevertheless, there remains substantial room for improvement before current models can fully saturate VisToolBench. The newly added results are highlighted in blue in Appendix B.1.

---

### Author Response · Authors · 2025-11-30
**Summary of the Rebuttal**

We would like to thank all the reviewers for their constructive feedback, which has greatly helped us improve our paper. VisToolBench is the **first multimodal benchmark** that evaluates how frontier multimodal language models (MLLMs) **think with images** to solve complex vision-reasoning tasks using tools (including vision tool for active image processing). This complements existing multimodal benchmarks, which mainly focus on **think about images** paradigm where models use visual inputs passively. Every task within VisToolBench was constructed by human experts and underwent multiple rounds of human review to ensure high-quality data samples. We also introduced rubric-based evaluations to reveal more detailed model behaviors beyond simple accuracy. Our results show that VisToolBench poses a significant challenge: even the best model achieves only modest overall performance (**below 30%**). We have addressed each reviewer’s concerns with new experimental results and expanded benchmark analyses. Below, we summarize the new results added during the rebuttal period.

1. **Fine-grained Tool-Use Error Analysis.** Reviewers requested a deeper analysis of vision tool-use failures. In response, we conducted a new tool-use failure study on three representative MLLMs (GPT-5, Gemini-2.5-pro, Claude-opus-4.1). Specifically, we expanded our analysis to cover **eleven categories** of vision tool-related errors, including visual misinterpretation, OCR errors, incorrect cropping, incorrect rotation/flipping, and more. The results show that **visual misinterpretation** (i.e., visual perception issues) remains the dominant failure mode, indicating that current models still need to learn how to use vision tools more effectively for key visual content extraction. Detailed results are provided in Appendix B.8, with additional illustrative examples in Appendix F.2 to F.4 of the revised paper.

2. **More Baseline Models.** Reviewers requested evaluations of additional recent models, including those specifically trained for vision tool use. Accordingly, we conducted new experiments on a broader range of MLLMs, including frontier models (**Gemini-3-pro, Claude-sonnet-4.5, Claude-sonnet-4.5-thinking**), mid-sized models (**Qwen3-VL-30B, Qwen3-VL-235B**), and smaller specialized models (**Thyme-7B, Deepeyes-7B**). Gemini-3-pro (released on Nov 18) achieves state-of-the-art performance on VisToolBench, while most other models still perform poorly. Notably, Deepeyes-7B outperforms several much larger models, suggesting that targeted training for vision tool use is beneficial. Full results are provided in Appendix B.1.

3. **Ablation Study on Vision-Tool Design.** We originally used a general vision tool (*python_image_processing*) that lets models write arbitrary image-processing code. As Reviewer 97Na noted, this may mix coding ability with tool-use performance. To isolate this factor, we replaced it with an **atomic vision-tool set** of ten predefined tools (e.g., *crop_image*, *rotate_image*). Under this setup, GPT-5’s performance drops, while GPT-4.1 shows only slight gains, suggesting that atomic tools may limit expressive image manipulation. This also indicates that **coding ability is not the bottleneck**, the key challenge is **using tools effectively to extract critical visual content**. Full results are provided in Appendix E.2 in the revised paper.

4. **More Benchmark Data Analysis.** As requested by Reviewer 5rAQ, we have included additional benchmark statistics, such as **sub-domain coverage** and **image-category distributions**, which are available in Appendix D of the revised paper. We have also added representative images from each domain in Appendix D.3 (Figures 10–14). These additions demonstrate that VisToolBench covers a **broad range of domains** and includes **diverse, realistic images derived from real-world photos**.

5. **More End-to-End Examples.** As suggested by Reviewer dBhG, we have discussed how models use tools via function calling and included end-to-end success and failure examples illustrating how the model solves tasks with tool callings. These examples can be found in Appendix F of the revised paper.

6. **More Analysis on Gemini-2.5-pro Ablation Study.** Our ablation study shows that Gemini-2.5-pro performs slightly better without tools. To understand this behavior, we conducted a detailed analysis of when tool helps or hurts Gemini-2.5-pro. The full discussion is provided in Appendix F.1 of the revised paper.

We have also addressed the remaining minor comments from each reviewer in the detailed rebuttal below. Overall, VisToolBench establishes a new and challenging testbed for evaluating MLLMs under the think-with-images paradigm.

---

### Meta-Review · Area_Chair_AvzA · 2026-01-12

**Summary:**

- The use of a general python_image_processing tool conflated visual reasoning with programming ability. A cleaner design with "atomic" tools is requested. There was also confusion about how tools were actually invoked and a lack of clarity on the "reference tool-use chain."

- A deeper, more fine-grained error analysis beyond the broad "Visual Perception Error" category is requested. Reviewers wanted breakdowns of specific failure types (e.g., incorrect cropping, OCR errors) to pinpoint true bottlenecks.

- The benchmark lacked evaluations of several relevant state-of-the-art models, including recent frontier models (Gemini-3, Claude-Sonnet-4.5), mid-sized open-source models (Qwen3-VL), and specialized tool-augmented models.

- A lack of detailed statistics on task sub-domain coverage and image type distribution, and an insufficient justification for why tool use was necessary for the benchmark tasks. The evaluation metrics (APR, ASR) may not directly assess the quality of the tool-using process itself.

- The finding that Gemini-2.5-Pro performed slightly better without tools needed deeper investigation. There was also a concern that task selection might introduce bias.

**Reviewer Concerns:**

Addressed Concerns:

- The authors conducted a new study categorizing vision-tool failures into some detailed types (e.g., Visual Misinterpretation, Incorrect Cropping, OCR Error) across three models.

- The authors added results for Gemini-3-pro, Claude-sonnet-4.5, Claude-sonnet-4.5, Qwen3-VL, Thyme-7B, and Deepeyes-7B.

- The authors conducted the requested ablation, replacing the general tool with a set of 10 atomic tools. The results supported their original design choice while empirically investigating the conflation issue.

- The authors provided full sub-domain task distributions and image-type statistics in the appendix.

- The authors clarified that tools are called via structured JSON function-calling APIs (not plain text) and that the "reference tool-use chain" is a human-authored guide, not shown to models.

- The authors point to their existing ablation study showing a performance drop for GPT-5 without tools, and the high frequency of tool use by models, as evidence of necessity. They also note the human-centric task design.

- The authors provided a detailed breakdown of tasks where tools helped vs. hurt.


Outstanding or Partially Addressed Concerns:

- The core evaluation metrics (APR, ASR) primarily judge the final answer and reasoning faithfulness, not the quality of intermediate tool selection and argument formulation.

**Reviewer Scores:**

The rebuttal is strong and addresses some majority of concrete, actionable concerns. All reviewers are likely to increase their scores, i.e., from 4 to 5/6.

---

### Decision · Program_Chairs · 2026-01-26

Reject